# STABLE, EFFICIENT, AND FLEXIBLE MONOTONE OPERATOR IMPLICIT GRAPH NEURAL NETWORKS

## ABSTRACT

Implicit graph neural networks (IGNNs) that solve a fixed-point equilibrium equation for representation learning can learn the long-range dependencies (LRD) in the underlying graphs and show remarkable performance for various graph learning tasks. However, the expressivity of IGNNs is limited by the constraints for their well-posedness guarantee. Moreover, when IGNNs become effective for learning LRD, their eigenvalues converge to the value that slows down the convergence, and their performance is unstable across different tasks. In this paper, we provide a new well-posedness condition of IGNNs leveraging monotone operator theory. The new well-posedness characterization informs us to design effective parameterizations to improve the accuracy, efficiency, and stability of IGNNs. Leveraging accelerated operator splitting schemes and graph diffusion convolution, we design efficient and flexible implementations of monotone operator IGNNs that are significantly faster and more accurate than existing IGNNs.

## 1 INTRODUCTION

Implicit graph neural networks (IGNNs) that solve a fixed-point equilibrium equation for graph representation learning can learn long-range dependencies (LRD) in the underlying graphs, showing remarkable performance for various tasks [69; 39; 58; 63; 22]. Let $G = (V, E)$ represent a graph, where $V$ is the set of nodes, and $E \subseteq V \times V$ is the set of edges. The connectivity of $G$ can be represented by the adjacency matrix $\boldsymbol{A} \in \mathbb{R}^{n \times n}$ with $A_{ij} = 1$ if there is an edge connecting nodes $i, j \in V$; otherwise $A_{ij} = 0$. Let $\boldsymbol{X} \in \mathbb{R}^{d \times n}$ be the initial node features whose $i$-th column $\boldsymbol{x}_i \in \mathbb{R}^d$ is the initial feature of the $i$-th node. IGNN [39] learns the node representation by finding the fixed point, denoted as $\boldsymbol{Z}^*$, of the Picard iteration below

$$\boldsymbol{Z}^{(k+1)} = \sigma\big(\boldsymbol{W}\boldsymbol{Z}^{(k)}\boldsymbol{G} + g_{\boldsymbol{B}}(\boldsymbol{X})\big), \ \text{for } k = 0, 1, 2, \cdots, \tag{1}$$

where $\sigma$ is the nonlinearity (e.g. ReLU), $g_{\boldsymbol{B}}$ is a function parameterized by $\boldsymbol{B}$ (e.g. $g_{\boldsymbol{B}}(\boldsymbol{X}) = \boldsymbol{B}\boldsymbol{X}\boldsymbol{G}$), matrices $\boldsymbol{W}$ and $\boldsymbol{B} \in \mathbb{R}^{d \times d}$ are learnable weights, and $\boldsymbol{G}$ is a graph-related matrix. In IGNN, $\boldsymbol{G}$ is chosen as $\hat{\boldsymbol{A}} := \hat{\boldsymbol{D}}^{-1/2}(\boldsymbol{I} + \boldsymbol{A})\hat{\boldsymbol{D}}^{-1/2}$ with $\boldsymbol{I}$ being the identity matrix and $\hat{\boldsymbol{D}}$ is the degree matrix with $\hat{D}_{ii} = 1 + \sum_{j=1}^{n} A_{ij}$. IGNN constrains $\boldsymbol{W}$ using a tractable projected gradient descent method to ensure the well-posedness of Picard iteration at the cost of limiting the expressivity of IGNNs. The prediction of IGNN is given by $f_{\Theta}(\boldsymbol{Z}^*)$, a function parameterized by $\Theta$. IGNNs have several merits: 1) The depth of IGNN is adaptive to particular data and tasks rather than fixed. 2) Training

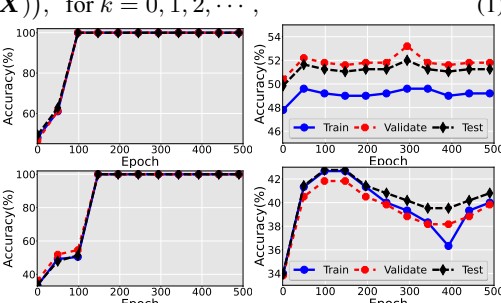

Figure 1: Epoch vs. training, validation, and test accuracy of IGNN for classifying directed chains. First row: binary chains of length 100 (left) and 250 (right). Second row: three-class chains of length 80 (left) and 100 (right).

IGNNs requires constant memory independent of their depth — leveraging implicit differentiation [66; 2; 51; 13]. 3) IGNNs have better potential to capture LRD of the underlying graph compared to existing GNNs, including GCN [75], GAT [73], SSE [23], and SGC [79]. The latter GNNs lack the capability to learn LRD as they suffer from over-smoothing [56; 84; 62; 20]. Several methods have been proposed to alleviate over-smoothing and hence improve learning LRD by adding residual connections [37; 21; 55], by geometric aggregation [65], by adding a fully-adjacent layer [3], by improving breadth-wise backpropagation [59], and by adding oscillatory layers [27; 67].

**Issue 1:  Well-posedness of IGNN Limits Its Expressivity.**  One bottleneck of IGNN is that the magnitude of $\boldsymbol{W}$'s eigenvalues has to be less than one for its well-posedness guarantee; see Sec. 2 for details. This limits the selection of $\boldsymbol{W}$ and thereby limits the expressivity of IGNNs.

**Issue 2:**  **When can IGNNs learn LRD?** To understand when IGNN can learn LRD, we run IGNN using the settings in [39] to classify directed chains. Directed chains is a synthetic dataset designed to test the effectiveness of GNNs in learning LRD for node classification [71; 39]. Fig. 1 plots epoch vs. accuracy of IGNN for the chain classification. Here, each epoch means iterating Equation (1) until convergence and then updating $\boldsymbol{W}$ and $\boldsymbol{B}$ at the end. IGNN can classify the binary chain task perfectly at length 100 but performs near random guesses when the length is 250, as illustrated in Fig. 1. For the three-class chains, IGNN's performance is very poor at chain length 100 but performs quite well at length 80. We investigate the results above by studying the dynamics of eigenvalues of the matrix $|\boldsymbol{W}|^1$. For illustrative purpose, we consider $\lambda_1(|\boldsymbol{W}|)$ and $\lambda_2(|\boldsymbol{W}|)$, the largest and the second largest eigenvalue of $|\boldsymbol{W}|$ in magnitude. Fig. 2 (left) contrasts the evolution of the magnitude of $\lambda_1(|\boldsymbol{W}|)$ and $\lambda_2(|\boldsymbol{W}|)$ of IGNN when classifying nodes on chains with different lengths. We see that the magnitude of both eigenvalues goes to 1 when IGNN becomes accurate. However, Fig. 2 (right) shows that IGNN takes many more iterations in each epoch when the magnitude of eigenvalues gets close to 1. Indeed, when $\lambda_1(|\boldsymbol{W}|) \to 1$, the Lipschitz constant of the linear map $\boldsymbol{W}\boldsymbol{Z}\boldsymbol{G} + g_{\boldsymbol{B}}(\boldsymbol{X})$ is close to 1, slowing down the convergence of the Picard iterations. The results in Fig. 2 echo our intuition; the representation of a given node aggregates one more hop of information after each Picard iteration; when the magnitude of eigenvalues gets close to 1, Equation (1) converges slowly so that IGNN can capture LRD before fixed point convergence.

We report the classification results of different lengths in Appendix I; these results show prevalently that IGNNs suffer from two bottlenecks: 1) An inherent *tradeoff between computational efficiency and capability for learning LRD*. 2) The performance of IGNNs, based on Picard iteration, is *unstable* in the sense that their performance varies substantially across tasks. In particular, starting from random Gaussian initialization of $\boldsymbol{W}$ — the default initialization of $\boldsymbol{W}$ — IGNN cannot learn LRD if none of the eigenvalues of $\boldsymbol{W}$ get close to 1 in magnitude.

## 1.1  OUR CONTRIBUTION

We develop accurate, stable, and efficient monotone operator IGNNs (MIGNNs)[2]. In particular, we derive a new well-posedness condition for MIGNN leveraging monotone operator theory; see Sec 2. The new well-posedness condition informs us to design 1) a *monotone parameterization* of $\boldsymbol{W}$, whose eigenvalues can take a much wider range than that of IGNNs, to boost the expressivity of MIGNNs, addressing **Issue 1**. And 2) a Cayley transform-based *orthogonal parameterization* of $\boldsymbol{W}$ to improve the stability and efficiency of MIGNN for learning LRD, addressing **Issue 2**; see Sec. 3. Picard iteration is inefficient or impossible to find the fixed point of

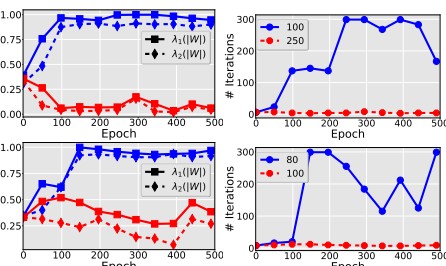

Figure 2:  Epoch vs. the magnitude of $\lambda_1(|\boldsymbol{W}|)$ and $\lambda_2(|\boldsymbol{W}|)$ and the iterations required for each epoch. First row: binary chains, second row: Three-class chains.

MIGNN with monotone or orthogonal parameterization. As such, we implement MIGNNs leveraging Anderson-accelerated operator splitting schemes; see Sec. 4. We verify the efficacy of MIGNN on various benchmark tasks; see Sec. 5.

## 1.2  ADDITIONAL RELATED WORK

We briefly review some representative related works in three directions: deep equilibrium models (DEQs), GNNs, and orthogonal parameterizations for recurrent neural networks (RNNs).

**DEQ.** IGNN is related to DEQs [7; 26; 8], but the equilibrium equation of IGNN differs from DEQs in that IGNN encodes graph structure. DEQs are a class of infinite depth weight-tied feedforward neural networks with forward propagation using root-finding and backpropagation using implicit differentiation. As a result, training DEQs only requires constant memory independent of the network's depth. Monotone operator theory has been used to guarantee the convergence of DEQs [77] and to improve the robustness of implicit neural networks [44]. The convergence of DEQs has also

---

[1]The matrix $|\boldsymbol{W}|$ is obtained by taking the entry-wise absolute value of the matrix $\boldsymbol{W}$.

[2]Starting from here, we use MIGNN to stress that the model is based on monotone operator theory.

been considered by constraining the network's weights [49]. Linearized DEQs are studied in [46]. Jacobian regularization has been used to stabilize the training of DEQs [9]. Anderson-accelerated DEQs with learned acceleration-related hyperparameters are also proposed [10].

**Graph neural networks.** Classical GNNs are defined by stacking explicitly defined graph filtering layers. Examples include graph convolutional networks (GCNs) [17; 24; 48], recurrent GNNs [38; 30; 57; 21] GraphSAGE [40], neural graph fingerprints [25], graph isomorphism network (GIN) [80], message passing neural networks [36], graph attention networks (GATs) [73], GCNs with convolution kernels learned based on paths (PAN [60] and pathGCN [28]), and higher-order message passing networks [15; 14]. There are some recent advances in IGNNs: EIGNN removes the nonlinearity in each intermediate iteration and derives a closed form of the infinite iterations [58], convergent graph solver (CGS) is an IGNN model with convergence guarantees by constructing the input-dependent linear contracting iterative maps [63], GIND leverages implicit nonlinear diffusion to access infinite hops of neighbors [22]. In addition to Picard iteration, implicit GNNs have also been defined by parametrizing the diffusion equation on graphs, see e.g. [18; 72; 19].

**Orthogonal parameterization for deep learning.** The fixed point iteration Equation (1) is related to the hidden state updates of RNNs [66; 29; 2; 50]. Learning LRD is challenging for RNNs due to exploding and vanishing gradient during backpropagation through time [76; 12; 64]. Enforcing orthogonal parameterization for RNNs is an effective approach to overcome exploding and vanishing gradients, benefiting RNNs for learning LRD [5; 78; 45; 74; 61; 41].

### 1.3 NOTATION

We denote scalars by lower- or upper-case letters and vectors/matrices by lower-/upper-case boldface letters. For a vector $\boldsymbol{a}$, we use $\|\boldsymbol{a}\|/\|\boldsymbol{a}\|_\infty$ to denotes its $\ell_2$-/$\ell_\infty$-norm. We use $\boldsymbol{I}$ to denote the identity matrix whose dimension can be inferred from the context. For a matrix $\boldsymbol{A}$, we denote its transpose as $\boldsymbol{A}^\top$, its inverse as $\boldsymbol{A}^{-1}$, its Frobenius norm/2-norm/$\infty$-norm as $\|\boldsymbol{A}\|_F/\|\boldsymbol{A}\|/|\boldsymbol{A}\|_\infty$, and we denote its $i$-th largest eigenvalue in magnitude as $\lambda_i(\boldsymbol{W})$. Given two matrices $\boldsymbol{A}$ and $\boldsymbol{B}$, we denote their Kronecker/entry-wise product as $\boldsymbol{A} \otimes \boldsymbol{B}/\boldsymbol{A} \odot \boldsymbol{B}$, and denote $\boldsymbol{A} \succ \boldsymbol{B}$ ($\boldsymbol{A} \succeq \boldsymbol{B}$) if $\boldsymbol{A} - \boldsymbol{B}$ is positive definite (semi-positive definite). We use $\mathrm{vec}(\boldsymbol{A})$ to denote the vectorization of the matrix $\boldsymbol{A}$ in column-major order. The meaning of other notations can be inferred from the context.

## 2 WELL-POSEDNESS OF MIGNN: A MONOTONE OPERATOR PERSPECTIVE

In this section, we characterize the well-posedness of MIGNN leveraging monotone operator theory, see Appendix B for a brief review of monotone operator theory. Using the Kronecker product[3] and vectorization of a matrix, we can rewrite Equation (1) into the following equivalent vectorized form

$$\mathrm{vec}(\boldsymbol{Z}^{(k+1)}) = \sigma\big(\boldsymbol{G}^\top \otimes \boldsymbol{W} \mathrm{vec}(\boldsymbol{Z}^{(k)}) + \mathrm{vec}(g_{\boldsymbol{B}}(\boldsymbol{X}))\big). \tag{2}$$

Gu et al. propose the well-posedness condition of IGNN as $\lambda_1(|\boldsymbol{G}^\top \otimes \boldsymbol{W}|) < 1$, guaranteeing that the unique fixed point of Equation (2) can be found by Picard iteration. Selecting $\boldsymbol{G} = \hat{\boldsymbol{A}}$, all eigenvalues of $\boldsymbol{G}$ are in $[-1, 1]$ with $\lambda_1(\boldsymbol{G}) = 1$. Therefore, well-posedness of IGNN is equivalent to $\lambda_1(|\boldsymbol{W}|) < 1$ as $\lambda_1(|\boldsymbol{G}^\top \otimes \boldsymbol{W}|) = \lambda_1(\boldsymbol{G})\lambda_1(|\boldsymbol{W}|) = \lambda_1(|\boldsymbol{W}|)$. Then, IGNN parameterizes $\boldsymbol{W}$ by relaxing the well-posedness condition $\lambda_1(|\boldsymbol{W}|) < 1$ to $\|\boldsymbol{W}\|_\infty < 1$, which constrains the magnitudes of eigenvalues of $\boldsymbol{W}$ to be less than 1.

We seek to apply the monotone operator theory to improve the expressivity and efficiency of existing IGNNs. According to the monotone operator theory [68; 77], finding the fixed point of Equation (2) is equivalent to solving the monotone inclusion problem: find $\boldsymbol{0} \in (\mathcal{F} + \mathcal{G})(\mathrm{vec}(\boldsymbol{Z}))$ with $\mathcal{F}$ and $\mathcal{G}$ being two set-valued functions that are given below

$$\mathcal{F}(\mathrm{vec}(\boldsymbol{Z})) = (\boldsymbol{I} - \boldsymbol{G}^\top \otimes \boldsymbol{W})\mathrm{vec}(\boldsymbol{Z}) - \mathrm{vec}(g_{\boldsymbol{B}}(\boldsymbol{X})) \text{ and } \mathcal{G} = \partial f, \tag{3}$$

where $\partial f$ denotes the subgradient of a convex closed proper function $f$ that satisfies $\sigma = \mathrm{prox}_f^1$ with $\mathrm{prox}_f^\alpha(x) \equiv \arg\min_z \big\{\frac{1}{2}\|x - z\|^2 + \alpha f(z)\big\}$. When $\sigma$ is ReLU, then $\sigma = \mathrm{prox}_f^\alpha$ for $\forall \alpha > 0$ with $f$ being the indicator of the positive octant, i.e. $f(x) = I\{x \geq 0\}$. The above monotone inclusion problem admits a unique solution if the operator $\mathcal{F}$ is strongly monotone, i.e. $\boldsymbol{I} - \boldsymbol{G}^\top \otimes \boldsymbol{W} \succeq m\boldsymbol{I}$ or,

$$\frac{1}{2}(\boldsymbol{G}^\top \otimes \boldsymbol{W} + \boldsymbol{G} \otimes \boldsymbol{W}^\top) \preceq (1 - m)\boldsymbol{I}.$$

Therefore, we obtain the following well-posedness condition for MIGNN:

---

[3]See Appendix D for a review of some properties about the Kronecker product.

**Proposition 1** (Well-posedness condition for MIGNN). *Let the non-linearity $\sigma$ be ReLU and $\boldsymbol{K} = \frac{1}{2}(\boldsymbol{G}^\top \otimes \boldsymbol{W} + \boldsymbol{G} \otimes \boldsymbol{W}^\top)$. Then the MIGNN model Equation (2) is well-posed as long as $\boldsymbol{K} \preceq (1-m)\boldsymbol{I}$ for some $m > 0$. As $\boldsymbol{K}$ is symmetric, $\boldsymbol{K} \preceq (1-m)\boldsymbol{I}$ is equivalent to requiring that each eigenvalue of $\boldsymbol{K}$ is no more than $1 - m$.*

We provide the proof of Proposition 1 in the appendix; similarly, the proof of all the subsequent theoretical results are provided in the appendix. The well-posedness condition in Proposition 1 allows for more flexible parametrizations than [39] by enabling the real part of eigenvalues of $\boldsymbol{W}$ to be in the range $(-\infty, 1)$ and the imaginary part to be arbitrary. Along with providing a more flexible well-posedness condition for MIGNN, monotone operator theory guides us in designing efficient algorithms for implementing MIGNN; see Sec. 4.

## 3 FLEXIBLE PARAMETERIZATION OF MIGNN

This section presents the monotone and orthogonal parameterizations of $\boldsymbol{W}$ for MIGNN in Equation (2). *The monotone parameterization can enhance IGNN's expressivity*, and *the orthogonal parameterization can stabilize and accelerate the training of MIGNNs.*

### 3.1 MONOTONE PARAMETERIZATION

Proposition 1 informs us to design a more expressive parameterization of $\boldsymbol{W}$ for MIGNN than that used for IGNN leveraging monotone operator theory.

**Proposition 2** (Monotone parameterization). *Let $G = (V, E)$ be a graph and let $\boldsymbol{G}$ be $\boldsymbol{L}/2$ with $\boldsymbol{L} := \boldsymbol{D}^{-1/2}(\boldsymbol{D} - \boldsymbol{A})\boldsymbol{D}^{-1/2}$ being the normalized Laplacian, where $\boldsymbol{A}$ is the adjacency matrix and $\boldsymbol{D}$ is the degree matrix with $D_{ii} = \sum_{j=1}^n A_{ij}$. Then the MIGNN model $\boldsymbol{Z}^{(k+1)} = \sigma(\boldsymbol{W}\boldsymbol{Z}^{(k)}\boldsymbol{G} + g_{\boldsymbol{B}}(\boldsymbol{X}))$ is well-posed when the weight matrix $\boldsymbol{W}$ is parameterized as follows*

$$\boldsymbol{W} = (1-m)\boldsymbol{I} - \boldsymbol{C}\boldsymbol{C}^\top + \boldsymbol{F} - \boldsymbol{F}^\top, \tag{4}$$

*where $\boldsymbol{C}, \boldsymbol{F} \in \mathbb{R}^{d \times d}$ are arbitrary matrices, and $m > 0$.*

**Remark 1.** *In monotone parameterization, we first set the graph-related matrix $\boldsymbol{G}$ to be $\boldsymbol{L}/2$, whose eigenvalues are in $[0, 1]$. In contrast, the range of the eigenvalues of $\hat{\boldsymbol{A}}$ used in IGNN, see Sec. 1, is $[-1, 1]$. Next, we parameterize $\boldsymbol{W}$ as in Equation (4), whose eigenvalues have real part in $(-\infty, 1-m]$. Thus, $\frac{1}{2}(\boldsymbol{G}^\top \otimes \boldsymbol{W} + \boldsymbol{G} \otimes \boldsymbol{W}^\top) \preceq (1-m)\boldsymbol{I}$, guaranteeing the well-posedness of MIGNN. Moreover, $\boldsymbol{W} = (1-m)\boldsymbol{I} - \boldsymbol{C}\boldsymbol{C}^\top + \boldsymbol{F} - \boldsymbol{F}^\top$ describes all possible $\boldsymbol{W}$ that satisfy $\boldsymbol{W} \preceq (1-m)\boldsymbol{I}$.*

### 3.2 ORTHOGONAL PARAMETERIZATION

As discussed in Sec. 1, IGNN learns LRD when $\lambda_1(|\boldsymbol{W}|)$ approaches 1 in magnitude. This is often not the case when starting from Gaussian random initialization — making IGNN unstable for learning LRD. Inspired by the unitary RNN [5], we propose to use the orthogonal parameterization [41; 54; 53] with a learnable scaling factor to stabilize MIGNN in learning LRD. In particular, we parameterize $\boldsymbol{W}$ by the following scaled Cayley map

$$\boldsymbol{W} = \phi(\gamma)(\boldsymbol{I} - \boldsymbol{S})(\boldsymbol{I} + \boldsymbol{S})^{-1}, \tag{5}$$

where $\phi(\cdot)$ is the sigmoid function and $\gamma \in \mathbb{R}$ is a learnable parameter ensuring $\phi(\gamma) \in (0, 1)$. $\boldsymbol{S} = \boldsymbol{C} - \boldsymbol{C}^\top$ is a skew-symmetric matrix with $\boldsymbol{C} \in \mathbb{R}^{d \times d}$ being an arbitrary parameterized matrix. It is evident that MIGNN with the parameterization in Equation (5) is well-posed with $\boldsymbol{G}$ being $\hat{\boldsymbol{A}}$ defined in Sec. 1. Also, all eigenvalues of $(\boldsymbol{I} - \boldsymbol{S})(\boldsymbol{I} + \boldsymbol{S})^{-1}$ have magnitude 1, see a derivation in Appendix E.3. To effectively learn LRD, MIGNN only requires the scalar $\phi(\gamma)$ to converge to 1.

## 4 ACCELERATED OPERATOR SPLITTING FOR IMPLEMENTING IGNNS

It is worth noting that monotone and orthogonal parameterizations are beyond the efficient convergence regime of the Picard iteration. Thus, we leverage the operator splitting schemes to find the fixed point of the equilibrium equation with monotone or orthogonal parameterization. Operator splitting schemes often converge faster than Picard iteration and can guarantee convergence of IGNNs even when Picard iteration fails [68]. In particular, for small graphs and tasks where learning LRD is not crucial, we use Anderson-accelerated forward-backward splitting (FB) to implement MIGNN with monotone parameterization. For tasks that require learning LRD, we employ

Anderson-accelerated Peaceman-Rachford splitting (PR)[4], with the Neumann series approximation accompanied by diffusion convolution, to implement MIGNN with orthogonal parameterization.

We structure this section as follows: In Sec. 4.1, we present FB (Sec. 4.1.1)/PR (Sec. 4.1.2) for finding the fixed point of MIGNNs using monotone/orthogonal parameterization. In Sec. 4.2, we present backward propagation algorithms for updating the parameters of MIGNN.

## 4.1 FORWARD PROPAGATION FOR FINDING THE FIXED POINT

### 4.1.1 FB SPLITTING

We can find the fixed point of MIGNN in Equation (2) via FB splitting with iterative scheme

$$\boldsymbol{Z}^{(k+1)} := F_\alpha^{\mathrm{FB}}(\boldsymbol{Z}^{(k)}) := \mathrm{prox}_f^\alpha \left( \boldsymbol{Z}^{(k)} - \alpha \cdot \left( \boldsymbol{Z}^{(k)} - \boldsymbol{W}\boldsymbol{Z}^{(k)}\boldsymbol{G} - g_{\boldsymbol{B}}(\boldsymbol{X}) \right) \right), \ \alpha > 0 \text{ is a constant. (6)}$$

We provide a detailed implementation of FB splitting in Appendix F.1. Note that the Lipschitz constant of the FB iteration is $L^{\mathrm{FB}} := \sqrt{1 - 2\alpha m + \alpha^2 \|\boldsymbol{I} - \boldsymbol{G}^\top \otimes \boldsymbol{W}\|^2}$ [68, Section 5]. Therefore, FB splitting converges to the fixed point if $\alpha < 2m/\|\boldsymbol{I} - \boldsymbol{G}^\top \otimes \boldsymbol{W}\|^2$. By choosing a proper $\alpha$, FB splitting can converge in the regime that Picard iteration does not. However, when the monotone parameterization is used $\|\boldsymbol{W}\|$ can be arbitrarily large. Thus $\alpha$ needs to be small to guarantee the convergence of FB splitting, in which case the Lipschitz constant is close to 1, and the convergence of FB splitting will be significantly slowed. FB splitting is appealing for learning with small graphs and tasks where learning LRD is not crucial. In this case, we use monotone parameterization to improve the expressivity of the model, and we denote the MIGNN with monotone parameterization using FB splitting as `MIGNN-Mon`. For large graphs and tasks that require learning LRD, FB splitting suffers from slow convergence. Next, we will present PR splitting, which is better for learning large-scale graphs and LRD. Furthermore, we argue that PR splitting is not suitable for implementing MIGNN with monotone parameterization.

### 4.1.2 PR SPLITTING

PR splitting used in [77] is guaranteed to converge for a much broader choice of $\alpha$ and requires fewer iterations than FB splitting. However, each iteration of PR splitting requires inverting large matrices, which is computationally much more expensive and less scalable than FB splitting. PR splitting finds the solution $\boldsymbol{Z}^*$ of the MIGNN by letting $\boldsymbol{Z}^* = \mathrm{prox}_f^\alpha(\boldsymbol{U}^*)$ where $\boldsymbol{U}^* \in \mathbb{R}^{d \times n}$ is obtained from the fixed-point iteration $\mathrm{vec}(\boldsymbol{U}^{(k+1)}) = F_\alpha^{\mathrm{PR}}(\mathrm{vec}(\boldsymbol{U}^{(k)})) := \mathcal{C}_\mathcal{F}\mathcal{C}_\mathcal{G}(\mathrm{vec}(\boldsymbol{U}^{(k)}))$ with $\mathcal{C}_\mathcal{F}$ and $\mathcal{C}_\mathcal{G}$ being the Cayley operators (see Appendix B for details) of $\mathcal{F}$ and $\mathcal{G}$, respectively. Let $\boldsymbol{u}^{(k)}$ be the shorthand notation of $\mathrm{vec}(\boldsymbol{U}^{(k)})$. Then we can formulate the PR splitting as follows

$$\boldsymbol{u}^{(k+1)} := F_\alpha^{\mathrm{PR}}(\boldsymbol{u}^{(k)}) = 2\boldsymbol{V}\left(2\,\mathrm{prox}_f^\alpha(\boldsymbol{u}^{(k)}) - \boldsymbol{u}^{(k)} + \alpha\,\mathrm{vec}(g_{\boldsymbol{B}}(\boldsymbol{X}))\right) - 2\,\mathrm{prox}_f^\alpha(\boldsymbol{u}^{(k)}) + \boldsymbol{u}^{(k)}, \quad (7)$$

where the matrix $\boldsymbol{V} := (\boldsymbol{I} + \alpha(\boldsymbol{I} - \boldsymbol{G}^\top \otimes \boldsymbol{W}))^{-1}$ and $\boldsymbol{u}^{(0)}$ is the zero vector. With the parametrizations discussed in Sec. 3, the linear operator $\mathcal{F}$ in Equation (3) is strongly monotone and $L$-Lipschitz where $L = \|\boldsymbol{I} - \boldsymbol{G}^\top \otimes \boldsymbol{W}\|$. Therefore, its Cayley operator $\mathcal{C}_\mathcal{F}$ and hence $F_\alpha^{\mathrm{PR}}$ is contractive with the optimal choice of $\alpha$ being $1/L$, see [68, Section 6]. In particular, it is suggested to choose $\alpha = 1/(1 + \phi(\gamma))$ when using orthogonal parametrization $\boldsymbol{W} = \phi(\gamma)(\boldsymbol{I} - \boldsymbol{S})(1 + \boldsymbol{S})^{-1}$. The pseudocode for the detailed implementation of PR splitting in Equation (7) can be found in Appendix F.1.

**Remark 2.** *Douglas-Rachford (DR) splitting is another option for solving MIGNN, which is often faster than PR. However, in our case PR is contractive, making it faster than DR for the same $\alpha$.*

PR splitting also benefits MIGNNs in learning LRD when an orthogonal parameterization is used. To see this, we have the following Neumann series expansion of $\boldsymbol{V}(\boldsymbol{u}^{(k)})$

$$\boldsymbol{V}(\boldsymbol{u}^{(k)}) = (\boldsymbol{I} + \alpha(\boldsymbol{I} - \boldsymbol{G}^\top \otimes \boldsymbol{W}))^{-1}(\boldsymbol{u}^{(k)}) = \frac{1}{1+\alpha}\left(\boldsymbol{I} - \frac{\boldsymbol{G}^\top \otimes \boldsymbol{W}}{1+1/\alpha}\right)^{-1}(\boldsymbol{u}^{(k)}) = \frac{1}{1+\alpha}\sum_{i=0}^\infty \frac{\mathrm{vec}\left(\boldsymbol{W}^i\boldsymbol{U}^{(k)}\boldsymbol{G}^i\right)}{(1+1/\alpha)^i} \quad (8)$$

where the last equality follows from $(\boldsymbol{A} \otimes \boldsymbol{B})^k = \boldsymbol{A}^k \otimes \boldsymbol{B}^k$, and $(\boldsymbol{A} \otimes \boldsymbol{B})\mathrm{vec}(\boldsymbol{C}) = \mathrm{vec}(\boldsymbol{B}\boldsymbol{C}\boldsymbol{A}^\top)$ for $\forall \boldsymbol{A}, \boldsymbol{B}$ and $\boldsymbol{C}$ that satisfy dimensional consistency. Equation (8) indicates that each node can access information from its $\infty$-hop neighbors in a single PR iteration for MIGNN with orthogonal parameterization. This cannot be said of monotone parameterization with large $\|\boldsymbol{W}\|$, as

---

[4]For the sake of presentation, we denote Anderson-accelerated FB and PR splitting as FB and PR.

the Neumann series expansion in the last equality of Equation (8) no longer applies. Evaluating $\frac{1}{1+\alpha}\left(\boldsymbol{I} - \frac{\boldsymbol{G}^\top \otimes \boldsymbol{W}}{1+1/\alpha}\right)^{-1}(\boldsymbol{u}^{(k)})$ can be carried out by using Bartels–Stewart algorithm [11], which converts computing $\boldsymbol{V}$ into diagonalizing the matrix $\boldsymbol{G}^\top$ and $\boldsymbol{W}$, respectively. From Equation (8), we have

$$\boldsymbol{V}(\text{vec}(\boldsymbol{U}^{(k)})) = \frac{1}{1+\alpha}\text{vec}\Big(\boldsymbol{Q_W}\Big[\boldsymbol{H} \odot \Big(\boldsymbol{Q_W^{-1}}\boldsymbol{U}^{(k)}\boldsymbol{Q_{G^\top}}\Big)\Big]\boldsymbol{Q_{G^\top}^\top}\Big) \tag{9}$$

where $\boldsymbol{Q_{G^\top}}\boldsymbol{\Lambda_{G^\top}}\boldsymbol{Q_{G^\top}^\top}$ and $\boldsymbol{Q_W}\boldsymbol{\Lambda_W}\boldsymbol{Q_W^{-1}}$ are the eigen-decomposition of $\boldsymbol{G}^\top$ and of $\boldsymbol{W}$, respectively, and $\boldsymbol{H} \in \mathbb{R}^{d \times n}$ whose $(i,j)$-th entry is $H_{ij} = 1/(1 - \frac{1}{1+1/\alpha}(\boldsymbol{\Lambda_W})_{ii}(\boldsymbol{\Lambda_{G^\top}})_{jj})$. We provide a proof of Equation (9) in Appendix E.4. According to Equation (9), one only needs to calculate the eigen-decomposition of $\boldsymbol{G}$ once prior to training and the eigen-decomposition of $\boldsymbol{W}$ once per epoch. The above matrix inversion procedure echos the idea of EIGNN [58]. MIGNN has multiple layers, with each fixed point iteration representing one layer. In contrast, EIGNN is reducible to a one-layer model; see Appendix A.2 for details on EIGNN.

Although PR splitting can capture LRD in a single iteration, computing $\boldsymbol{V}$ in Equation (7) requires computationally prohibitive matrix inversion. We provide two remedies to address this issue for MIGNN using orthogonal parameterization: 1) We use Neumann series expansion to approximate the matrix inversion when orthogonal parameterization is used. 2) We replace the graph-related matrix $\boldsymbol{G}$ with a generalized graph diffusion convolution matrix, e.g. heat kernel or the personalized PageRank [34; 33]. Notice that the above two remedies do not work for MIGNN using monotone parameterization since we can no longer use the Neumann series approximation. Therefore, MIGNN with monotone parameterization using PR splitting is not scalable to learning large graphs.

**Neumann series approximation.** In the orthogonal parameterization of $\boldsymbol{W}$ we have $\|\frac{\boldsymbol{G}^\top \otimes \boldsymbol{W}}{1+1/\alpha}\| < 1$, ensuring efficient approximation of $\boldsymbol{V}$ in Equation (7) using only a few terms of its Neumann series expansion. The $K$-th order Neumann series expansion of $\boldsymbol{V}(\text{vec}(\boldsymbol{U}^{(k)}))$ is given by

$$\boldsymbol{N}_K(\text{vec}(\boldsymbol{U}^{(k)})) := \frac{1}{1+\alpha}\sum_{i=0}^{K}\frac{\text{vec}\Big(\boldsymbol{W}^i\boldsymbol{U}^{(k)}\boldsymbol{G}^i\Big)}{(1+1/\alpha)^i}. \tag{10}$$

According to Equation (7), the $K$-th order Neumann series approximated PR iteration function, denoted as $\tilde{F}_\alpha^{\text{PR,K}}$, can be written as follows

$$\boldsymbol{u}^{(k+1)} := \tilde{F}_\alpha^{\text{PR,K}}(\boldsymbol{u}^{(k)}) = 2\boldsymbol{N}_K\Big(2\,\text{prox}_f^\alpha(\boldsymbol{u}^{(k)}) - \boldsymbol{u}^{(k)} + \alpha\,\text{vec}(g_{\boldsymbol{B}}(\boldsymbol{X}))\Big) - 2\,\text{prox}_f^\alpha(\boldsymbol{u}^{(k)}) + \boldsymbol{u}^{(k)}. \tag{11}$$

Each node can access information from its $K$-hop neighbors using the $K$-th order Neumann series approximated PR iteration, which is more efficient than the existing IGNN. Also, such a treatment can significantly accelerate forward propagation. We can intuitively understand this as follows: Each iteration of MIGNN, with $K$-th order Neumann series approximated PR iteration, aggregates information from $K$-hop neighbors, enabling the use of much fewer iterations than that of IGNN, which aggregates one hop per iteration. MIGNN can use a much smaller $\lambda_1(|\boldsymbol{W}|)$ than IGNN to reach the same number of hops, meaning MIGNN converges much faster than IGNN.

**MIGNN with diffusion convolution.** We can also improve MIGNNs for learning LRD using graph diffusion convolution [34; 1], i.e. instead of using $\hat{\boldsymbol{A}}$ or $\boldsymbol{L}$ defined in the previous context, we can set $\boldsymbol{G}$ to be the combination of higher powers of $\hat{\boldsymbol{A}}$ or $\boldsymbol{L}$, so that each node aggregates features from multi-hop neighbors at each iteration. In particular, we let $\boldsymbol{G} = \tilde{\boldsymbol{D}}^{-1/2}(\boldsymbol{A} + \cdots + \boldsymbol{A}^P)\tilde{\boldsymbol{D}}^{-1/2}$ for any positive integer $P$, where $\tilde{\boldsymbol{D}}$ is the degree matrix with $\tilde{D}_{ii} = \sum_{j=1}^{n}\sum_{k=1}^{P}(\boldsymbol{A}^k)_{ij}$; other choices of $\boldsymbol{G}$ can be found in [34]. We can show that the eigenvalues of $\tilde{\boldsymbol{D}}^{-1/2}(\boldsymbol{A} + \cdots + \boldsymbol{A}^P)\tilde{\boldsymbol{D}}^{-1/2}$ are all within $[-1,1]$; see E.4 for a proof. As such, the orthogonal parameterization of $\boldsymbol{W}$ still ensures the well-posedness of MIGNN. We write the MIGNN with $P$-th order diffusion matrix $\boldsymbol{G}$ as follows

$$\boldsymbol{Z} = \sigma(\boldsymbol{W}\boldsymbol{Z}\tilde{\boldsymbol{D}}^{-1/2}(\boldsymbol{A} + \boldsymbol{A}^2 + \cdots + \boldsymbol{A}^P)\tilde{\boldsymbol{D}}^{-1/2} + g_{\boldsymbol{B}}(\boldsymbol{X})). \tag{12}$$

We can further apply the operator splitting schemes to Equation (12). In particular, we denote the model as `MIGNN-N`$K$`D`$P$ when $\boldsymbol{W}$ is orthogonal, and Equation (12) is implemented using $P$-th order diffusion and $K$-th order Neumann series approximated PR iteration.

Now we discussion the time complexity of MIGNN-N$K$D$P$. The $P$-th order diffusion matrix only needs to be pre-computed once in preprocessing with time complexity $\mathcal{O}(nP|E_P|)$ where $n$ is the number of nodes, and $|E_P|$ denotes the number of non-zero entries in $\boldsymbol{A}^P$. In each epoch, the

parameter $K$ in the $K$-th order Neumann series affects the training time complexity linearly as $\mathcal{O}(KMd|E_P|)$ where $M$ denotes the maximal number of iterations, and $d$ is the feature dimension which is much smaller than the number of nodes.

### 4.1.3 ANDERSON ACCELERATION

We have already seen that the main steps in both FB and PR splitting schemes involve solving iterative equations, e.g. Equations (6) and (7), and we can utilize Anderson acceleration [4] to accelerate the convergence of these iterative equations. We provide the detailed formulation and pseudocode for Anderson-accelerated operator splitting-based MIGNNs in Appendix F.3.

### 4.2 BACKWARD PROPAGATION FOR UPDATING MIGNNS

We derive backpropagation for MIGNN based on implicit differentiation [35; 7; 26]. Recall that the vectorized MIGNN $\mathrm{vec}(\boldsymbol{Z}) = \sigma\left(\boldsymbol{G}^\top \otimes \boldsymbol{W}\,\mathrm{vec}(\boldsymbol{Z}) + \mathrm{vec}(g_{\boldsymbol{B}}(\boldsymbol{X}))\right)$, has equilibrium point $\mathrm{vec}(\boldsymbol{Z}^*)$. For any loss function $\ell$ and any parameter $\theta$ ($\boldsymbol{W}$ or $\boldsymbol{B}$), we have

$$\frac{\partial \ell}{\partial \theta} = \frac{\partial \ell}{\partial \mathrm{vec}(\boldsymbol{Z}^*)}\left(\boldsymbol{I} - \boldsymbol{J}\left(\boldsymbol{G}^\top \otimes \boldsymbol{W}\right)\right)^{-1} \frac{\partial \sigma\left(\boldsymbol{G}^\top \otimes \boldsymbol{W}\,\mathrm{vec}(\boldsymbol{Z}^*) + \mathrm{vec}(g_{\boldsymbol{B}}(\boldsymbol{X}))\right)}{\partial \theta} \tag{13}$$

where $\boldsymbol{J}$ is the Jacobian of $\sigma$ evaluated at $\boldsymbol{G}^\top \otimes \boldsymbol{W}\,\mathrm{vec}(\boldsymbol{Z}^*) + \mathrm{vec}(g_{\boldsymbol{B}}(\boldsymbol{X}))$. The values of the first and last term in Equation (13) can be found through automatic differentiation by running one more iteration in the forward pass. Note that the product of the first two terms remains the same for any $\theta$. Hence one only needs to compute it once in each backward pass. However, it can still be expensive to find $(\partial \ell)/(\partial \mathrm{vec}(\boldsymbol{Z}^*))(\boldsymbol{I} - \boldsymbol{J}\left(\boldsymbol{G}^\top \otimes \boldsymbol{W}\right))^{-1}$. Following [77, Theorem 2], the operator splitting methods can be used in the backward pass so that computing $(\boldsymbol{I} - \boldsymbol{J}(\boldsymbol{G}^\top \otimes \boldsymbol{W}))^{-1}$ can be converted into computing $\boldsymbol{V} = (\boldsymbol{I} - (\boldsymbol{G}^\top \otimes \boldsymbol{W}))^{-1}$, which is already calculated in the forward pass; see Appendix F.2. Similar to the forward propagation, the backpropagation can also benefit from Anderson acceleration using an iterative formulation, and we provide more details in Appendix F.2.

## 5 EXPERIMENTAL RESULTS

In this section, we compare the performance of MIGNN-Mon (*MIGNN with monotone parameterization implemented via FB splitting*) and MIGNN-N$K$D$P$ (*MIGNN with orthogonal parameterization implemented via PR splitting accompanied by $K$-th order Neumann series approximation and $P$-th order graph diffusion convolution*) with IGNN and several other popular GNNs on various graph classification tasks at both node and graph levels. We aim to show that 1) MIGNN-Mon is significantly more expressive than IGNN for both node and graph classifications, and 2) MIGNN-N$K$D$P$ can learn LRD effectively, efficiently, and stably. The hyperparameters used in each model are provided in Appendix K. We conduct all experiments using NVIDIA RTX 3090 graphics cards.

### 5.1 DIRECTED CHAIN CLASSIFICATION

To show that MIGNNs can capture LRD in the underlying graphs, we test them on the synthetic chain task using the experimental setup from [58]. The chain task dataset comprises of $c$ classes and $n_c$ single-linked directed chains, each containing $l$ nodes. For each chain, only the feature on the first node encodes the label information.

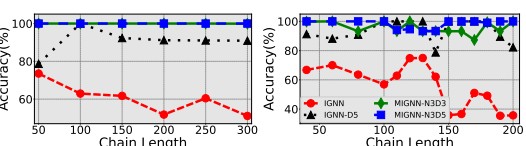

Figure 3: The accuracy of IGNN and MIGNN of different configurations for classifying directed chains of different lengths. Left: binary classification ($c = 2$). Right: three-class classification ($c = 3$).

The data is partitioned into training, validation, and test sets of 5%, 10%, and 85%, respectively. We consider binary ($c = 2$) and three-class classification ($c = 3$) problems over several different chain lengths. For IGNN, we use the experimental settings used in [71]. For MIGNN, we consider MIGNN-N$K$D$P$ for this task. Fig. 3 shows the averaged test accuracy over 5 random seeds of different models for classifying directed chains of length ranging from 50 to 300 in an increment of 50 for the binary case and from 40 to 200 in an increment of 20 for the three-class case. For binary classification, MIGNN-N3D3 and MIGNN-N3D5 both score perfectly for all random initializations of the considered chain lengths. For the three-class task, both MIGNN models achieve high accuracy consistently with the higher order diffusion models, and the higher order diffusion model outperforms the lower order diffusion model on longer chains. In contrast, the accuracy of IGNN is *much lower and less stable* than that of MIGNNs, and in general, IGNN's performance becomes worse as the chain length increases. We provide an ablation study of the impact of the order of

Neumann series approximation and graph diffusion convolution on the chain classification accuracy and computational time in Appendix G and H, respectively.

We can also set $G$ to be the diffusion matrix in Equation (12) to enhance IGNN's capability in learning LRD. E.g. we can equip IGNN with a diffusion matrix of order 5, and we denote the resulting model as IGNN-D5. Fig. 3 further contrasts the performance of diffusion enhanced models, and we observe that MIGNN is more consistent and more accurate as the chain length increases.

Based on the operator splitting theory, we expect that MIGNNs are more computationally efficient than IGNNs when both models can accurately classify the chain nodes. Fig. 4 compares the accuracy and computational efficiency

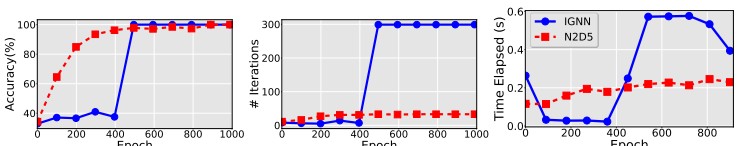

Figure 4: The accuracy and efficiency of MIGNN-N2D5 over IGNN for three class chains, of length 140, classification.

of MIGNN-N2D5 over IGNN for three-class chain classification. We see that MIGNN-N2D5 stably approaches perfect accuracy compared to IGNN, which abruptly changes around epoch 400. When both models accurately classify the chains, MIGNN-N2D5 requires fewer iterations and less computational time than IGNN.

## 5.2 GRAPH NODE CLASSIFICATION

In this subsection, we contrast MIGNN-Mon and MIGNN-N1D1 with some existing GNNs for several benchmark graph node classification tasks, including Cora, Citeseer, and Pubmed; each dataset's statistics of nodes/edge/average shortest path length between nodes are 2485/5069/5.27, 2120/3679/9.31, and 19717/44324/6.34, respectively. We use the

| Datasets | Cora | Citeseer | Pubmed |
|---|---|---|---|
| Geom-GCN [65] | 85.27 | **77.99** | **90.05** |
| GCNII [21] | **88.49** | 77.08 | 89.57 |
| APPNP [32] | 85.09 | 75.73 | 79.73 |
| GCN+GDC [34] | 83.58 | 73.35 | 78.72 |
| GIND [22] | 88.25 | 76.81 | 89.22 |
| IGNN [39] | 85.80 | 75.24 | 87.66 |
| EIGNN [80] | 85.89 | 75.31 | 87.92 |
| MIGNN-Mon | 86.82 | 76.59 | 88.00 |
| MIGNN-N5D1 | 87.04 | 74.91 | 83.55 |

Table 1: Node classification mean accuracy (%) for 10-fold cross-validation.

training procedure outlined in [22] and report the mean accuracy of 10-fold cross validation in Table 1. The MIGNN-Mon outperforms the implicit model benchmarks IGNN and EIGNN on all three tasks. We provide an ablation study of the impact of the order of Neumann series and graph diffusion convolution for graph node classification in Appendix G and H, respectively.

## 5.3 GRAPH CLASSIFICATION

In this subsection, we verify that MIGNN-Mon can be more expressive than IGNN for graph classification since the eigenvalues of monotone parameterization are more flexible than IGNN. We consider five bioinformatics-related graph classification benchmarks: MUTAG, PTC, COX2, PROTEINS, and NCI1 [81], and some details of these datasets are provided in Appendix J. The training is performed using 10-fold cross-validation using the experimental

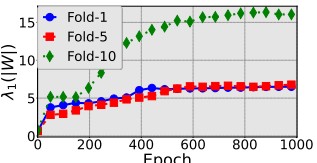

Figure 5: $\lambda_1(|W|)$ of MIGNN-Mon vs. Epoch on MUTAG.

setup of [71]. The averaged test accuracy and standard deviation across the 10 folds are shown in Table 2. For both IGNN and MIGNN-Mon, we use the hyperparameters outlined in [71]. We present the results for both IGNN and MIGNN-Mon in Table 2. Clearly, MIGNN-Mon outperforms IGNN on all tasks. To verify our theory, we report on the evolution of $\lambda_1(|W|)$ for three of the ten folds of MUTAG in Fig. 5. For all of the folds $\lambda_1(|W|)$ exceeds one. Table 2 also reports the accuracy of MIGNN-N3D1 against several baseline models where it performs better than IGNN and GIND on all tasks and achieves the best accuracy on COX2 and PROTEINS tasks among all studied models. These results show that learning LRD effectively is vital for classifying these graphs. We provide an ablation study of the impact of the order of Neumann series and graph diffusion convolution on classification accuracy and computational time in Appendix G and H, respectively.

## 5.4 LARGER SCALE GRAPH NODE CLASSIFICATION

We further show the advantages of MIGNN-N$K$D$P$ over IGNN and other GNNs for a larger scale graph node classification task — Amazon co-purchasing dataset, which contains 334863 nodes, 925872 edges, and the diameter of the graph is 44 [82]. We provide more details of the Amazon co-purchasing dataset in Appendix J. As in [23], we train on portions of the graph ranging from 5% to 9%, and test on sets representing 10% of the total graph. We then report both Macro-F1 and Micro-F1 consistent with [71]. Fig. 6 contrasts the computational cost of MIGNN-N1D1 with

| Datasets | MUTAG | PTC | COX2 | PROTEINS | NCI1 |
|---|---|---|---|---|---|
| # graphs/Avg # nodes | 188/17.9 | 344/25.5 | 467/41.2 | 1113/39.1 | 4110/29.8 |
| WL [70] | $84.1 \pm 1.9$ | $58.0 \pm 2.5$ | $83.2 \pm 0.2$ | $74.7 \pm 0.5$ | $\mathbf{84.5 \pm 0.5}$ |
| DCNN [6] | 67.0 | 56.6 | — | 61.3 | 62.6 |
| DGCNN [83] | 85.8 | 58.6 | — | 75.5 | 74.4 |
| GIN [80] | $89.4 \pm 5.6$ | $64.6 \pm 7.0$ | — | $76.2 \pm 3.4$ | $82.7 \pm 1.7$ |
| FDGNN [31] | $88.5 \pm 3.8$ | $63.4 \pm 5.4$ | $83.3 \pm 2.9$ | $76.8 \pm 2.9$ | $77.8 \pm 1.6$ |
| IGNN [39] | $76.0 \pm 13.4$ | $60.5 \pm 6.4$ | $79.7 \pm 3.4$ | $76.5 \pm 3.4$ | $73.5 \pm 1.9$ |
| GIND [22] | $89.3 \pm 7.4$ | $66.9 \pm 6.6$ | $84.8 \pm 4.2$ | $77.2 \pm 2.9$ | $78.8 \pm 2.9$ |
| GSN [16] | $92.2 \pm 7.5$ | $68.2 \pm 7.2$ | — | $76.6 \pm 5.0$ | $83.5 \pm 2.0$ |
| SIN [15] | — | — | — | $76.5 \pm 3.3$ | $82.8 \pm 2.2$ |
| CIN [14] | $\mathbf{92.7 \pm 6.1}$ | $68.2 \pm 5.6$ | — | $77.0 \pm 4.3$ | $83.6 \pm 1.4$ |
| MIGNN-Mon | $81.8 \pm 9.1$ | $\mathbf{72.6 \pm 6.7}$ | $85.0 \pm 5.3$ | $77.9 \pm 3.4$ | $73.6 \pm 2.0$ |
| MIGNN-N1D1 | $86.1 \pm 9.1$ | $70.9 \pm 6.5$ | $86.5 \pm 2.8$ | $79.0 \pm 3.3$ | $78.4 \pm 1.2$ |
| MIGNN-N3D1 | $91.4 \pm 7.5$ | $71.2 \pm 3.2$ | $\mathbf{88.2 \pm 4.1}$ | $\mathbf{80.1 \pm 3.8}$ | $80.8 \pm 1.81$ |

Table 2: Graph classification mean accuracy (%) $\pm$ standard deviation for 10-fold cross-validation. We take the results of the baseline models from [22] which are consistent with our reproduced results.

IGNN using 5% of the graph for training. $\lambda_1(|\boldsymbol{W}|)$ of MIGNN-N1D1 is much smaller than that of IGNN, implying faster convergence of MIGNN-N1D1 than IGNN as confirmed by the fact that MIGNN-N1D1 saves significantly on the number of iterations and computational time over IGNN.

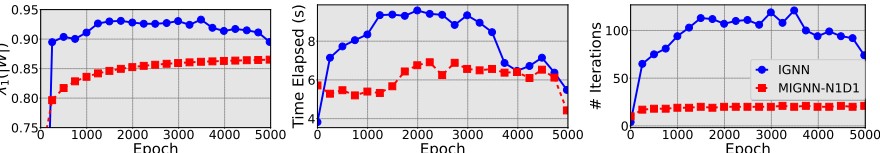

Figure 6: Epoch vs. $\lambda_1(|\boldsymbol{W}|)$, the time required for each epoch, and iterations required for each epoch of IGNN and MIGNN-N1D1 for the Amazon dataset with 5% training portion.

Fig. 7 contrasts MIGNN-N1D1 with baseline models when trained on portions of the graph ranging from 5% to 9%. We see that MIGNN-N1D1 outperforms almost all baseline models over all different portions of the graph for the training. Though MIGNN-N1D1 does not outperform IGNN significantly, MIGNN-N1D1 enjoys significant computational advantages over IGNN.

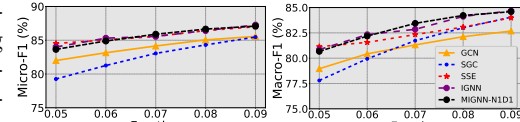

Figure 7: Fraction vs. Micro-F1 (left) and Macro-F1 (right) training accuracy on the Amazon dataset.

## 5.5 Physical diffusion in networks

We further consider a physical problem of fluid flow in porous media, following [63]. The model is a 3D graph whose nodes and edges correspond to pore chambers and throats. We sample training graphs of different sizes between 100 and 500, which are generated to fit into 0.1 m$^3$ cubes. We aim to predict the equilibrium pressures $\boldsymbol{Z}^*$ inside pore networks $G$. We train MIGNN to minimize the mean-squared error (MSE) between the prediction and $\boldsymbol{Z}^*$. We utilize the experimental setup of [63] and include their reported results for IGNN. Both IGNN and MIGNN use the same encoder and decoder architecture. Graphs of $50-200$ nodes are sampled in training and 1000 test graphs are generated for pore counts from 200 to 500. Fig. 8 shows the MSE for the test graphs as the number of nodes (pores)

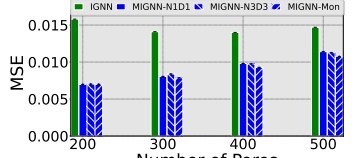

Figure 8: The average MSE of 500 sampled test iterations vs. the number of pores. The error bars represent the standard error of the prediction. MIGNN with different parameterizations outperforms IGNN by a significant amount.

varies from 200 to 500. MIGNN with both monotone and orthogonal parameterizations outperform IGNN by a significant margin. For this task of learning physical diffusion in networks, CGS [63] performs better than MIGNN and IGNN in accuracy. As a future direction, we plan to integrate the idea of the learnable graph-related matrix $\boldsymbol{G}$ that is used in CGS with our proposed MIGNN to further improve the performance of MIGNN for learning physical diffusion in networks.

## 6 Concluding Remarks

We propose MIGNN based on a monotone operator viewpoint of IGNN. In particular, MIGNN can be parameterized more flexibly than the benchmark IGNN. We provide efficient implementations of MIGNN that integrates diffusion convolution leveraging different operator splitting schemes with Anderson acceleration. Numerically, MIGNN remarkably outperforms the existing IGNN in accuracy, stability, computational efficiency, and learning LRD. As IGNNs are closely related to RNNs, an interesting future direction is to explore if the ideas from other RNN architectures [42; 59] can be adapted to the improvement of IGNNs.

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

# A  A BRIEF REVIEW OF IGNN AND RELATED MODELS

## A.1  IGNN: FORWARD AND BACKWARD PROPAGATION

IGNN employs a projected gradient descent method in the training phase to ensure their proposed well-posedness condition is satisfied. In forward propagation, IGNN finds the equilibrium through direct Picard iteration. During backward propagation, IGNN uses the implicit function theorem at the equilibrium to compute the gradient. The computationally expensive terms related to $\frac{\partial \ell}{\partial \mathrm{vec}(\boldsymbol{Z}^*)} \left( \boldsymbol{I} - \boldsymbol{J} \left( \boldsymbol{G}^\top \otimes \boldsymbol{W} \right) \right)^{-1}$ (see Section 4.2 for notations) is also computed implicitly through Picard iteration.

## A.2  EIGNN, CGS, AND GIND

**EIGNN**  Efficient infinite-depth graph neural networks (EIGNN) is an implicit graph neural network model proposed by Liu et al. [58] whose counterpart in explicit GNN is simple graph convolution (SGC) [79]. The main update step in EIGNN is given by

$$\boldsymbol{Z}^{(k+1)} = \gamma g(\boldsymbol{F})\boldsymbol{Z}^{(k)}\boldsymbol{G} + \boldsymbol{X} \tag{14}$$

where $\boldsymbol{Z}^{(\cdot)}$ denotes the hidden feature, $\boldsymbol{G}$ is the normalized augmented adjacency matrix $\hat{\boldsymbol{A}}$ (See Section 1), $\boldsymbol{X}$ is the input feature, $g(\boldsymbol{F})$ is the weight matrix which is parameterized to guarantee convergence, and $\gamma$ is a constant scalar in $(0, 1)$. Note that, there is no non-linearity in the fixed-point Equation (14) and this allows EIGNN to find the equilibrium by the following closed formula:

$$\lim_{k \to \infty} \mathrm{vec}\left( \boldsymbol{Z}^{(k+1)} \right) = (\boldsymbol{I} - \gamma(\boldsymbol{G}^\top \otimes g(\boldsymbol{F})))^{-1} \mathrm{vec}(\boldsymbol{X}). \tag{15}$$

For computation efficiency consideration, the matrix inverse operation is reduced to eigenvalue decomposition of $\boldsymbol{G}^\top$ and $g(\boldsymbol{F})$ where the eigenvalue decomposition $\boldsymbol{G}^\top$ is pre-calculated before training.

**CGS**  Convergent graph solver (CGS) is an implicit graph neural network proposed by Park et al. in [63] where the fixed point equation in use can be described as follows

$$\boldsymbol{Z}^{(k+1)} = \gamma \boldsymbol{Z}^{(k)}\boldsymbol{G}_\theta + g_{\boldsymbol{B}}(\boldsymbol{X}) \tag{16}$$

where $\boldsymbol{Z}^{(\cdot)}$ is the hidden feature, $\gamma$ is the contraction factor, $\boldsymbol{G}_\theta \in \mathbb{R}^{n \times n}$ is the graph-related matrix that is learnable and $g_{\boldsymbol{B}}(\boldsymbol{X})$ is the input-dependent bias term. Similar to the EIGNN case, the linearity in Equation 16 allows the fixed point to be found by a closed formula.

**GIND**  The optimization-induced graph implicit nonlinear diffusion (GIND) is an implicit graph neural network proposed by Chen et al. [22]. GIND involves a fixed point iteration equation of the following form:

$$\boldsymbol{Z}^{(k+1)} = -\boldsymbol{W}^\top \sigma(\boldsymbol{W}(\boldsymbol{Z}^{(k)} + g_{\boldsymbol{B}}(\boldsymbol{X}))\boldsymbol{G})\boldsymbol{G}^\top, \tag{17}$$

where $\boldsymbol{Z}^{(\cdot)}$ is the hidden feature, $\boldsymbol{W}$ is the weight matrix, $g_{\boldsymbol{B}}(\boldsymbol{X})$ is some input-dependent bias term, and $\boldsymbol{G}$ is a normalization of the adjacency matrix $\boldsymbol{A}$. The precise definition of $\boldsymbol{G}$ is given as $\boldsymbol{G} := \hat{\boldsymbol{D}}^{-1/2}\boldsymbol{A}/\sqrt{2}$ where $\hat{\boldsymbol{D}}$ is the degree matrix of the augmented adjacency matrix $\boldsymbol{A} + \boldsymbol{I}$ given as $\hat{D}_{ii} := 1 + \sum_j A_{ij}$. The weight matrix $\boldsymbol{W}$ is parameterized so that $\|\boldsymbol{W}\|\|\boldsymbol{G}\| < 1$. Similar to IGNN, the Picard iteration is employed to find the fixed point. The authors claimed that the new fixed-point equation (Equation 17) represents a nonlinear diffusion process with anisotropic properties while IGNN only represents a linear isotropic diffusion. However, we observe that GIND is closely related to the following simple variant of IGNN where the main change is to

$$\boldsymbol{Z}^{(k+1)} = \sigma\left( \boldsymbol{W}(-\boldsymbol{W}^\top)\boldsymbol{Z}^{(k)}\boldsymbol{G}^\top\boldsymbol{G} + \boldsymbol{W}g_{\boldsymbol{B}}(\boldsymbol{X})\boldsymbol{G} \right) \tag{18}$$

where the notations are the same as in Equation 17. In fact, once $\|\boldsymbol{W}\|\|\boldsymbol{G}\| < 1$, and assuming $\sigma$ is a non-expansive activation function (for example, tanh, ReLU, ELU), then Equation 18 is contractive and hence its fixed point exists. Let $\boldsymbol{Z}^*$ be the fixed-point of Equation (18), then we claim that

$\tilde{Z} = -W^\top Z^* G^\top$ is the fixed point of Equation (17) with the same $W$, $G$, and $g_B(X)$ used in both Equation 18 and Equation 17. This can be seen from the following direct calculation:

$$\tilde{Z} = -W^\top Z^* G^\top$$
$$= -W^\top \sigma \left( W(-W^\top) Z^* G^\top G + W g_B(X) G \right) G^\top$$
$$= -W^\top \sigma \left( W \tilde{Z} G + W g_B(X) G \right) G^\top$$
$$= -W^\top \sigma \left( W(\tilde{Z} + g_B(X)) G \right) G^\top.$$

# B  A BRIEF REVIEW OF MONOTONE OPERATOR THEORY

## B.1  OPERATORS

In this section, we briefly review the definition and basic theory of monotone operators, more details can be found in [68]. We sat $\mathcal{T}$ is a *(set-valued) operator* if $\mathcal{T}$ maps a point in $\mathbb{R}^d$ to a subset of $\mathbb{R}^d$. and we denote this as $\mathcal{T} : \mathbb{R}^d \Rightarrow \mathbb{R}^d$. We define the graph of an operator as

$$\text{Gra } \mathcal{T} = \{(x, u) | u \in \mathcal{T}(x)\}.$$

Mathematically, an operator and its graph are equivalent. In other words, we can view $\mathcal{T} : \mathbb{R}^d \Rightarrow \mathbb{R}^d$ as a point-to-set mapping and as a subset of $\mathbb{R}^d \times \mathbb{R}^d$.

Many notions for functions can be extended to operators. For example, the domain and range of an operator $\mathcal{T}$ are defined as

$$\text{dom } \mathcal{T} = \{x \mid \mathcal{T}(x) \neq \emptyset\}, \quad \text{range } \mathcal{T} = \{y \mid y = \mathcal{T}(x), x \in \mathbb{R}^d\}.$$

If $\mathcal{T}$ and $\mathcal{S}$ are two operators, we define their composition as

$$\mathcal{T} \circ \mathcal{S}(x) = \mathcal{T}\mathcal{S}(x) = \mathcal{T}(\mathcal{S}(x)),$$

and their sum as

$$(\mathcal{T} + \mathcal{S})(x) = \mathcal{T}(x) + \mathcal{S}(x).$$

Alternately, we can define the operator composition and sum using their graphs,

$$\mathcal{T}\mathcal{S} = \big\{(x, z) \mid \exists\, y\ (x, y) \in \mathcal{S}, (y, z) \in \mathcal{T}\big\},$$

$$\mathcal{T} + \mathcal{S} = \big\{(x, y + z) \mid (x, y) \in \mathcal{T}, (x, z) \in \mathcal{S}\big\}.$$

The identity ($\mathcal{I}$) and zero ($\mathbf{0}$) operators are defined as follows

$$\mathcal{I} = \{(x, x) \mid x \in \mathbb{R}^d\}, \quad \mathbf{0} = \{(x, \mathbf{0}) \mid x \in \mathbb{R}^d\}.$$

We say an operator $\mathcal{T}$ is $L$-Lipschitz ($L > 0$) if

$$\|\mathcal{T}(x) - \mathcal{T}(y)\| \leq L\|x - y\|, \ \ \forall x, y \in \text{dom } \mathcal{T},$$

i.e.,

$$\|u - v\| \leq L\|x - y\|, \quad \forall (x, u), (y, v) \in \mathcal{T}.$$

The *inverse operator* of $\mathcal{T}$ is defined as

$$\mathcal{T}^{-1} = \{(y, x) \mid (x, y) \in \mathcal{T}\}.$$

When $\mathbf{0} \in \mathcal{T}(x)$, we say that $x$ is a *zero* of $\mathcal{T}$. We write the zero set of an operator $\mathcal{T}$ as

$$\text{Zer } \mathcal{T} = \{x \mid \mathbf{0} \in \mathcal{T}(x)\} = \mathcal{T}^{-1}(\mathbf{0}).$$

### B.2 MONOTONE OPERATORS

An operator $\mathcal{T}$ on $\mathbb{R}^d$ is said to be *monotone* if

$$\langle \boldsymbol{u} - \boldsymbol{v}, \boldsymbol{x} - \boldsymbol{y} \rangle \geq 0, \quad \forall (\boldsymbol{x}, \boldsymbol{u}), (\boldsymbol{y}, \boldsymbol{v}) \in \mathcal{T},$$

where $\langle \cdot, \cdot \rangle$ denotes the inner product between two vectors. Equivalently, we can express monotonicity as

$$\langle \mathcal{T}(\boldsymbol{x}) - \mathcal{T}(\boldsymbol{y}), \boldsymbol{x} - \boldsymbol{y} \rangle \geq 0, \quad \forall \boldsymbol{x}, \boldsymbol{y} \in \mathbb{R}^d.$$

Furthermore, we say the operator $\mathcal{T}$ is *maximal monotone* if there is no other monotone operator $\mathcal{S}$ s.t. $\mathrm{Gra}\,\mathcal{T} \subset \mathrm{Gra}\,\mathcal{S}$ properly. In other words, if the monotone operator $\mathcal{T}$ is not maximal, then there exists $(\boldsymbol{x}, \boldsymbol{u}) \notin \mathcal{T}$ s.t. $\mathcal{T} \cup \{(\boldsymbol{x}, \boldsymbol{u})\}$ is still monotone. A continuous monotone function $\mathcal{F} : \mathbb{R}^d \to \mathbb{R}^d$ is maximal monotone.

An operator $\mathcal{T} : \mathbb{R}^d \rightrightarrows \mathbb{R}^d$ is *B-strongly monotone* or $B$-coercive if $B > 0$ and

$$\langle \boldsymbol{u} - \boldsymbol{v}, \boldsymbol{x} - \boldsymbol{y} \rangle \geq B \|\boldsymbol{x} - \boldsymbol{y}\|^2, \quad \forall (\boldsymbol{x}, \boldsymbol{u}), (\boldsymbol{y}, \boldsymbol{v}) \in \mathcal{T}.$$

We say $\mathcal{T}$ is strongly monotone if it is $B$-strongly monotone for some unspecified constant $B \in (0, \infty)$. In particular, a linear operator $\mathcal{F}(\boldsymbol{x}) = \boldsymbol{G}\boldsymbol{x} + \boldsymbol{h}$ for $\boldsymbol{G} \in \mathbb{R}^{d \times d}$ and $\boldsymbol{h} \in \mathbb{R}^d$ is maximal monotone if and only if $\boldsymbol{G} + \boldsymbol{G}^\top \succeq \boldsymbol{0}$ ($\boldsymbol{0}$ stands for the matrix whose entries are all zero) and $B$-strongly monotone if $\frac{1}{2}(\boldsymbol{G} + \boldsymbol{G}^\top) \succeq B\boldsymbol{I}$. Similarly, a subdifferentiable operator $\partial f$ is maximal monotone if and only if $f$ is a convex closed proper (CCP) function.

An operator $\mathcal{T}$ is *$\beta$-cocoercive* or *$\beta$-inverse strongly monotone* if $\beta > 0$ and

$$\langle \boldsymbol{u} - \boldsymbol{v}, \boldsymbol{x} - \boldsymbol{y} \rangle \geq \beta \|\boldsymbol{u} - \boldsymbol{v}\|^2, \quad \forall (\boldsymbol{x}, \boldsymbol{u}), (\boldsymbol{y}, \boldsymbol{v}) \in \mathcal{T}.$$

We say $\mathcal{T}$ is cocoercive if it is $\beta$-cocoercive for some unspecified constant $\beta \in (0, \infty)$. In particular, if the linear operator $\mathcal{F}(\boldsymbol{x}) = \boldsymbol{G}\boldsymbol{x} + \boldsymbol{h}$ is $B$-strongly monotone and $L$-Lipschitz, then $\mathcal{F}$ is $\frac{B}{L^2}$-cocoercive.

## C A BRIEF REVIEW OF OPERATOR SPLITTING SCHEMES

In this section, we provide a brief review of a few celebrated operator splitting schemes for solving fixed-point equilibrium equations.

### C.1 RESOLVENT AND CAYLEY OPERATORS

The *resolvent* and *Cayley* operators of an operator $\mathcal{T}$ is defined as, respectively, as follows

$$\mathcal{R}_\mathcal{T} = (\mathcal{I} + \alpha \mathcal{T})^{-1},$$

and

$$\mathcal{C}_\mathcal{T} = 2\mathcal{R}_\mathcal{T} - \mathcal{I},$$

where $\alpha > 0$ is a constant. The resolvent and Cayley operators are both non-expansive, i.e. they both have Lipschitz constant $L \leq 1$ for any maximal monotone operator $\mathcal{T}$, and the resolvent operator $\mathcal{R}_\mathcal{T}$ is contractive (i.e. $L < 1$) for strongly monotone $\mathcal{T}$, the Cayley operator $\mathcal{C}_\mathcal{T}$ is contractive for strongly monotone and Lipschitz $\mathcal{T}$.

There are two well-known properties associated with the resolvent operators:

- First, when $\mathcal{F}(\boldsymbol{x}) = \boldsymbol{G}\boldsymbol{x} + \boldsymbol{h}$ is a linear operator, then

$$\mathcal{R}_\mathcal{F}(\boldsymbol{x}) = (\boldsymbol{I} + \alpha \boldsymbol{G})^{-1}(\boldsymbol{x} - \alpha \boldsymbol{h}).$$

- Second, when $\mathcal{F} = \partial f$ for some CCP function $f$, then the resolvent is given by the following proximal operator

$$\mathcal{R}_\mathcal{F}(\boldsymbol{x}) = \mathrm{prox}_f^\alpha(\boldsymbol{x}) := \arg\min_{\boldsymbol{z}} \left\{ \frac{1}{2} \|\boldsymbol{x} - \boldsymbol{z}\|^2 + \alpha f(\boldsymbol{z}) \right\}.$$

## C.2 OPERATOR SPLITTING SCHEMES

Operator splitting schemes refer to methods to find a zero in a sum of operators (assumed here to be maximal monotone), i.e. find $\boldsymbol{x}$ s.t.

$$\boldsymbol{0} \in (\mathcal{F} + \mathcal{G})(\boldsymbol{x}).$$

We present a few popular operator splitting schemes for solving the above monotone inclusion problem.

- *Forward-backward splitting (FB)*: Consider the monotone inclusion problem

$$\text{find}_{\boldsymbol{x} \in \mathbb{R}^d} \ \boldsymbol{0} \in (\mathcal{F} + \mathcal{G})(\boldsymbol{x}),$$

  where $\mathcal{F}$ and $\mathcal{G}$ are maximal monotone and $\mathcal{F}$ is single-valued. Then for any $\alpha > 0$, we have

$$\begin{aligned}
\boldsymbol{0} \in (\mathcal{F} + \mathcal{G})(\boldsymbol{x}) &\Leftrightarrow \boldsymbol{0} \in (\mathcal{I} + \alpha\mathcal{G})(\boldsymbol{x}) - (\mathcal{I} - \alpha\mathcal{F})(\boldsymbol{x}) \\
&\Leftrightarrow (\mathcal{I} + \alpha\mathcal{G})(\boldsymbol{x}) \ni (\mathcal{I} - \alpha\mathcal{F})(\boldsymbol{x}) \\
&\Leftrightarrow \boldsymbol{x} = \mathcal{R}_{\mathcal{G}}(\mathcal{I} - \alpha\mathcal{F})(\boldsymbol{x}).
\end{aligned}$$

  Therefore, $\boldsymbol{x}$ is a solution if and only if it is a fixed point of $\mathcal{R}_{\mathcal{G}}(\mathcal{I} - \alpha\mathcal{F})$. Moreover, assume $\mathcal{F}$ is $\beta$-cocoercive, then the Picard iteration using forward-backward splitting can be written as

$$\boldsymbol{x}^{(k+1)} = \mathcal{R}_{\mathcal{G}}(\boldsymbol{x}^{(k)} - \alpha\mathcal{F}\boldsymbol{x}^{(k)}),$$

  which converges if $\alpha \in (0, 2\beta)$ and $\text{Zer}(\mathcal{F} + \mathcal{G}) \neq \emptyset$.

- *Peaceman-Rachford splitting (PR)*: Consider the following monotone inclusion problem

$$\text{find}_{\boldsymbol{x} \in \mathbb{R}^d} \ \boldsymbol{0} \in (\mathcal{F} + \mathcal{G})(\boldsymbol{x}),$$

  where $\mathcal{F}$ and $\mathcal{G}$ are maximal monotone. For any $\alpha > 0$, we have

$$\begin{aligned}
\boldsymbol{0} \in (\mathcal{F} + \mathcal{G})(\boldsymbol{x}) &\Leftrightarrow \boldsymbol{0} \in (\mathcal{I} + \alpha\mathcal{F})(\boldsymbol{x}) - (\mathcal{I} - \alpha\mathcal{G})(\boldsymbol{x}) \\
&\Leftrightarrow \boldsymbol{0} \in (\mathcal{I} + \alpha\mathcal{F})(\boldsymbol{x}) - \mathcal{C}_{\mathcal{G}}(\mathcal{I} + \alpha\mathcal{G})(\boldsymbol{x}) \\
&\Leftrightarrow \boldsymbol{0} \in (\mathcal{I} + \alpha\mathcal{F})(\boldsymbol{x}) - \mathcal{C}_{\mathcal{G}}(\boldsymbol{z}), \ \boldsymbol{z} \in (\mathcal{I} + \alpha\mathcal{G})(\boldsymbol{x}) \\
&\Leftrightarrow \mathcal{C}_{\mathcal{G}}(\boldsymbol{z}) \in (\mathcal{I} + \alpha\mathcal{F})\mathcal{R}_{\mathcal{G}}(\boldsymbol{z}), \ \boldsymbol{x} = \mathcal{R}_{\mathcal{G}}(\boldsymbol{z}) \\
&\Leftrightarrow \mathcal{R}_{\mathcal{F}}\mathcal{C}_{\mathcal{G}}(\boldsymbol{z}) = \mathcal{R}_{\mathcal{G}}(\boldsymbol{z}), \ \boldsymbol{x} = \mathcal{R}_{\mathcal{G}}(\boldsymbol{z}) \\
&\Leftrightarrow \mathcal{C}_{\mathcal{F}}\mathcal{C}_{\mathcal{G}}(\boldsymbol{z}) = \boldsymbol{z}, \ \boldsymbol{x} = \mathcal{R}_{\mathcal{G}}(\boldsymbol{z}).
\end{aligned}$$

  Therefore, $\boldsymbol{x}$ is a solution if and only if there is a solution of the fixed-point equilibrium equation $\boldsymbol{z} = \mathcal{C}_{\mathcal{F}}\mathcal{C}_{\mathcal{G}}(\boldsymbol{z})$ and $\boldsymbol{x} = \mathcal{R}_{\mathcal{G}}(\boldsymbol{z})$, which is called *Peaceman-Rachford splitting*.

- *Douglas-Rachford splitting (DR)*: Sometimes the operator $\mathcal{C}_{\mathcal{F}}\mathcal{C}_{\mathcal{G}}$ is merely nonexpansive, the Picard iteration with PR given below

$$\boldsymbol{z}^{(k+1)} = \mathcal{C}_{\mathcal{F}}\mathcal{C}_{\mathcal{G}}(\boldsymbol{z}^{(k)})$$

  is not guaranteed to converge. To guarantee convergence, we note that for any $\forall \alpha > 0$, we have

$$\boldsymbol{0} \in (\mathcal{F} + \mathcal{G})(\boldsymbol{x}) \ \Leftrightarrow \ \left(\frac{1}{2}\mathcal{I} + \frac{1}{2}\mathcal{C}_{\mathcal{F}}\mathcal{C}_{\mathcal{G}}\right)(\boldsymbol{z}) = \boldsymbol{z}, \ \boldsymbol{x} = \mathcal{J}_{\mathcal{G}}(\boldsymbol{z}).$$

  And the above splitting is called *Douglas-Rachford splitting*. The Picard iteration with DR can be written as follows

$$\begin{aligned}
\boldsymbol{x}^{(k+1/2)} &= \mathcal{R}_{\mathcal{G}}(\boldsymbol{z}^{(k)}) \\
\boldsymbol{x}^{(k+1)} &= \mathcal{R}_{\mathcal{F}}(2\boldsymbol{x}^{(k+1/2)} - \boldsymbol{z}^{(k)}) \\
\boldsymbol{z}^{(k+1)} &= \boldsymbol{z}^{(k)} + \boldsymbol{x}^{(k+1)} - \boldsymbol{x}^{(k+1/2)}
\end{aligned}$$

  which converges for any $\alpha > 0$ if $\text{Zer}(\mathcal{F} + \mathcal{G}) \neq \emptyset$.

## D    PROPERTIES OF KRONECKER PRODUCT

In this section, we collect some Kronecker product results that are used in this paper.

**Definition 1.** *Let $\boldsymbol{A} \in \mathbb{R}^{p \times q}$, $\boldsymbol{B} \in \mathbb{R}^{r \times s}$ be two matrices. Their Kronecker product $\boldsymbol{A} \times \boldsymbol{B} \in \mathbb{R}^{pr \times qs}$ is defined as follows:*

$$
\boldsymbol{A} \otimes \boldsymbol{B} = \begin{bmatrix} A_{11}\boldsymbol{B} & \dots & A_{1q}\boldsymbol{B} \\ \vdots & & \vdots \\ A_{p1}\boldsymbol{B} & \dots & A_{pq}\boldsymbol{B} \end{bmatrix}
$$

The following identities about Kronecker product hold:

- $(\boldsymbol{A} \otimes \boldsymbol{B})^\top = \boldsymbol{A}^\top \otimes \boldsymbol{B}^\top \quad \forall \boldsymbol{A} \in \mathbb{R}^{p \times q}, B \in \mathbb{R}^{r \times s}$
- $\|\boldsymbol{A} \otimes \boldsymbol{B}\| = \|\boldsymbol{A}\|\|\boldsymbol{B}\| \quad \forall \boldsymbol{A} \in \mathbb{R}^{p \times q}, B \in \mathbb{R}^{r \times s}$
- $\|\boldsymbol{A} \otimes \boldsymbol{B}\|_\infty = \|\boldsymbol{A}\|_\infty \|\boldsymbol{B}\|_\infty \quad \forall \boldsymbol{A} \in \mathbb{R}^{p \times q}, B \in \mathbb{R}^{r \times s}$
- $(\boldsymbol{A} \otimes \boldsymbol{B})\mathrm{vec}(\boldsymbol{C}) = \mathrm{vec}(\boldsymbol{B}\boldsymbol{C}\boldsymbol{A}^\top) \quad \forall \boldsymbol{A} \in \mathbb{R}^{s,r}, \boldsymbol{B} \in \mathbb{R}^{p \times q}, \boldsymbol{C} \in \mathbb{R}^{q \times r}$
- $(\boldsymbol{A} \otimes \boldsymbol{B}) \otimes \boldsymbol{C} = \boldsymbol{A} \otimes (\boldsymbol{B} \otimes \boldsymbol{C}) \quad \forall \boldsymbol{A} \in \mathbb{R}^{m,n}, \boldsymbol{B} \in \mathbb{R}^{p \times q}, \boldsymbol{C} \in \mathbb{R}^{r \times s}$
- $\boldsymbol{A} \otimes (\boldsymbol{B} + \boldsymbol{C}) = \boldsymbol{A} \otimes \boldsymbol{B} + \boldsymbol{A} \otimes \boldsymbol{C} \quad \forall \boldsymbol{A} \in \mathbb{R}^{p \times q}, \boldsymbol{B}, \boldsymbol{C} \in \mathbb{R}^{r \times s}$
- $(\boldsymbol{A} + \boldsymbol{B}) \otimes \boldsymbol{C} = \boldsymbol{A} \otimes \boldsymbol{C} + \boldsymbol{B} \otimes \boldsymbol{C} \quad \forall \boldsymbol{A}, \boldsymbol{B} \in \mathbb{R}^{p \times q}, \boldsymbol{C} \in \mathbb{R}^{r \times s}$
- $(\boldsymbol{A} \otimes \boldsymbol{B})(\boldsymbol{C} \otimes \boldsymbol{D}) = \boldsymbol{A}\boldsymbol{C} \otimes \boldsymbol{B}\boldsymbol{D} \quad \forall \boldsymbol{A} \in \mathbb{R}^{p \times q}, \boldsymbol{B} \in \mathbb{R}^{r \times s}, \boldsymbol{C} \in \mathbb{R}^{q \times k}, \boldsymbol{D} \in \mathbb{R}^{s \times l}$

**Proposition 3** ([43, Theorem 4.2.12]). *Let $\boldsymbol{A} \in \mathbb{R}^{n \times n}$ and $\boldsymbol{B} \in \mathbb{R}^{m \times m}$. If we denote the eigenvalue sets of $\boldsymbol{A}$ and $\boldsymbol{B}$ as $\Lambda(\boldsymbol{A}) = \{\lambda_1(\boldsymbol{A}), \dots, \lambda_n(\boldsymbol{A})\}$ and $\Lambda(\boldsymbol{B}) = \{\lambda_1(\boldsymbol{B}), \dots, \lambda_m(\boldsymbol{B})\}$, then the eigenvalue set of $\boldsymbol{A} \otimes \boldsymbol{B}$ is $\Lambda(\boldsymbol{A} \otimes \boldsymbol{B}) = \{\lambda_i(\boldsymbol{A}) \cdot \lambda_j(\boldsymbol{B}), i = 1, \dots, n, j = 1, \dots, m\}$.*

## E    TECHNICAL PROOFS

### E.1    LIPSCHITZ CONSTANT VS. LARGEST MAGNITUDE OF EIGENVALUE

Let $f(\boldsymbol{Z}) = \boldsymbol{W}\boldsymbol{Z}\boldsymbol{G} + \boldsymbol{B}$ be a linear map. With slight abuse of notation, we still denote the vectorized version of $f$ as $f$ which reads $f(\mathrm{vec}(\boldsymbol{Z})) = (\boldsymbol{G}^\top \otimes \boldsymbol{W})\mathrm{vec}(\boldsymbol{Z}) + \mathrm{vec}(\boldsymbol{B})$ (See Appendix D for properties of the Kronecker product). The Lipschitz constant $\mathrm{Lip}_\infty(f)$ of the linear map $f$ with respect to the $\ell_\infty$ vector norm is exactly the $\infty$-norm $\|\boldsymbol{G} \otimes \boldsymbol{W}\|_\infty = \|\boldsymbol{G}^\top\|_\infty \|\boldsymbol{W}\|_\infty$. Recall the following general result about matrix norm and the largest magnitude of eigenvalue.

**Theorem 1** ([47, Theorem 4 in Section 4.6]). *The largest magnitude of eigenvalue $\lambda_1(\boldsymbol{A})$ of a matrix $\boldsymbol{A}$ satisfies*

$$
\lambda_1(\boldsymbol{A}) = \inf_{\|\cdot\|_\mathrm{M}} \|\boldsymbol{A}\|_\mathrm{M}
$$

*in which the infimum is taken over all subordinate matrix norms $\|\cdot\|_\mathrm{M}$ including 2-norm and $\infty$-norm.*

Meanwhile, note that one has $\|\boldsymbol{W}\|_\infty = \||\boldsymbol{W}|\|_\infty$ by definition. Hence one has $\mathrm{Lip}_\infty(f) = \|\boldsymbol{G}^\top\|_\infty \|\boldsymbol{W}\|_\infty \geq \lambda_1(\boldsymbol{G}^\top)\lambda_1(|\boldsymbol{W}|)$. Note that, when $\boldsymbol{G}$ is the normalized adjacency matrix of undirected graph $\hat{\boldsymbol{A}}$, we have $\lambda_1(\boldsymbol{G}^\top) = \lambda_1(\boldsymbol{G}) = 1$ and hence we have $\mathrm{Lip}_\infty(f) \geq \lambda_1(|\boldsymbol{W}|)$.

### E.2    PROOFS FOR SECTION 2

*Proof of Proposition 1.* First recall the operator splitting problem 3 in Section 1:

$$
\text{find } \boldsymbol{0} \in (\mathcal{F} + \mathcal{G})(\mathrm{vec}(\boldsymbol{Z})),
$$

where

$$
\mathcal{F}(\mathrm{vec}(\boldsymbol{Z})) = (\boldsymbol{I} - \boldsymbol{G}^\top \otimes \boldsymbol{W})\mathrm{vec}(\boldsymbol{Z}) - \mathrm{vec}(g_{\boldsymbol{B}}(\boldsymbol{X})) \text{ and } \mathcal{G} = \partial f,
$$

here $f$ is the indicator of the positive octant, i.e. $f(x) = I\{x \geq 0\}$ for which we have $\mathrm{prox}_f^\alpha$ equals $\sigma$, the ReLU activation function, for all $\alpha > 0$. Note that, from the condition $\boldsymbol{K} = \frac{1}{2}(\boldsymbol{G}^\top \otimes \boldsymbol{W} + \boldsymbol{G} \otimes \boldsymbol{W}^\top) \preceq (1-m)\boldsymbol{I}$, one has $\boldsymbol{G}^\top \otimes \boldsymbol{W} \preceq (1-m)\boldsymbol{I}$ and hence

$$
\boldsymbol{I} - \boldsymbol{G}^\top \otimes \boldsymbol{W} \succeq m\boldsymbol{I}
$$

which says $\mathcal{F}$ is $m$-strongly monotone for some $m > 0$. As the function $\mathcal{F}$ is a linear and hence continuous function defined on the entire $\mathbb{R}^{d \times n}$, it is then automatically maximal monotone once it is monotone. Since $f$ is a CCP function, its subdifferential operator $\mathcal{G} = \partial f$ is maximal monotone. In particular, as the linear map $\mathcal{F}$ is single-valued, we can apply the FB splitting scheme in Appendix C.2 as the following: for any $\alpha > 0$, we have

$$\mathbf{0} \in (\mathcal{F} + \mathcal{G})(\text{vec}(\boldsymbol{Z})) \Leftrightarrow \text{vec}(\boldsymbol{Z}) = \mathcal{R}_{\mathcal{G}}(\mathcal{I} - \alpha\mathcal{F})(\text{vec}(\boldsymbol{Z})).$$

$$\Leftrightarrow \text{vec}(\boldsymbol{Z}) = \text{prox}_f^{\alpha}\left(\text{vec}(\boldsymbol{Z}) - \alpha \cdot \left(\text{vec}(\boldsymbol{Z}) - \boldsymbol{G}^{\top} \otimes \boldsymbol{W}\text{vec}(\boldsymbol{Z}) - \text{vec}(g_{\boldsymbol{B}}(\boldsymbol{X}))\right)\right),$$

$$\Leftrightarrow \text{vec}(\boldsymbol{Z}) = \sigma\left(\text{vec}(\boldsymbol{Z}) - \alpha \cdot \left(\text{vec}(\boldsymbol{Z}) - \boldsymbol{G}^{\top} \otimes \boldsymbol{W}\text{vec}(\boldsymbol{Z}) - \text{vec}(g_{\boldsymbol{B}}(\boldsymbol{X}))\right)\right).$$

When $\alpha = 1$ in the last above, we recover the MIGNN model 2:

$$\text{vec}(\boldsymbol{Z}) = \sigma(\boldsymbol{G}^{\top} \otimes \boldsymbol{W}\text{vec}(\boldsymbol{Z}) + \text{vec}(g_{\boldsymbol{B}}(\boldsymbol{X})))$$

This shows the equivalence between finding a fixed point of MIGNN model 2 and finding a zero of the operator splitting problem 3. Therefore, when $\boldsymbol{K} \preceq (1 - m)\boldsymbol{I}$, the linear map $\mathcal{F}$ is strongly monotone and Lipschitz, the monotone splitting problem and hence the MIGNN model is well-sposed, see Appendix C.2. □

### E.3 PROOFS FOR SECTION 3

The following properties of the Cayley map are used in this paper.

**Proposition 4.** *Let $\boldsymbol{S}$ be a skew-symmetric matrix. Then its image under the Cayley map $\text{Cay}(\boldsymbol{S}) := (\boldsymbol{I} - \boldsymbol{S})(\boldsymbol{I} + \boldsymbol{S})^{-1}$ is an orthogonal matrix, and hence the magnitude of all its eigenvalues is $1$.*

*Proof.* To verify that the Cayley map is well-defined, it suffices to show that $-1$ is not an eigenvalue of $\boldsymbol{S}$. This can be derived from the general fact that each eigenvalue of any skew-symmetric matrix is purely imaginary. To see this, let $\lambda$ be an eigenvalue of $\boldsymbol{S}$ with corresponding eigenvector $\boldsymbol{v}$ where both $\lambda$ and $\boldsymbol{v}$ possibly contain complex numbers. Let $\boldsymbol{v}^{\text{H}}$ and $\boldsymbol{S}^{\text{H}}$ denote the conjugate transpose of the vector $\boldsymbol{v}$ and the matrix $\boldsymbol{S}$ respectively. We then have

$$\boldsymbol{v}^{\text{H}}\boldsymbol{S}\boldsymbol{v} = \boldsymbol{v}^{\text{H}}(\lambda\boldsymbol{v}) = \lambda|\boldsymbol{v}|_{\mathbb{C}}^2,$$

where $||_{\mathbb{C}}$ denotes the Euclidean norm for a complex vector. At the same time, one has

$$\boldsymbol{v}^{\text{H}}\boldsymbol{S}\boldsymbol{v} = (\boldsymbol{S}^{\text{H}}\boldsymbol{v})^{\text{H}}\boldsymbol{v} = (-\boldsymbol{S}\boldsymbol{v})^{\text{H}}\boldsymbol{v} = -\bar{\lambda}|\boldsymbol{v}|_{\mathbb{C}}^2,$$

where $\bar{\lambda}$ denotes the complex conjugate of $\lambda$. Hence $\lambda = -\bar\lambda$, that is $\lambda$ is purely imaginary. This concludes the proof that $(\boldsymbol{I} - \boldsymbol{S})(\boldsymbol{I} + \boldsymbol{S})^{-1}$ is well-defined.

Note that $(\boldsymbol{I} - \boldsymbol{S})(\boldsymbol{I} + \boldsymbol{S})^{-1}\left((\boldsymbol{I} - \boldsymbol{S})(\boldsymbol{I} + \boldsymbol{S})^{-1}\right)^{\top} = (\boldsymbol{I} - \boldsymbol{S})(\boldsymbol{I} + \boldsymbol{S})^{-1}(\boldsymbol{I} + \boldsymbol{S})(\boldsymbol{I} - \boldsymbol{S})^{-1} = \boldsymbol{I}$. Therefore, $(\boldsymbol{I} - \boldsymbol{S})(\boldsymbol{I} + \boldsymbol{S})^{-1}$ is (real) orthogonal.

In the last part, we present a short proof that the magnitude of all eigenvalues of a (real) orthogonal matrix $\boldsymbol{O}$ equals 1. Let $\lambda_{\boldsymbol{O}}$ be an eigenvalue of $\boldsymbol{O}$ and $\boldsymbol{w}$ is its eigenvector. Then we have

$$|\lambda_{\boldsymbol{O}}||\boldsymbol{w}|_{\mathbb{C}}^2 = (\boldsymbol{O}\boldsymbol{w})^{\text{H}}(\boldsymbol{O}\boldsymbol{w}) = \boldsymbol{w}^{\text{H}}\boldsymbol{O}^{\text{H}}\boldsymbol{O}\boldsymbol{w} = (\boldsymbol{O}\boldsymbol{w})^{\text{H}}(\boldsymbol{O}\boldsymbol{w}) = \boldsymbol{w}^{\text{H}}\boldsymbol{O}^{\top}\boldsymbol{O}\boldsymbol{w} = |\boldsymbol{w}|_{\mathbb{C}}^2.$$

Hence, $|\lambda_{\boldsymbol{O}}| = 1$. □

*Proof of Proposition 2.* Since the normalized Laplacian $\boldsymbol{L}$ is symmetric, we have

$$\boldsymbol{K} = \frac{1}{2}\left(\frac{1}{2}\boldsymbol{L}^{\top} \otimes \boldsymbol{W} + \frac{1}{2}\boldsymbol{L} \otimes \boldsymbol{W}^{\top}\right) = \frac{1}{2}\boldsymbol{L} \otimes \left(\frac{1}{2}\left(\boldsymbol{W} + \boldsymbol{W}^{\top}\right)\right).$$

The property of Kronecker product (Theorem 3) tells us that the eigenvalues of $\boldsymbol{K}$ are the products of the eigenvalues of $\boldsymbol{L}$ and $\left(\frac{1}{2}(\boldsymbol{W} + \boldsymbol{W}^{\top})\right)$. Therefore, the MIGNN model satisfies the well-posedness condition in Proposition 1 once

$$\lambda_i\left(\frac{1}{2}\boldsymbol{L}\right)\lambda_j\left(\frac{1}{2}(\boldsymbol{W} + \boldsymbol{W}^{\top})\right) \leq 1 - m$$

for all eigenvalues from $\frac{1}{2}\boldsymbol{L}$ and $\left(\frac{1}{2}(\boldsymbol{W} + \boldsymbol{W}^\top)\right)$. Notice that $\frac{1}{2}\boldsymbol{L}$ is positive semi-definite and all its eigenvalues are within $[0, 1]$. Therefore, $\boldsymbol{W}$ guarantees the well-posedness of MIGNN as long as all eigenvalues satisfy

$$\lambda_i \left(\frac{1}{2}(\boldsymbol{W} + \boldsymbol{W}^\top)\right) \leq 1 - m.$$

When $\boldsymbol{W} = (1 - m)\boldsymbol{I} - \boldsymbol{C}\boldsymbol{C}^\top + \boldsymbol{F} - \boldsymbol{F}^\top$, we have $\frac{1}{2}(\boldsymbol{W} + \boldsymbol{W}^\top) = (1 - m)\boldsymbol{I} - \boldsymbol{C}\boldsymbol{C}^\top$. As $\boldsymbol{C}\boldsymbol{C}^\top$ is positive semi-definite, all eigenvalues of $\frac{1}{2}(\boldsymbol{W} + \boldsymbol{W}^\top)$ are no more than $(1 - m)$. $\qquad\square$

### E.4 Proofs for Section 4

The following result about Kronecker product is adapted from [58] which we include here for completeness.

*Proof of Formula 9 used in Section 4.* Since $\boldsymbol{G}^\top$ is symmetric, it admits an eigen-decomposition $\boldsymbol{G}^\top = \boldsymbol{Q}_{\boldsymbol{G}^\top}\boldsymbol{\Lambda}_{\boldsymbol{G}^\top}\boldsymbol{Q}_{\boldsymbol{G}^\top}^\top$ where $\boldsymbol{Q}_{\boldsymbol{G}^\top}$ is orthogonal and hence satisfies $\boldsymbol{Q}_{\boldsymbol{G}^\top}^{-1} = \boldsymbol{Q}_{\boldsymbol{G}^\top}$. As $\boldsymbol{W}$ is diagonalizable, it admits a eigen-decomposition $\boldsymbol{W} = \boldsymbol{Q}_{\boldsymbol{W}}\boldsymbol{\Lambda}_{\boldsymbol{W}}\boldsymbol{Q}_{\boldsymbol{W}}^{-1}$. Then we can write

$$\boldsymbol{G}^\top \otimes \boldsymbol{W} = [\boldsymbol{Q}_{\boldsymbol{G}^\top}\boldsymbol{\Lambda}_{\boldsymbol{G}^\top}\boldsymbol{Q}_{\boldsymbol{G}^\top}^\top] \otimes [\boldsymbol{Q}_{\boldsymbol{W}}\boldsymbol{\Lambda}_{\boldsymbol{W}}\boldsymbol{Q}_{\boldsymbol{W}}^{-1}] = [\boldsymbol{Q}_{\boldsymbol{G}^\top} \otimes \boldsymbol{Q}_{\boldsymbol{W}}][\boldsymbol{\Lambda}_{\boldsymbol{G}^\top} \otimes \boldsymbol{\Lambda}_{\boldsymbol{W}}][\boldsymbol{Q}_{\boldsymbol{G}^\top}^\top \otimes \boldsymbol{Q}_{\boldsymbol{W}}^{-1}]$$

Let $n = \dim(\boldsymbol{G})$ and $d = \dim(\boldsymbol{W})$, we have

$$\boldsymbol{I}_{nd} = \boldsymbol{I}_n \otimes \boldsymbol{I}_d = [\boldsymbol{Q}_{\boldsymbol{G}^\top}\boldsymbol{I}_n\boldsymbol{Q}_{\boldsymbol{G}^\top}^\top] \otimes [\boldsymbol{Q}_{\boldsymbol{W}}\boldsymbol{I}_m\boldsymbol{Q}_{\boldsymbol{W}}^{-1}] = [\boldsymbol{Q}_{\boldsymbol{G}^\top} \otimes \boldsymbol{Q}_{\boldsymbol{W}}][\boldsymbol{I}_n \otimes \boldsymbol{I}_m][\boldsymbol{Q}_{\boldsymbol{G}^\top}^\top \otimes \boldsymbol{Q}_{\boldsymbol{W}}^{-1}]$$

Therefore, for some matrix $\boldsymbol{B} \in \mathbb{R}^{d \times n}$,

$$\boldsymbol{V}(\mathrm{vec}(\boldsymbol{U})) = \frac{1}{1 + \alpha}\left(\boldsymbol{I}_{nd} - \frac{\alpha}{1 + \alpha}(\boldsymbol{G}^\top \otimes \boldsymbol{W})\right)^{-1}(\mathrm{vec}(\boldsymbol{U}))$$

$$= \frac{1}{1 + \alpha}\left(\boldsymbol{I}_{nd} - \frac{\alpha}{1 + \alpha}(\boldsymbol{G}^\top \otimes \boldsymbol{W})\right)^{-1}(\mathrm{vec}(\boldsymbol{U}))$$

$$\frac{1}{1 + \alpha}\left([\boldsymbol{Q}_{\boldsymbol{G}^\top} \otimes \boldsymbol{Q}_{\boldsymbol{W}}]\left[\boldsymbol{I}_{nd} - \frac{\alpha}{1 + \alpha}\boldsymbol{\Lambda}_{\boldsymbol{G}^\top} \otimes \boldsymbol{\Lambda}_{\boldsymbol{W}}\right][\boldsymbol{Q}_{\boldsymbol{G}^\top}^\top \otimes \boldsymbol{Q}_{\boldsymbol{W}}^{-1}]\right)^{-1}(\mathrm{vec}(\boldsymbol{U}))$$

$$\frac{1}{1 + \alpha}\left([\boldsymbol{Q}_{\boldsymbol{G}^\top} \otimes \boldsymbol{Q}_{\boldsymbol{W}}]\left[\boldsymbol{I}_{nd} - \frac{\alpha}{1 + \alpha}\boldsymbol{\Lambda}_{\boldsymbol{G}^\top} \otimes \boldsymbol{\Lambda}_{\boldsymbol{W}}\right]^{-1}[\boldsymbol{Q}_{\boldsymbol{G}^\top}^\top \otimes \boldsymbol{Q}_{\boldsymbol{W}}^{-1}]\right)(\mathrm{vec}(\boldsymbol{U}))$$

Note that $\left[\boldsymbol{I}_{nd} - \frac{\alpha}{1+\alpha}\boldsymbol{\Lambda}_{\boldsymbol{G}^\top} \otimes \boldsymbol{\Lambda}_{\boldsymbol{W}}\right]$ is a diagonal matrix whose inverse is given by the diagonal matrix $\mathrm{Diag}(\mathrm{vec}(\boldsymbol{H}))$ where the entires of $\boldsymbol{H}$ is given as $H_{ij} := 1/\left(1 - \frac{\alpha}{1+\alpha}(\boldsymbol{\Lambda}_{\boldsymbol{W}})_{ii}(\boldsymbol{\Lambda}_{\boldsymbol{G}^\top})_{jj}\right)$. Here the notation $\mathrm{Diag}(\boldsymbol{v})$ denotes the diagonal matrix that has $\boldsymbol{v}$ as its diagonal for any vector $\boldsymbol{v}$. From this we have,

$$\boldsymbol{V}(\mathrm{vec}(\boldsymbol{U})) = \frac{1}{1 + \alpha}\left([\boldsymbol{Q}_{\boldsymbol{G}^\top} \otimes \boldsymbol{Q}_{\boldsymbol{W}}]\mathrm{Diag}(\mathrm{vec}(\boldsymbol{H}))\left[\boldsymbol{Q}_{\boldsymbol{G}^\top}^\top \otimes \boldsymbol{Q}_{\boldsymbol{W}}^{-1}\right]\right)(\mathrm{vec}(\boldsymbol{U}))$$

$$= \frac{1}{1 + \alpha}\left([\boldsymbol{Q}_{\boldsymbol{G}^\top} \otimes \boldsymbol{Q}_{\boldsymbol{W}}]\mathrm{Diag}(\mathrm{vec}(\boldsymbol{H}))\,\mathrm{vec}(\boldsymbol{Q}_{\boldsymbol{W}}^{-1}\boldsymbol{U}\boldsymbol{Q}_{\boldsymbol{G}^\top})\right)$$

$$= \frac{1}{1 + \alpha}[\boldsymbol{Q}_{\boldsymbol{G}^\top} \otimes \boldsymbol{Q}_{\boldsymbol{W}}]\mathrm{vec}\left(\boldsymbol{H} \odot [\boldsymbol{Q}_{\boldsymbol{W}}^{-1}\boldsymbol{U}\boldsymbol{Q}_{\boldsymbol{G}^\top}]\right)$$

$$= \frac{1}{1 + \alpha}\mathrm{vec}\left(\boldsymbol{Q}_{\boldsymbol{W}}[\boldsymbol{H} \odot [\boldsymbol{Q}_{\boldsymbol{W}}^{-1}\boldsymbol{U}\boldsymbol{Q}_{\boldsymbol{G}^\top}]]\boldsymbol{Q}_{\boldsymbol{G}^\top}^\top\right)$$

where $\odot$ denotes entry-wise multiplication. $\qquad\square$

For the reader's convenience, we present the following fact that implies $\tilde{\boldsymbol{D}}^{-1/2}(\boldsymbol{A} + \boldsymbol{A}^2 + \cdots + \boldsymbol{A}^P)\tilde{\boldsymbol{D}}^{-1/2}$ has its eigenvalues within $[-1, 1]$ which is used in MIGNN with diffusion convolution (Equation 12).

**Proposition 5.** *Let $\boldsymbol{S} \in \mathbb{R}^{n \times n}$ be non-singular symmetric matrix and let $\boldsymbol{D}$ be the degree matrix defined as the diagonal matrix where $D_{ii} = \sum_{j=1}^n |S_{ij}|$. Since $\boldsymbol{S}$ is non-singular, $\boldsymbol{D}^{-1/2}$ is well-defined. Then the normalization $\tilde{\boldsymbol{S}} := \boldsymbol{D}^{-1/2}\boldsymbol{S}\boldsymbol{D}^{-1/2}$ of $\boldsymbol{S}$ has its eigenvalues with $[-1, 1]$.*

*Proof.* Note that, the normalization $\tilde{S}$ satisfies

$$\tilde{S}^{\top} = D^{-1/2} S^{\top} D^{-1/2} = D^{-1/2} S D^{-1/2} = \tilde{S},$$

that is, $\tilde{S}$ is symmetric. To complete the proof, it then suffices to show that both $I + \tilde{S}$ and $I - \tilde{S}$ are positive semi-definite. Indeed, from the construction, both symmetric matrices $D - S$ and $D + S$ are diagonal dominant, and their diagonal entries are positive, hence they are positive semi-definite by Gershgorin's Circle Theorem. Meanwhile, for any vector $v \in \mathbb{R}^n$, we have

$$
\begin{aligned}
v^{\top} (I + \tilde{S}) v &= v^{\top} (I + D^{-1/2} S D^{-1/2}) v \\
&= v^{\top} D^{-1/2} (D + S) D^{-1/2} v \\
&= (D^{-1/2} v)^{\top} (D + S) (D^{-1/2} v) \\
&\geq 0
\end{aligned}
$$

This shows that $I + \tilde{S}$ is positive semi-definite. Similarly, one can derive that $I - \tilde{S}$ is positive semi-definite from $D - S$ is positive semi-definite. $\qquad\square$

## F  MIGNN VIA ANDERSON-ACCELERATED OPERATOR SPLITTING SCHEMES

In this section, we present the pseudocodes of Anderson-accelerated MIGNN operator splitting schemes discussed in Section 4.

### F.1  PSEUDOCODE FOR MIGNN WITH OPERATOR SPLITTING SCHEMES

**FB Splitting.** The detail of the FB splitting scheme iteration function Equation (6) of solving MIGNN is presented in Algorithm 1.

---
**Algorithm 1** FB-forward-MIGNN
---
$\quad Z := 0; \quad \text{err} := 1$
$\quad$**while** err $> \epsilon$ **do**
$\quad\quad Z^{(+)} := (1 - \alpha) Z + \alpha W Z G + \alpha g_B(X)$
$\quad\quad Z^{(+)} := \text{prox}_f^{\alpha}(Z^{(+)})$
$\quad\quad \text{err} := \frac{\|Z^{(+)} - Z\|}{\|Z^{(+)}\|}$
$\quad\quad Z := Z^{(+)}$
$\quad$**end while**
$\quad$**return** $Z$

---

**PR splitting.** The details of the PR splitting scheme encoded in the iteration function Equation (7) of solving MIGNN is presented in Algorithm 2.

---
**Algorithm 2** PR-forward-MIGNN
---
$\quad z, u = \text{vec}(U) := 0; \quad \text{err} := 1; \quad V := (I + \alpha(I - G^{\top} \otimes W))^{-1}$
$\quad$**while** err $> \epsilon$ **do**
$\quad\quad z^{(1/2)} := \text{prox}_f^{\alpha}(u)$
$\quad\quad u^{(1/2)} := 2 z^{(1/2)} - u$
$\quad\quad z^{(+)} := V(u^{(1/2)} + \alpha \, \text{vec}(g_B(X)))$
$\quad\quad u^{(+)} := 2 z^{(+)} - u^{(1/2)}$
$\quad\quad \text{err} := \frac{\|u^{(+)} - u\|}{\|u^{(+)}\|}$
$\quad\quad z, u := z^{(+)}, u^{(+)}$
$\quad$**end while**
$\quad$**return** $\text{prox}_f^{\alpha}(u)$

---

### F.2  MORE DETAILS ON BACKWARD PROPAGATION

In the backward propagation, the following result from [77] allows us to convert the computing of the inverse Jacobian term $(I - J(G^{\top} \otimes W))^{-\top}$ to the (transpose of) matrix inverse term $V = (I - G^{\top} \otimes W))^{-1}$ which is already calculated in the forward pass.

**Proposition 6** (Adapted from [77, Theorem 3]). *Let* $\mathrm{vec}(\boldsymbol{Z}^*)$ *be the fixed point of the MIGNN model (2) and* $\boldsymbol{J}$ *is the Jacobian* $\sigma$ *of the non-linearity at the* $\boldsymbol{G}^\top \otimes \boldsymbol{W} \mathrm{vec}(\boldsymbol{Z}^*) + \mathrm{vec}(g_{\boldsymbol{B}}(\boldsymbol{X}))$. *For any* $\boldsymbol{v} \in \mathbb{R}^n$ *the solution* $\boldsymbol{u}^*$ *of the equation*

$$\boldsymbol{u}^* = (\boldsymbol{I} - \boldsymbol{J}(\boldsymbol{G}^\top \otimes \boldsymbol{W}))^{-\top} \boldsymbol{v}$$

*is given by*

$$\boldsymbol{u}^* = \boldsymbol{v} + (\boldsymbol{G} \otimes \boldsymbol{W}^\top) \tilde{\boldsymbol{u}}^*$$

*where* $\tilde{\boldsymbol{u}}$ *is a solution of the operator splitting problem* $0 \in (\tilde{F} + \tilde{G})(\tilde{u})$, *with operators defined as*

$$\tilde{F}(\tilde{\boldsymbol{u}}) = \left(\boldsymbol{I} - \boldsymbol{G} \otimes \boldsymbol{W}^\top\right)(\tilde{\boldsymbol{u}}), \quad \tilde{G}(\tilde{\boldsymbol{u}}) = \boldsymbol{D}\tilde{\boldsymbol{u}} - \boldsymbol{v} \tag{19}$$

*where* $\boldsymbol{D}$ *is the diagonal matrix defined by* $\boldsymbol{J} = (\boldsymbol{I} + \boldsymbol{D})^{-1}$ *(where* $D_{ii} = \infty$ *if* $J_{ii} = 0$*).*

Note that, since the non-linearity $\sigma$ is applied entry-wise, the Jacobian $\boldsymbol{J}$ is a diagonal matrix, and its diagonal entries consist of the vectorization of the Jacobian $\frac{\partial \sigma(\boldsymbol{W}\boldsymbol{Z}\boldsymbol{G}^\top)}{\partial \boldsymbol{Z}}|_{\boldsymbol{Z}^*}$. Therefore, the Jacobian $\boldsymbol{J}$ and hence $\boldsymbol{D}$ can be efficiently computed. We provide the pseudo-codes of FB and PR splitting schemes for the backward propagation described in the above proposition as Algorithm 3 and Algorithm 4 respectively and their Anderson-accelerated version can be found in Algorithm 7 and Algorithm 8.

**FB backward propagation** We now present the pseudo-code of FB splitting method (Algorithm 3) for the backward propagation with the procedure described in Proposition 6.

---
**Algorithm 3** FB-backward-MIGNN
---

$\boldsymbol{u} = \mathrm{vec}(\boldsymbol{U}) := 0; \quad \mathrm{err} := 1; \quad \boldsymbol{v} := \frac{\partial \ell}{\partial \mathrm{vec}(\boldsymbol{Z}^*)}$

**while** err $> \epsilon$ **do**

$\quad \boldsymbol{u}^{(+)} := (1 - \alpha)\boldsymbol{u} + \alpha\, \mathrm{vec}(\boldsymbol{W}^\top \boldsymbol{U} \boldsymbol{G}^\top)$

$\quad u_i^{(+)} := \begin{cases} \frac{u_i^{(+)} + \alpha v_i}{1 + \alpha(1 + D_{ii})} & \text{if } D_{ii} < \infty \\ 0 & \text{if } D_{ii} = \infty \end{cases}$

$\quad \mathrm{err} := \frac{\|\boldsymbol{u}^{(+)} - \boldsymbol{u}\|}{\|\boldsymbol{u}^{(+)}\|}$

$\quad \boldsymbol{u} := \boldsymbol{u}^{(+)}$

**end while**

Set $\boldsymbol{U} := \mathrm{vec}^{-1}(\boldsymbol{u})$

**return** $\boldsymbol{v} + \mathrm{vec}(\boldsymbol{W}^\top \boldsymbol{U} \boldsymbol{G}^\top)$

---

Let $\boldsymbol{u}^{(k)}$ be the intermediate variable, the procedure of applying FB splitting on monotone splitting problem 19 can be summarized as finding the fixed-point $\boldsymbol{u}^*$ of the following iteration function

$$\boldsymbol{u}^{(k+1)} := B_\alpha^{\mathrm{FB}}(\boldsymbol{u}^{(k)}) = (\boldsymbol{I} + \alpha\boldsymbol{D})^{-1}((1 - \alpha)\boldsymbol{u}^{(k)} + \alpha\boldsymbol{W}^\top\boldsymbol{v}). \tag{20}$$

**PR backward propagation** We now present the pseudo-code of PR splitting method (Algorithm 4) for the backward propagation with the procedure described in Proposition 6.

---
**Algorithm 4** PR-backward-MIGNN
---

$\boldsymbol{y} := 0; \boldsymbol{u} = \mathrm{vec}(\boldsymbol{U}) := 0; \quad \mathrm{err} := 1; \quad \boldsymbol{v} := \frac{\partial \ell}{\partial \mathrm{vec}(\boldsymbol{Z}^*)}; \quad \boldsymbol{V} := (\boldsymbol{I} + \alpha(\boldsymbol{I} - \boldsymbol{G}^\top \otimes \boldsymbol{W}))^{-1}$

**while** err $> \epsilon$ **do**

$\quad u_i^{(1/2)} := \begin{cases} \frac{y_i + \alpha v_i}{1 + \alpha(1 + D_{ii})} & \text{if } D_{ii} < \infty \\ 0 & \text{if } D_{ii} = \infty \end{cases}$

$\quad \boldsymbol{y}^{(1/2)} := 2\boldsymbol{u}^{(1/2)} - \boldsymbol{y}$

$\quad \boldsymbol{u}^{(+)} := \boldsymbol{V}^\top \boldsymbol{y}^{(1/2)}$

$\quad \boldsymbol{y}^{(+)} := 2\boldsymbol{u}^{(+)} - \boldsymbol{y}^{(1/2)}$

$\quad \mathrm{err} := \frac{\|\boldsymbol{y}^{(+)} - \boldsymbol{y}\|}{\|\boldsymbol{y}^{(+)}\|}$

$\quad \boldsymbol{y}, \boldsymbol{u} := \boldsymbol{y}^{(+)}, \boldsymbol{u}^{(+)}$

**end while**

Compute $\boldsymbol{u}$ where $u_i := \begin{cases} \frac{y_i + \alpha v_i}{1 + \alpha(1 + D_{ii})} & \text{if } D_{ii} < \infty \\ 0 & \text{if } D_{ii} := \infty \end{cases}$

Set $\boldsymbol{U} := \mathrm{vec}^{-1}(\boldsymbol{u})$

**return** $\boldsymbol{v} + \mathrm{vec}(\boldsymbol{W}^\top \boldsymbol{U} \boldsymbol{G}^\top)$

---

Let $\boldsymbol{y}^{(k)}$ be the intermediate variable, the procedure of applying PR splitting on Equation (19) can be summarized as first finding the fixed-point $\boldsymbol{y}^*$ of the following iteration function

$$\boldsymbol{y}^{(k+1)} := B_\alpha^{\mathrm{PR}}(\boldsymbol{y}^{(k)}) = 2\boldsymbol{V}^\top \left( 2(\boldsymbol{I} + \alpha\boldsymbol{D})^{-1}(\boldsymbol{y}^{(k)} + \alpha\boldsymbol{v}) - \boldsymbol{y}^{(k)} \right) - 2(\boldsymbol{I} + \alpha\boldsymbol{D})^{-1}(\boldsymbol{y}^{(k)} + \alpha\boldsymbol{v}) + \boldsymbol{y}^{(k)} \tag{21}$$

and then the final solution of the operator splitting problem is $\tilde{\boldsymbol{u}} = (\boldsymbol{I} + \alpha\boldsymbol{D})^{-1}(\boldsymbol{y}^* + \alpha\boldsymbol{v})$.

## F.3 ANDERSON ACCELERATION

We first introduce the general Anderson acceleration scheme. Let $f : \mathbb{R}^n \to \mathbb{R}^n$ be a function s.t. the Lipschitz constant $L(f) < 1$. Therefore, the function $f$ admits a unique fixed point and can be obtained through Picard iteration. Let $h(\boldsymbol{x}) = f(\boldsymbol{x}) - \boldsymbol{x}$ be the residual function. Let $\boldsymbol{x}^{(0)}$ be the initial guess, $\beta \in (0, 1)$ be a relaxation parameter, and $m > 1$ be an integer parameter. Then the Anderson acceleration update $\boldsymbol{x}^{(k)}$ as

$$\boldsymbol{x}^{(k+1)} = (1 - \beta) \sum_{i=0}^{m} \gamma_i^{(k)} \boldsymbol{x}^{(k-m+i)} + \beta \sum_{i=0}^{m} \gamma_i^{(k)} h\left( \boldsymbol{x}^{(k-m+i)} \right) \tag{22}$$

where the coefficients $\boldsymbol{\gamma}^{(k)} = \left( \gamma_0^{(k)}, \dots, \gamma_m^{(k)} \right)^\top$ are determined by a least-square problem as the following:

$$\min_{\boldsymbol{\gamma}=(\gamma_0,\dots,\gamma_m)^\top} \left\| \sum_{i}^{m} h(\boldsymbol{x}^{(k-m+i)})\gamma_i \right\| \text{ s.t.} \sum_{i=0}^{m} \gamma_i = 1.$$

Note that, when $\beta = 1$, the trivial weight $\boldsymbol{\gamma}^{(k)} = (0, \dots, 0, 1)^\top$ recovers Picard iteration. Therefore, when the Picard iteration converges, the Anderson acceleration also converges and typically faster.

In Algorithm 5, we present the FB MIGNN forward propagation with Anderson acceleration on the FB iteration function $F_\alpha^{\mathrm{FB}}$ which is introduced in Section 4 and recalled here:

$$\boldsymbol{Z}^{(k+1)} := F_\alpha^{\mathrm{FB}}(\boldsymbol{Z}^{(k)}) := \mathrm{prox}_f^\alpha \left( \boldsymbol{Z}^{(k)} - \alpha \cdot \left( \boldsymbol{Z}^{(k)} - \boldsymbol{W}\boldsymbol{Z}^{(k)}\boldsymbol{G} - g_B(\boldsymbol{X}) \right) \right).$$

---

**Algorithm 5** MIGNN-FB-Forward: FB MIGNN forward propagation

---

**Input:** initial point $\boldsymbol{Z}^{(0)} := \boldsymbol{0}$, FB damping parameter $\alpha$, AA relaxation parameter $\beta$, max storage size $m \geq 1$.
Compute $\boldsymbol{F}^{(0)} = F_\alpha^{\mathrm{PB}}(\boldsymbol{Z}^{(0)})$, $\boldsymbol{H}^{(0)} = \boldsymbol{F}^{(0)} - \boldsymbol{Z}^{(0)}$.
**for** $k = 1, \dots, K$ **do**
    Set $m_k = \min(m, k)$
    Compute $\boldsymbol{F}^{(k)} = F_\alpha^{\mathrm{PB}}\left( \boldsymbol{Z}^{(k)} \right)$, $\boldsymbol{H}^{(k)} = \boldsymbol{F}^{(k)} - \boldsymbol{Z}^{(k)}$
    Update $\boldsymbol{H} := (\boldsymbol{H}^{(k-m_k)}, \dots, \boldsymbol{H}^{(k)})$
    Determine $\boldsymbol{\gamma}^{(k)} = \left( \gamma_0^{(k)}, \dots, \gamma_{m_k}^{(k)} \right)^\top$ that solves

$$\min_{\boldsymbol{\gamma}=\left(\gamma_0,\dots,\gamma_{m_k}\right)^\top} \|\boldsymbol{H}\boldsymbol{\gamma}\| \text{ s.t.} \sum_{i=0}^{m_k} \gamma_i = 1.$$

    Set

$$\boldsymbol{Z}^{(k+1)} := \beta \sum_{i=0}^{m_k} \gamma_i^{(k)} F_\alpha^{\mathrm{PB}}(\boldsymbol{Z}^{((k-m_k)+i)}) + (1 - \beta) \sum_{i=0}^{m_k} \gamma_i^{(k)} \boldsymbol{Z}^{((k-m_k)+i)}.$$

**end for**
**return** $\boldsymbol{Z}^{(k+1)}$

---

In Algorithm 6, we present the PR MIGNN forward propagation with Anderson acceleration on the PR iteration function $F_\alpha^{\mathrm{PR}}$ which is introduced in Section 4 and recalled here:

$$\boldsymbol{u}^{(k+1)} := F_\alpha^{\mathrm{PR}}(\boldsymbol{u}^{(k)}) = 2\boldsymbol{V}\left( 2\,\mathrm{prox}_f^\alpha(\boldsymbol{u}^{(k)}) - \boldsymbol{u}^{(k)} + \alpha\,\mathrm{vec}(g_B(\boldsymbol{X})) \right) - 2\,\mathrm{prox}_f^\alpha(\boldsymbol{u}^{(k)}) + \boldsymbol{u}^{(k)}, \tag{23}$$

---

**Algorithm 6** MIGNN-PR-forward: PR MIGNN forward propagation

---

**Input:** initial point $\boldsymbol{u}^{(0)} = \mathrm{vec}(\boldsymbol{U}^{(0)}) := \boldsymbol{0}$, PR damping parameter $\alpha$, AA relaxation parameter $\beta$, max storage size $m \geq 1$.
Compute $\boldsymbol{f}^{(0)} := F_\alpha^{\mathrm{PR}}(\boldsymbol{u}^{(0)}), \boldsymbol{h}^{(0)} := \boldsymbol{f}^{(0)} - \boldsymbol{u}^{(0)}$.
**for** $k = 1, \ldots, K$ **do**
    Set $m_k := \min(m, k)$
    Compute $\boldsymbol{f}^{(k)} := F_\alpha^{\mathrm{PR}}(\boldsymbol{u}^{(k)}), \boldsymbol{h}^{(k)} := \boldsymbol{f}^{(k)} - \boldsymbol{u}^{(k)}$
    Update $\boldsymbol{H} := (\boldsymbol{h}^{(k-m_k)}, \ldots, \boldsymbol{h}^{(k)})$
    Determine $\boldsymbol{\gamma}^{(k)} = \left(\gamma_0^{(k)}, \ldots, \gamma_{m_k}^{(k)}\right)^\top$ that solves

$$\min_{\boldsymbol{\gamma} = \left(\gamma_0, \ldots, \gamma_{m_k}\right)^\top} \|\boldsymbol{H}\boldsymbol{\gamma}\| \text{ s.t. } \sum_{i=0}^{m_k} \gamma_i = 1.$$

    Set

$$\boldsymbol{u}^{(k+1)} := \beta \sum_{i=0}^{m_k} \gamma_i^{(k)} F_\alpha^{\mathrm{PR}}(\boldsymbol{u}^{((k-m_k)+i)}) + (1-\beta) \sum_{i=0}^{m_k} \gamma_i^{(k)} \boldsymbol{u}^{((k-m_k)+i)}.$$

**end for**

Set $\boldsymbol{U}^{(k+1)} := \mathrm{vec}^{-1}(\boldsymbol{u}^{(k+1)})$
**return** $\mathrm{prox}_f^\alpha(\boldsymbol{U}^{(k+1)})$

---

The FB iteration function for the backpropagation $B_\alpha^{\mathrm{FB}}$ is introduced in Appendix F.2 and recalled here:

$$\boldsymbol{u}^{(k+1)} := B_\alpha^{\mathrm{FB}}(\boldsymbol{u}^{(k)}) = (\boldsymbol{I} + \alpha\boldsymbol{D})^{-1}((1-\alpha)\boldsymbol{u}^{(k)} + \alpha\boldsymbol{W}^\top\boldsymbol{v}). \tag{24}$$

We now present the Anderson-accelerated FB MIGNN backward propagation as Algorithm 7.

---

**Algorithm 7** MIGNN-FB-Backward: FB MIGNN backward propagation

---

**Input:** initial point $\boldsymbol{u}^{(0)} := \mathrm{vec}(\boldsymbol{U}) := \boldsymbol{0}$, $\boldsymbol{v} := \frac{\partial \ell}{\partial \mathrm{vec}(\boldsymbol{Z}^*)}$, PR damping parameter $\alpha$, AA relaxation parameter $\beta$, max storage size $m \geq 1$.
Compute $\boldsymbol{f}^{(0)} := B_\alpha^{\mathrm{FB}}(\boldsymbol{u}^{(0)}), \boldsymbol{h}^{(0)} := \boldsymbol{f}^{(0)} - \boldsymbol{u}^{(0)}$.
**for** $k = 1, \ldots, K$ **do**
    Set $m_k := \min(m, k)$
    Compute $\boldsymbol{f}^{(k)} := B_\alpha^{\mathrm{FB}}(\boldsymbol{u}^{(k)}), \boldsymbol{h}^{(k)} := \boldsymbol{f}^{(k)} - \boldsymbol{u}^{(k)}$
    Update $\boldsymbol{H} := (\boldsymbol{h}^{(k-m_k)}, \ldots, \boldsymbol{h}^{(k)})$
    Determine $\boldsymbol{\gamma}^{(k)} = \left(\gamma_0^{(k)}, \ldots, \gamma_{m_k}^{(k)}\right)^\top$ that solves

$$\min_{\boldsymbol{\gamma} = \left(\gamma_0, \ldots, \gamma_{m_k}\right)^\top} \|\boldsymbol{H}\boldsymbol{\gamma}\| \text{ s.t. } \sum_{i=0}^{m_k} \gamma_i = 1.$$

    Set

$$\boldsymbol{u}^{(k+1)} := \beta \sum_{i=0}^{m_k} \gamma_i^{(k)} B_\alpha^{\mathrm{FB}}(\boldsymbol{u}^{((k-m_k)+i)}) + (1-\beta) \sum_{i=0}^{m_k} \gamma_i^{(k)} \boldsymbol{u}^{((k-m_k)+i)}.$$

**end for**

Set $\boldsymbol{U}^{(k+1)} := \mathrm{vec}^{-1}(\boldsymbol{u}^{(k+1)})$
**return** $\boldsymbol{v} + \mathrm{vec}(\boldsymbol{W}^\top\boldsymbol{U}^{(k+1)}\boldsymbol{G}^\top)$

---

The PR iteration function for the backpropagation $B_\alpha^{\mathrm{PR}}$ is introduced in Appendix F.2 and recalled here: let $\boldsymbol{y}^{(k)}$ be the intermediate variable,

$$\boldsymbol{y}^{(k+1)} := B_\alpha^{\mathrm{PR}}(\boldsymbol{y}^{(k)}) = 2\boldsymbol{V}^\top \left( 2(\boldsymbol{I} + \alpha\boldsymbol{D})^{-1}(\boldsymbol{y}^{(k)} + \alpha\boldsymbol{v}) - \boldsymbol{y}^{(k)} \right) - 2(\boldsymbol{I} + \alpha\boldsymbol{D})^{-1}(\boldsymbol{y}^{(k)} + \alpha\boldsymbol{v}) + \boldsymbol{y}^{(k)} \quad (25)$$

and then the final solution of the operator splitting problem is $\tilde{u} = (\boldsymbol{I} + \alpha\boldsymbol{D})^{-1}(\boldsymbol{y}^* + \alpha\boldsymbol{v})$. We now present the Anderson-accelerated PR MIGNN backward propagation as Algorithm 8.

---

**Algorithm 8** MIGNN-PR-Backward: PR MIGNN backward propagation

---

**Input:** initial point $\boldsymbol{y}^{(0)} := \boldsymbol{0}$, $\boldsymbol{v} := \frac{\partial\ell}{\partial\mathrm{vec}(\boldsymbol{Z}^*)}$, PR damping parameter $\alpha$, AA relaxation parameter $\beta$, max storage size $m \geq 1$.

Compute $\boldsymbol{f}^{(0)} := B_\alpha^{\mathrm{PR}}(\boldsymbol{y}^{(0)})$, $\boldsymbol{h}^{(0)} := \boldsymbol{f}^{(0)} - \boldsymbol{y}^{(0)}$.

**for** $k = 1, \ldots, K$ **do**

    Set $m_k := \min(m, k)$

    Compute $\boldsymbol{f}^{(k)} := B_\alpha^{\mathrm{PR}}\left(\boldsymbol{y}^{(k)}\right)$, $\boldsymbol{h}^{(k)} := \boldsymbol{f}^{(k)} - \boldsymbol{y}^{(k)}$

    Update $\boldsymbol{H} := (\boldsymbol{h}^{(k-m_k)}, \ldots, \boldsymbol{h}^{(k)})$

    Determine $\boldsymbol{\gamma}^{(k)} = \left(\gamma_0^{(k)}, \ldots, \gamma_{m_k}^{(k)}\right)^\top$ that solves

$$\min_{\boldsymbol{\gamma}=\left(\gamma_0,\ldots,\gamma_{m_k}\right)^\top} \|\boldsymbol{H}\boldsymbol{\gamma}\| \text{ s.t. } \sum_{i=0}^{m_k} \gamma_i = 1.$$

    Set

$$\boldsymbol{y}^{(k+1)} := \beta \sum_{i=0}^{m_k} \gamma_i^{(k)} B_\alpha^{\mathrm{PR}}(\boldsymbol{y}^{((k-m_k)+i)}) + (1-\beta)\sum_{i=0}^{m_k} \gamma_i^{(k)} \boldsymbol{y}^{((k-m_k)+i)}.$$

**end for**

Compute $\boldsymbol{u}^{(k+1)}$ where $u_i^{(k+1)} := \begin{cases} \frac{y_i^{(k+1)} + \alpha v_i}{1 + \alpha(1 + D_{ii})} & \text{if } D_{ii} < \infty \\ 0 & \text{if } D_{ii} = \infty \end{cases}$

Set $\boldsymbol{U}^{(k+1)} := \mathrm{vec}^{-1}(\boldsymbol{u}^{(k+1)})$

**return** $\boldsymbol{v} + \mathrm{vec}(\boldsymbol{W}^\top \boldsymbol{U}^{(k+1)} \boldsymbol{G}^\top)$

---

## G  EFFECTS OF THE ORDER OF NEUMANN SERIES EXPANSION

In this section, we perform ablation studies on the effects of the order of the Neumann series for approximating matrix $(\boldsymbol{I} + \alpha(\boldsymbol{I} - \boldsymbol{G}^\top \otimes \boldsymbol{W}))^{-1}$ in MIGNN-N$K$D$P$ with fixed $P = 1$. We study the performance of MIGNN-N$K$D$P$ for synthetic directed chain classification, benchmark graph node and graph classification.

### G.1  DIRECTED CHAIN CLASSIFICATION

Examining the Neumann series expansion for the synthetic chain classification task demonstrates the trade-off between accuracy and time complexity. We train MIGNN-N$K$D1 for three-class classification, where the order $K$ ranges from 1 to 5 in increments of 1. Fig. 9 plots the resulting test accuracy, number of iterations, and time elapsed for each training epoch.

We make three observations as the order of the Neumann series increases. First the accuracy increases with respect to the order with diminishing returns. Second the number of iterations increases relative to the order up 3. Finally the time elapsed also increases with respect to the order up to 4 and 5 which are similar. These observations underscore the trade-off between accuracy and time complexity as the order increases.

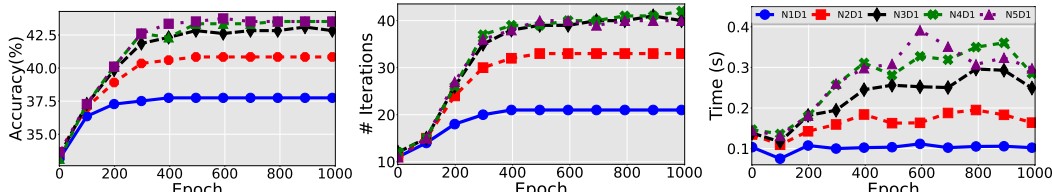

Figure 9: Comparison of Neumann expansion for accuracy, number of iterations, and elapsed time using three-class chain classifications with chain length 140.

## G.2 NODE CLASSIFICATION

The graph node classification tasks also highlight the trade-off between accuracy and time complexity. We train MIGNN-N$K$D1 using 10-fold cross validation on Cora, Citeseer and Pubmed. We consider $K$ in the range from 1 to 5, incrementing by 1. The mean test accuracy and time elapsed along with their standard deviations are reported in Table 3.

| Datasets | Cora (Accuracy) | Cora (Time) | Citeseer (Accuracy) | Citeseer (Time) | Pubmed (Accuracy) | Pubmed (Time) |
|---|---|---|---|---|---|---|
| MIGNN-N1D1 | $86.7 \pm 1.81$ | $0.384 \pm 0.036$ | $69.8 \pm 6.8$ | $0.149 \pm 0.022$ | $80.9 \pm 3.97$ | $0.151 \pm 0.015$ |
| MIGNN-N2D1 | $86.8 \pm 1.37$ | $0.467 \pm 0.039$ | $73.2 \pm 5.1$ | $0.203 \pm 0.022$ | $83.1 \pm 0.66$ | $0.184 \pm 0.016$ |
| MIGNN-N3D1 | $86.8 \pm 1.55$ | $0.514 \pm 0.021$ | $73.6 \pm 5.2$ | $0.242 \pm 0.025$ | $83.3 \pm 0.76$ | $0.216 \pm 0.017$ |
| MIGNN-N4D1 | $86.7 \pm 1.40$ | $0.622 \pm 0.055$ | $74.8 \pm 2.3$ | $0.261 \pm 0.025$ | $83.6 \pm 0.67$ | $0.241 \pm 0.020$ |
| MIGNN-N5D1 | $87.0 \pm 1.42$ | $0.698 \pm 0.064$ | $74.9 \pm 2.3$ | $0.292 \pm 0.015$ | $83.6 \pm 0.66$ | $0.272 \pm 0.024$ |

Table 3: Graph node classification mean accuracy (%) $\pm$ standard deviation for 10-fold cross-validation.

For node classification we see a very clear trend across all datasets. Both the accuracy and time elapsed increase with the order of the Neumann expansion. However, the accuracy scales with diminishing returns; notice N4 and N5 have the same accuracy for both Citeseer and Pubmed.

## G.3 GRAPH CLASSIFICATION

In this subsection, we apply MIGNN-N$K$D1 to classify the MUTAG dataset, where $K$ ranges from 1 to 5 incrementing by 1. Fig. 10 plots the test accuracy, the number of iterations, and the time elapsed for training one fold of the 10-fold cross validation.

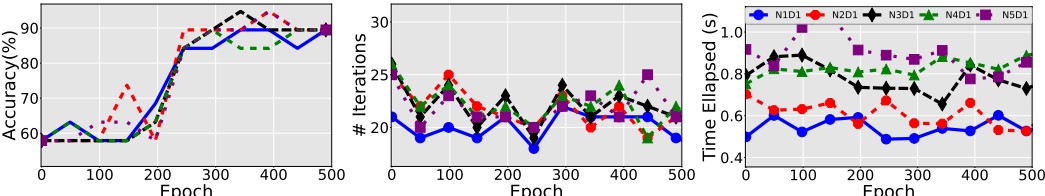

Figure 10: Comparison of Neumann expansion for accuracy, number of iterations and elapsed time using the first fold of the MUTAG graph data set.

Unlike the directed chains and node classification tasks, the graph classification does not show significant improvements from higher order Neumann expansion on this fold. However, from Table 2, we observe that over 10-fold cross validation diffusion improves the results. Although the accuracy and iteration count remain similar among all orders, the time elapsed still scales with the order.

## H EFFECTS OF THE ORDER OF GRAPH DIFFUSION CONVOLUTION

In this section, we use MIGNN-N$K$D$P$ with fixed $K = 1$ and varying order of graph diffusion $P$ to study the effects of the order of graph diffusion convolution. We report the performance of MIGNN benchmarking on synthetic directed chain classification, benchmark graph node and graph classification tasks.

### H.1 DIRECTED CHAIN CLASSIFICATION

The three-class chain classification task benefits tremendously for high orders of diffusion. We train MIGNN-N1D$P$ on chain lengths of 140, where $P$ ranges from 1 to 5 incrementing by 1. Fig. 11 plots the test accuracy, number of iterations, and time elapsed for each training epoch.

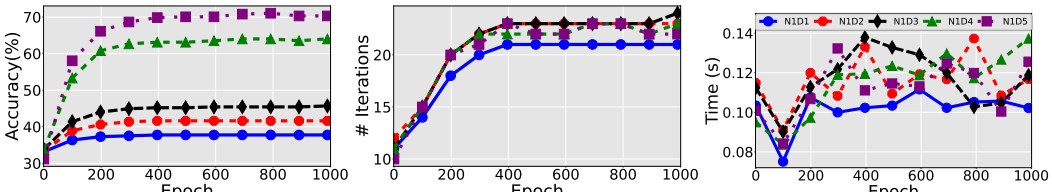

Figure 11: Comparison of graph diffusion convolution for accuracy, number of iterations and elapsed time using three-class chains of length 140.

For diffusion convolution we make two observations. First, the accuracy scales with the order of diffusion with a remarkable gap between D3 and D4. Second, the iteration count and time elapsed remain relatively constant among all orders, with D1 standing out as the least among all others.

Our theory informs us of the following: 1) Accuracy scaling occurs when the introduced $P$-hop edges contain relevant information for the task 2) Time elapsed scales relative to the number of edges in the higher order graph diffusion matrix. Our observations support our theory and strongly suggest using diffusion as an inexpensive improvement to simple learning tasks.

## H.2 NODE CLASSIFICATION

In this subsection, we study the effects of the order of diffusion convolution on the node classification tasks outlined in the citation datasets (Cora, Citeseer, Pubmed). We consider MIGNN-N1D$P$ with $P$ ranging from 1 to 3 with an increment of 1. Table 4 reports the test accuracy and the time elapsed for each epoch for different MIGNN models. We observe that diffusion does provide any benefit for graph node classification.

| Datasets | Cora (Accuracy) | Cora (Time) | Citeseer (Accuracy) | Citeseer (Time) | Pubmed (Accuracy) | Pubmed (Time) |
|---|---|---|---|---|---|---|
| MIGNN-N1D1 | $86.7 \pm 1.81$ | $0.384 \pm 0.036$ | $69.8 \pm 6.8$ | $0.141 \pm 0.017$ | $80.9 \pm 3.97$ | $0.151 \pm 0.015$ |
| MIGNN-N1D2 | $86.5 \pm 1.30$ | $0.367 \pm 0.032$ | $69.3 \pm 6.5$ | $0.146 \pm 0.021$ | $77.6 \pm 4.82$ | $0.410 \pm 0.040$ |
| MIGNN-N1D3 | $83.7 \pm 1.33$ | $0.766 \pm 0.021$ | $68.0 \pm 7.0$ | $0.164 \pm 0.057$ | $83.3 \pm 0.76$ | $6.25 \pm 1.14$ |

Table 4: Graph node classification mean accuracy (%) $\pm$ standard deviation for 10-fold cross-validation.

## H.3 GRAPH CLASSIFICATION

We further apply MIGNN-N1D$P$ to classify the MUTAG dataset, where $P$ ranges from 1 to 5 incrementing by 1. Fig. 12 plots the test accuracy, number of iterations, and time elapsed for each epoch for different MIGNN models.

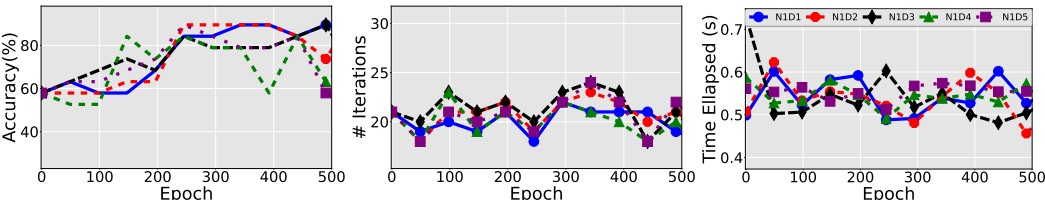

Figure 12: Comparison of diffusion convolution for accuracy, number of iterations and elapsed time using the first fold of the MUTAG graph data set.

We observe that higher order diffusion convolution has little impact on the time complexity when each connected subgraph is small relative to the underlying graph.

# I   MORE DISCUSSION ON WHEN IGNNS BECOME EXPRESSIVE FOR LEARNING LRD

In this section, we further confirm the interconnection between the accuracy of IGNN for classifying directed chains and the eigenvalues of $|\boldsymbol{W}|$. The accuracy and number of iterations of IGNN and the dynamics of the two leading eigenvalues are plotted in Figs. 13 and 14, respectively, for the binary and three-class cases. These results confirm the phenomena we have discussed in Sec. 1.

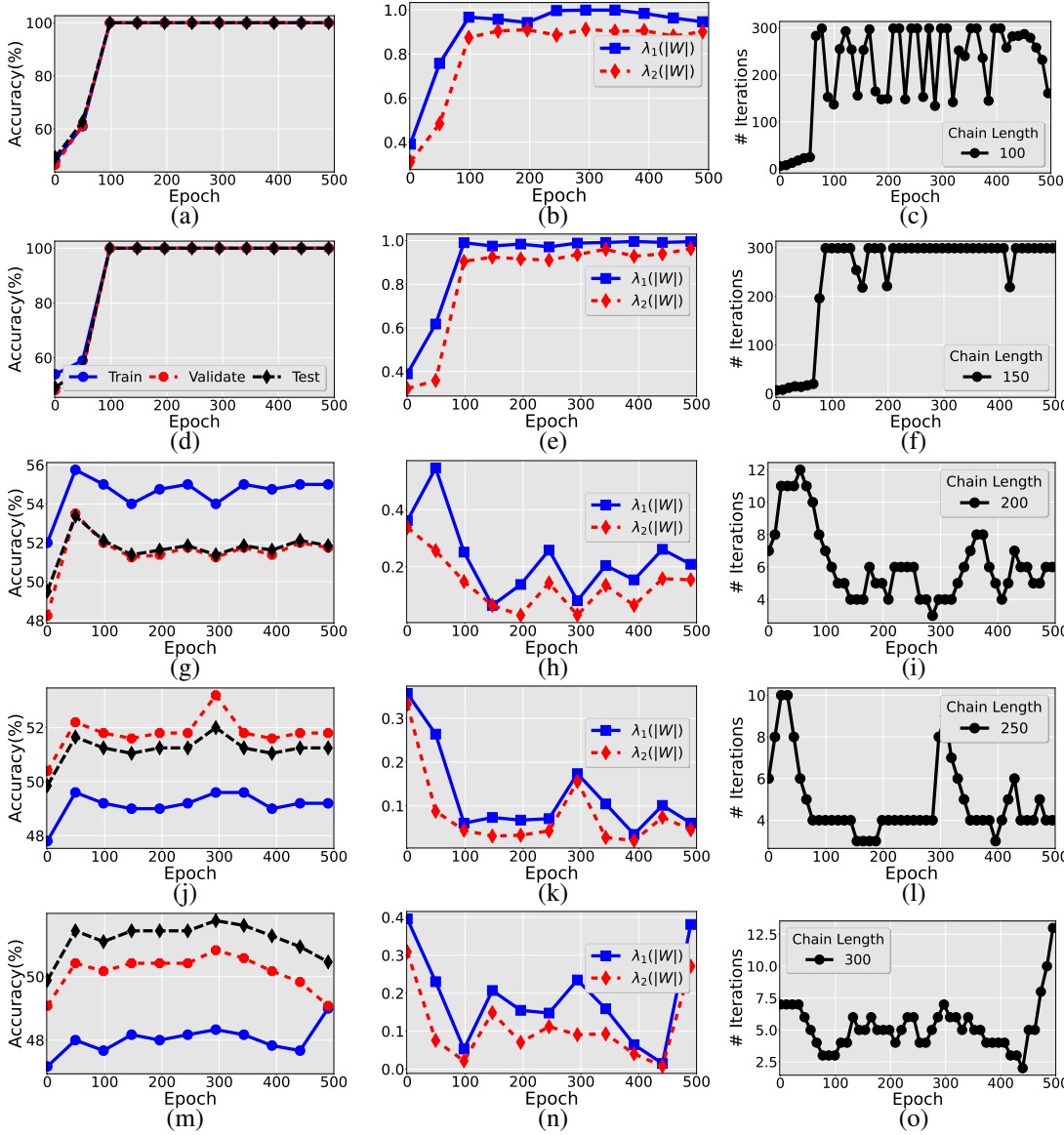

Figure 13: In the first column, the training, test, and validation accuracies of IGNN are depicted for several varying chain lengths. In the second column, the corresponding top two eigenvalues are plotted. The third column depicts the number of Picard iterations for each chain length. When IGNN becomes accurate for chain classification, the corresponding $\lambda_1(|\boldsymbol{W}|)$ becomes close to 1 and requires substantially more iterations for the Picard iteration to converge.

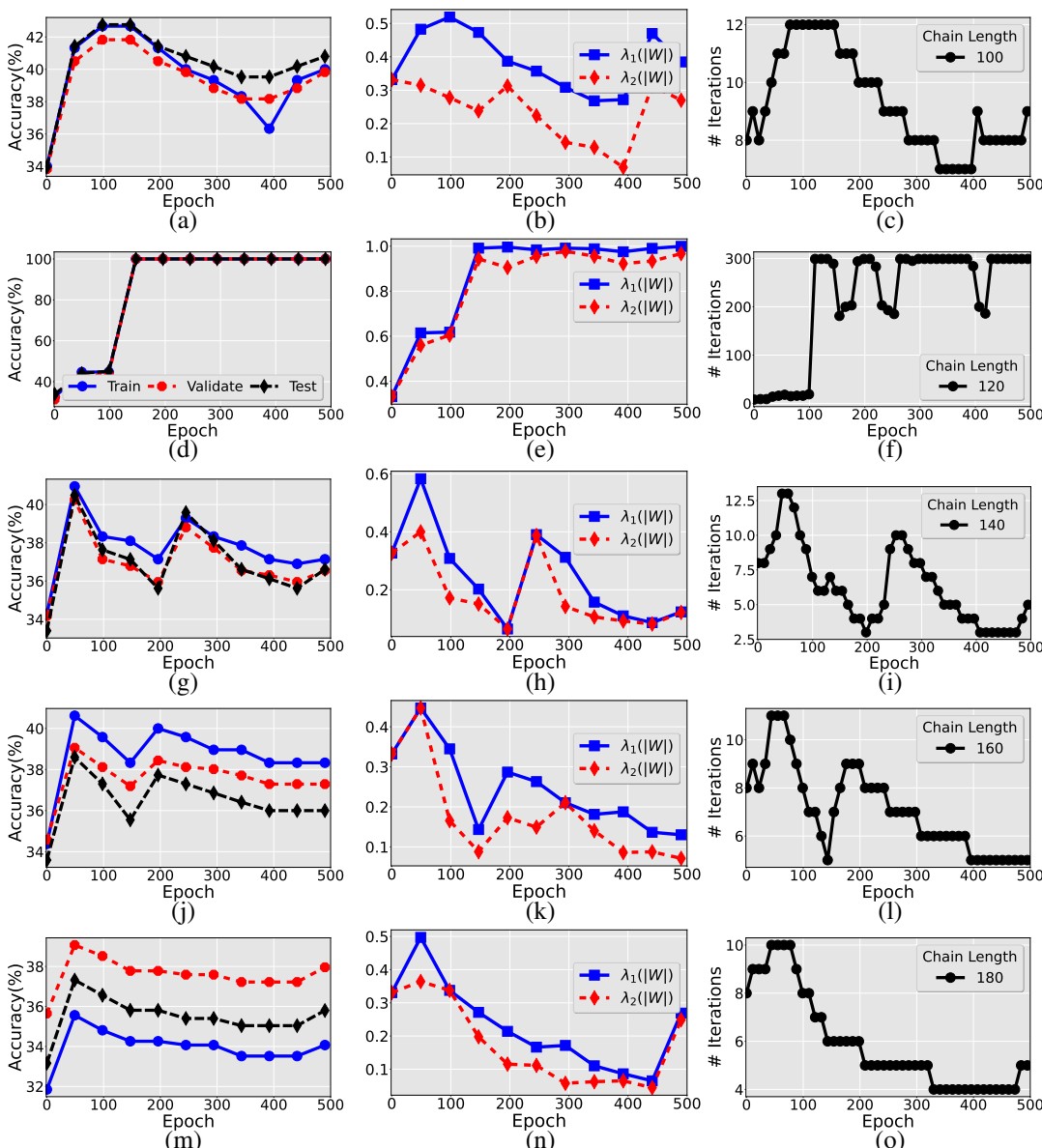

Figure 14: The first column shows the training, test, and validation accuracies of IGNN for several chain lengths of three classes. In the second column, we plot the corresponding top two eigenvalues. In the third column, we plot the number of Picard iterations for each chain length. As the maximum eigenvalue of the system approaches 1, IGNN becomes more accurate for chain classification at the cost of a significantly increased number of training iterations.

## J DETAILS ABOUT DATASETS

**Synthetic chains dataset.** To evaluate the LRD learning ability of models, we construct synthetic chains dataset as in Gu et al. [39]. Both binary classification and multiclass classification are considered. Let $c$ be the number of classes, that is, there are $c$ types of chains. The label information is only encoded as a one-hot vector in the first $c$-dimensions of the node feature of the starting nodes of each chain. With $c$ classes, $n_c$ chains for each class, and $l$ nodes in each chain, the chain dataset has $c \times n_c \times l$ nodes in total.

**Bioinformatics datasets.** MUTAG is a dataset of 188 mutagenic aromatic and heteroaromatic nitro compounds. PTC is a dataset of 344 chemical compounds that report carcinogenicity for male

and female rats. COX2 is a dataset of 467 cyclooxygenase-2 (COX-2) inhibitors. PROTEINS is a dataset of 1113 secondary structure elements (SSEs). NCI1 is a public dataset from the National Cancer Institute (NCI) and is a subset of balanced datasets of chemical compounds screened for the ability to suppress or inhibit the growth of a panel of human tumor cell lines.

**Amazon product co-purchasing network.** This dataset contains 334863 nodes (representing goods), 925872 edges, and 58 label types. An edge is formed between two nodes if the represented goods have been purchased together [52].

**Pore networks.** The pore network is a simulated dataset that models fluid flow in porous media. Each porous network is randomly generated inside a cubic domain of width 0.1m by Delaunay or Voronoi tessellation. The prediction of equilibrium pressure in a pore network under physical diffusion is introduced as a GNN task in [63]. The GNN model prediction accuracy is compared with the ground truth obtained through solving the diffusion equation directly, see [63, Appendix C] for more details.

**Citation dataset.** Cora and Citeseer are large citation datasets that describe the presence of specific words in publications. Pubmed is a large citation dataset that contains information about papers classified for studying one of the three diabetes. The following table adapted from [33] describes the statistics of the three datasets.

| Dataset | Type | Classes | Features | Nodes | Edges | Label rate | Avg. SP |
|---|---|---|---|---|---|---|---|
| Cora | Citation | 7 | 2879 | 2810 | 7981 | 0.047 | 5.27 |
| Citeseer | Citation | 6 | 3703 | 2110 | 3668 | 0.036 | 9.31 |
| Pubmed | Citation | 3 | 500 | 19717 | 44324 | 0.003 | 6.34 |

Table 5: Dataset statistics. The shortest path length is denoted by Avg. SP.

## K    DETAILS ABOUT HYPERPARAMETERS

The default parameter settings for MIGNN are the following. For the fixed-point schemes $\alpha = 0.9, \beta = 0.9$, the default maximum iteration is 300, the tolerance is $1e\text{-}6$, and convergence is measured in the $\ell_\infty$-norm of the difference between two consecutive fixed point iterations. The learnable parameter is initialized to $\gamma = 1.0$.

**Synthetic chains dataset.** For both binary and three-class classification we use the parameters outlined by IGNN [71]. In both classification tasks we make the same modifications. We set the clipping and dropout to 0.

**Citation dataset.** In the citation datasets we follow the training procedure used by GIND [22]. For the Cora dataset we set the weighted-decay to $1e\text{-}4$ and the fixed-point tolerance to $1e\text{-}3$. For all three models we set the fixed-point $\alpha = 0.5$, and the number of hidden layers to 64.

**Bioinformatics datasets.** In the bioinformatics datasets we follow the training procedure used by IGNN [71]. On the C12 dataset for MIGNN-Mon, we use $\alpha = 0.5$. On all other datasets we extend the number of training epochs to 500.

**Amazon product co-purchasing network.** In the Amazon product co-purchasing dataset we follow the training procedure used by IGNN [71].

**Pore networks.** In the physical diffusion pore networks we follow the training procedure used by CGS [63] and the default parameter values for MIGNN.

