# OpenReview forum: "Stable, Efficient, and Flexible Monotone Operator Implicit Graph Neural Networks"
_ICLR.cc/2023/Conference — Submitted to ICLR 2023_

### Official Review · Reviewer_Ti7k · 2022-10-23

**Confidence:** 4
**Correctness:** 3
**Technical Novelty And Significance:** 3
**Empirical Novelty And Significance:** 2
**Recommendation:** 6

**Clarity, Quality, Novelty And Reproducibility:**

Clarity: Good though the content is dense. To understand many things, one needs to be very familiar with monotone operators and check appendix.

Quality: Good. The technical contributions are great. The experiments make sense while are not as strong as their technical parts. The used datasets are small. The ablation studies are not sufficient.

Novelty: Good. The first work to build the connection between monotone operators and GNN modeling.

Reproducibility: Good.

**Strength And Weaknesses:**

Strengths
1. This work is strong in its math-grounded modeling technique. In particular, it connects GNN modeling with monotone operator theory, which is exciting. Although such connection has been leveraged to study deep equilibrium model [1], I think it is still novel to apply such connection to study GNN models.

2. The derivation and the discussion are super extensive, covering different types of operators to match different properties, different operator splitting iterative methods, acceleration ideas, back-propagation ideas.

3. The experiments make sense and show the benefits of the model.

Weaknesses
1. The main concern of this paper is about its exposition. As it tries to discuss too many things, many details in the main text are missed and I have to check the appendix even if I am fairly familiar with monotone operator theory. Some discussions may be not necessary, kind of detour, and may distract the readers.

For example, in Sec. 2, the authors advertise a lot about the new well-posed condition, where they claim that Eq. (2) is well posed even if $W$ has eigenvalue less than -1. However, this result is not very practically useful, because in sec. 3, their finally adopted $W$ still have to have eigenvalues with absolute values less than 1 to guarantee convergence. Moreover, in Remark 2, the way to make W symmetric will naturally remove the asymmetric part of W in the monotone operator ($F- F^T$).

2. Some ablation studies are missing. For example, the authors also adopt diffusion convolution in Eq.(12), where the diffusion convolution automatically has the benefit in capturing long range dependence. So, it is unclear that in the node classification tasks, whether the benefit comes from diffusion convolution or implicit operators. Here,  I also do not know what "N3D5", "N5D5" used in the model name in the experiment means, which are not defined. I assume they are related to the diffusion convolution. Please clarify. Also, provide the study of using different convolutions to show the benefits indeed come from implicit modeling.

3. Node-level experiments are too few. Actually, Long range dependence is not that crucial for graph-level tasks, because one can adopt transformer models and run on dense graphs by adding jump-hop edges. This is also why the proposed models are hard to outperform many baselines in graph-level tasks (the used baselines and datasets in this work are actually not SOTA and large). LRD is more important for node-level tasks. However, this work only provides one dataset for such evaluation.



**Summary Of The Paper:**

This work proposes to use monotone and orthogonal operator to build implicit graph neural network. The adopted approach is well build upon monotone operator theory. The new operators allows more stable (via orthogonal operator) or expressive (via monotone operator) implicit GNN models with provable convergence. The discussion is super extensive. The experiments on node-level and graph-level tasks show the superiority of the method.

**Summary Of The Review:**

The paper is strong in its technical contributions. The empirical contributions are about the acceptance bar. More extensive studies are suggested. The exposition of main text can be improved by showing the results more directly instead of detouring the discussion.

---

> ### Author Response · Authors · 2022-11-11
> **Response to Reviewer Ti7k**
>
> Thank you for your thoughtful review and valuable feedback. We are glad to hear that the reviewer found our idea interesting. We kindly invite the reviewer to read our general response. In what follows, we provide point-by-point responses to the raised comments.
>
> ---
>
> **Q1. The main concern of this paper is about its exposition. As it tries to discuss too many things, many details in the main text are missed and I have to check the appendix even if I am fairly familiar with monotone operator theory. Some discussions may be not necessary, kind of detour, and may distract the readers. For example, in Sec. 2, the authors advertise a lot about the new well-posed condition, where they claim that Eq. (2) is well-posed even if W has eigenvalue less than -1. However, this result is not very practically useful, because in sec. 3, their finally adopted W still have to have eigenvalues with absolute values less than 1 to guarantee convergence.**
>
> **Reply:** Thank you for pointing out the exposition issue to us. We have substantially revised our paper to make it easier to read. We realized that one major issue of our writing causing confusion is that the purpose and detailed implementation of MIGNN with the two parameterizations are not very clear. In particular, MIGNN with monotone parameterization. Please allow us to clarify this below briefly.
> We observe that the standard IGNN using Picard iteration has two bottlenecks: **1)** To learn long-range dependencies (LRD), IGNN has to push the magnitude of some eigenvalues of the weight matrix $\mathbf{W}$ towards 1, resulting in **slow convergence** of Picard iteration. Also, during the training of IGNN, the magnitude of eigenvalues of $\mathbf{W}$ (starting from random Gaussian initialization) may not converge to 1, resulting in **instability** in learning LRD. **2)** The parameterization used in IGNN constrains the magnitude of eigenvalues of $\mathbf{W}$ to be less than $1$, **limiting the expressivity** of IGNN.
>
> To address the above two issues of IGNN, we propose MIGNN with orthogonal and monotone parameterizations. **1)** We propose MIGNN with orthogonal parameterization to enhance the stability and efficiency of learning LRD --- as the magnitude of eigenvalues of orthogonal parameterization stably converges to one. **2)** We further propose monotone parameterization for MIGNN to improve the expressivity for graph learning. Notice that the magnitude of **the eigenvalues of $\mathbf{W}$ in IGNN has to be in the range of $[0,1)$** to guarantee the well-posedness of IGNN. In contrast, **MIGNN with monotone parameterization allows the real part of eigenvalues of $\mathbf{W}$ to be in the range $(-\infty,1)$ and the imaginary part to be arbitrary, which is much wider than that of IGNN**.
>
> Next, let us explain why we apply the Anderson-accelerated operator splitting schemes to find the fixed point of MIGNN with the two new parameterizations. This is because both parameterizations are **not in the efficient convergence regime of Picard iteration**. We have to apply other algorithms to find the fixed point of MIGNN with orthogonal or monotone parameterization. Particularly appealing algorithms are based on Anderson-accelerated operator splitting schemes. In particular, we apply the Anderson accelerated forward-backward operator splitting (FB) to find the fixed point of MIGNN with monotone parameterization, resulting in the model **MIGNN-Mon**. There are three popular operator splitting schemes: FB, Peaceman-Rachford (PR), and Douglas-Rachford (DR). PR and DR. PR and DR require inverting large matrices, which is inefficient and unscalable. Moreover, we cannot use the Neumann series to approximate the inversion of the matrices when the monotone parameterization is used since the parameterization has eigenvalues whose real part can be much less than $-1$. As such, we select FB to find the fixed point of MIGNN with monotone parameterization.
>
> Let us further explain why we use Anderson accelerated PR, with approximations, to find the fixed point of MIGNN with orthogonal parameterization. Among the three choices, PR is better than DR, as explained in Remark 2 in the revision. FB requires stronger conditions for convergence guarantee and converges slower than PR in general. When orthogonal parameterization is used, we can use the Neumann series to approximate the matrix inversion; see Section 4.1.2 for details. Moreover, we can further improve the performance of MIGNN with orthogonal parameterization using higher-order graph diffusion. We name MIGNN with monotone parameterization using PR splitting accompanied by $K$-th order Neumann series approximation and $P$-th order graph diffusion convolution**MIGNN-N$K$D$P$**.
>
> Based on our experimental results, we see that MIGNN-Mon is significantly more accurate than IGNN for both graph classification and graph node classification. In Figure 5, we also see that the magnitude of eigenvalues of $\mathbf{W}$ is much larger than 1.

---

> > ### Author Response · Authors · 2022-11-11
> > **Response to Reviewer Ti7k (cont'd)**
> >
> > ---
> >
> > **Q2. Moreover, in Remark 2 in the original submission, the way to make $W$ symmetric will naturally remove the asymmetric part of W in the monotone operator $(F−F^T)$.**
> >
> > **Reply:** Thank you for pointing out this. The original writing in the remark caused some confusion. We actually find the fixed point of MIGNN with monotone parameterization using the Anderson-accelerated forward-backward splitting where eigendecomposition is not needed, and the asymmetric part is retained in the parameterization. In the revision, we have deleted this remark and also revised the paper substantially to make the implementation of **MIGNN-Mon** and **MIGNN-N$K$D$P$** clear.
> >
> > ---
> >
> > **Q3. Some ablation studies are missing. For example, the authors also adopt diffusion convolution in Eq.(12), where the diffusion convolution automatically has the benefit in capturing long range dependence. So, it i s unclear that in the node classification tasks, whether the benefit comes from diffusion convolution or implicit operators. Here, I also do not know what "N3D5", "N5D5" used in the model name in the experiment means, which are not defined. I assume they are related to the diffusion convolution. Please clarify. Also, provide the study of using different convolutions to show the benefits indeed come from implicit modeling.**
> >
> > **Reply:**
> > **MIGNN-N$K$D$P$** denotes MIGNN with orthogonal parameterization, and we find the fixed point of the equilibrium equation using the Anderson-accelerated Peaceman-Rachford splitting (PR) accompanied by $K$-th order Neumann series approximation and $P$-th order graph diffusion convolution. In the revised paper, we have conducted ablation studies on the order of graph diffusion convolution and the order of Neumann series approximation in Appendices G and H, respectively. These ablation studies show that both higher-order graph diffusion convolution and Neumann series approximation improves the performance of MIGNN for graph classification and graph node classification. Also, the benefits from the higher-order Neumann series approximation are more significant than that of the higher-order graph diffusion convolution.
> >
> > In Table 1 of the revision, we have compared **MIGNN-N$K$D$P$** with IGNN-related models and a few other baseline GNNs for several benchmark graph node classification tasks, including Cora, Citeseer, and Pubmed. The results in Table 1 show that MIGNN, even using first-order graph diffusion convolution, outperforms two popular explicit GNN with high-order graph diffusion convolution APPNP [32] and GCN-GDC [34], showing the benefits of implicit modeling.
> >
> > Next, we compare higher-order graph diffusion for explicit and implicit GNNs on the binary directed chain classification task using the training procedure outlined in the original IGNN paper. The table below shows the mean accuracy over 5 random initializations. The chain lengths were chosen to range from 5 to 10 in increments of 1. These lengths are significantly shorter than those tested in the paper because the GCN is limited in LRD. We denote GCN with $P$-th order diffusion convolution as GCN-D$P$.
> >
> > | **Length 5** | **D1**  | **D2** | **D3** |
> > |:----|:----|:----|:----|
> > |GCN|54.82|54.82|54.82|
> > |IGNN|100.00|100.00|100.00|
> > |MIGNN-N1|100.00|100.00|100.00|
> > |MIGNN-N3|100.00|100.00|100.00|
> > | **Length 6** ||||
> > |GCN|57.25|55.19|50.10|
> > |IGNN|100.00|100.00|100.00|
> > |MIGNN-N1|100.00|100.00|100.00|
> > |MIGNN-N3|100.00|100.00|100.00|
> > | **Length 7** | | | | | |
> > |GCN|50.78|50.23|50.23|
> > |IGNN|100.00|100.00|100.00|
> > |MIGNN-N1|100.00|100.00|100.00|
> > |MIGNN-N3|100.00|100.00|100.00|
> > | **Length 8** | | | | | |
> > |GCN|51.41|50.97|50.50|
> > |IGNN|100.00|100.00|100.00|
> > |MIGNN-N1|100.00|100.00|100.00|
> > |MIGNN-N3|100.00|100.00|100.00|
> > | **Length 9** | | | | | |
> > |GCN|54.71|54.71|54.71|
> > |IGNN|100.00|100.00|98.80|
> > |MIGNN-N1|92.41|100.00|100.00|
> > |MIGNN-N3|100.00|100.00|100.00|
> > | **Length 10** | | | | | |
> > |GCN|54.12|53.06|53.06|
> > |IGNN|100.00|100.00|98.80|
> > |MIGNN-N1|100.00|100.00|100.00|
> > |MIGNN-N3|100.00|100.00|100.00|
> >
> >
> > [32] J. Gasteiger et al. Predict then propagate: Graph neural networks meet personalized PageRank. In ICLR, 2018.
> >
> > [34] J. Gasteiger et al. Diffusion improves graph learning. In NeurIPS, 2019.

---

> > > ### Author Response · Authors · 2022-11-11
> > > **Response to Reviewer Ti7k (cont'd)**
> > >
> > > ---
> > >
> > > **Q4. Node-level experiments are too few. Actually, Long range dependence is not that crucial for graph-level tasks, because one can adopt transformer models and run on dense graphs by adding jump-hop edges. This is also why the proposed models are hard to outperform many baselines in graph-level tasks (the used baselines and datasets in this work are actually not SOTA and large). LRD is more important for node-level tasks. However, this work only provides one dataset for such evaluation.**
> > >
> > >
> > > **Reply:** Thank you for pointing this out to us. We have done several more experiments on graph node classification tasks using MIGNN with monotone and orthogonal parameterizations. As shown in Table 1 in the revised paper, MIGNN with the new two parameterizations significantly outperforms the baseline IGNN for the benchmark Cora, Citeseer, and Pubmed graph node classification tasks.

---

> ### Author Response · Authors · 2022-11-20
> **Follow up with Reviewer Ti7k**
>
> Dear Reviewer Ti7k,
>
> We thank the reviewer for the time and effort put into reviewing our paper, and we appreciate the reviewer's support and constructive comments. With our best appreciation, we have substantially revised our paper and addressed each comment from the reviewer in a very detailed manner. The quality of our paper has been significantly improved based on the reviewer's feedback.
>
> We look forward to and appreciate your further feedback. We thank the reviewer for the effort made and will make on our work.
>
> Regards,
>
> Authors of Paper #483

---

> ### Comment · Reviewer_Ti7k · 2022-11-23
> **Thanks for the comments**
>
> Many thanks for the response. I think the new manuscript is more clear. As for the improved technique, I still lean to accept the paper. The reason I cannot give a strong recommendation is due to a fundamental question in my mind, i.e., whether we indeed need an implicit GNN model that captures a long dependence. My feeling is that the current bottleneck of capturing long dependence mainly comes from computation instead of the model architecture.

---

> > ### Author Response · Authors · 2022-11-24
> > **Further Response to Reviewer Ti7k**
> >
> > We thank the reviewer for considering our rebuttal and the support of our submission. Next, please allow us to address your concerns about giving our paper a strong recommendation.
> >
> > ---
> >
> > **Q: The reason I cannot give a strong recommendation is due to a fundamental question in my mind, i.e., whether we indeed need an implicit GNN model that captures a long dependence. My feeling is that the current bottleneck of capturing long dependence mainly comes from computation instead of the model architecture.**
> >
> > **Reply:** Implicit GNNs are important and beneficial for capturing the long-range dependencies (LRD) of the underlying graphs. In the following, we provide a detailed explanation of the importance and benefits of implicit GNNs for learning LRD.
> >
> > Due to the neighborhood message passing scheme, an explicit GNN model must be sufficiently deep to learn LRD. However, the increased depth in explicit GNNs comes with the cost of increased memory consumption and performance degradation. **1)** Notice that the memory complexity of an explicit GNN is roughly $\mathcal{O}(Lnd)$, where $L$ is the number of layers, $n$ is the number of nodes, and $d$ is the feature dimension. Therefore, the increased memory consumption is prohibitive for a large graph. **2)** Moreover, the prevalent over-smoothing issue often limits the performance gain of explicit GNNs with increased depth. In contrast, ***the implicit GNN has adaptive and potentially “infinite” depth with a memory cost of  $\mathcal{O}(nd)$*** ---  independent of the depth of the model ---  using implicit differentiation at the fixed point.
> >
> >
> > The above claims have been validated experimentally in Figure 1 of [39], where the IGNN significantly outperforms the explicit GNN models, including GCN, GAT [73], SGC [79], and SSE [23] on the synthetic chain classification task, where learning LRD is crucial. Moreover, in Appendix E.5 of [39], IGNN is further compared with a 10-layer GCN-II [21] and a 10-layer DropEdge [Rong] on classifying chains with lengths less than 10. In the experiment, IGNN achieves perfect accuracy with all chain lengths, while both GCN-II and DropEdge perform near-random guesses when the chain length is greater than 5.
> >
> > Another particular advantage of implicit GNNs is that their depths are adaptive to particular data and particular tasks. For most graph learning tasks, the diameter of the input graphs varies; using a GNN with fixed depth is suboptimal. In particular, to guarantee aggregating sufficient neighboring information, we may need to stack too many graph convolution layers.
> >
> > From our experimental observation of eigenvalue dynamics of the weight matrix, we find that the employed parametrization and Picard iteration limit the efficiency and stability of the existing IGNN in learning LRD. The proposed MIGNN can learn LRD more stably and efficiently through orthogonal parametrization and PR splitting, which has been extensively validated in our experiments.
> >
> >
> > We hope we have cleared your concerns about giving our paper a strong recommendation. We are glad to answer any further questions you have on our submission. Many thanks for your consideration, and we appreciate it.
> >
> > -----
> >
> > [39] F. Gu, H. Chang, W. Zhu, S. Sojoudi, and L. El Ghaoui. Implicit graph neural networks. In NeurIPS, 2020.
> >
> > [73] P. Velickovic, G. Cucurull, A. Casanova, A. Romero, P. Lio, and Y. Bengio. Graph attention networks. In ICLR, 2018.
> >
> > [79] F. Wu, A. Souza, T. Zhang, C. Fifty, T. Yu, and K. Weinberger. Simplifying graph convolutional networks. In ICML, 2019.
> >
> > [23] H. Dai, Z. Kozareva, B. Dai, A. Smola, and L. Song. Learning steady-states of iterative algorithms over graphs. In ICML, 2018.
> >
> > [21] M. Chen, Z. Wei, Z. Huang, B. Ding, and Y. Li. Simple and deep graph convolutional networks. In ICML, 2020.
> >
> > [Rong] Y. Rong, W. Huang, T. Xu, and J. Huang. DropEdge: Towards Deep Graph Convolutional Networks on Node Classification. In ICLR, 2020.

---

> > > ### Comment · Reviewer_Ti7k · 2022-11-26
> > > **Thanks for the further comment!**
> > >
> > > I understand implicit models save memory. My question is whether it can deal with large-scale networks. My understanding of the advantage of infinite layers appears when long-hop features have to be captured. So, there is a conflict that implicit models are proposed to do long-hop extraction while implicit models have not succeeded on large-scale networks where long-hop features may be useful. So, I am actually looking forward to seeing if implicit models can leverage their memory-saving advantages and succeed in tasks over 10M or even 100M+ sized graphs.
> > >
> > > I see the datasets used in the work are small. Even, for the node-level task, only 10k-sized networks are used. Also, implicit models seem to find a hard time outperforming other models, e.g, GCN II. Overall, I like the technique proposed in the work. However, from the perspectives of impact and applications, I am not fully convinced.

---

### Official Review · Reviewer_FLVV · 2022-10-23

**Confidence:** 4
**Correctness:** 3
**Technical Novelty And Significance:** 3
**Empirical Novelty And Significance:** 2
**Recommendation:** 5

**Clarity, Quality, Novelty And Reproducibility:**

Clarity - The paper is mostly clearly written.

Quality - The paper lacks comparison with many existing methods and also a background discussion on other methods that capture long range dependencies.

Novelty - The method itself seems new.

Reproducibility - Many details to reproduce the method from the text are missing.

**Strength And Weaknesses:**

Pros:
1. The authors provide a sufficient background of implicit gnns
2. The paper is well motivated and mostly easy to follow
3. There are several interesting experiments

Cons:
1. The paper lacks discussion of relevant prior work. For example, the use of Cayley parameterization was previously proposed in "CayleyNets: Graph Convolutional Neural Networks with Complex Rational Spectral Filters". In the sense of learning long-range dependencies several works proposed various approaches, for instance learning multiple hop filters "Path Integral Based Convolution and Pooling for Graph Neural Networks" or dedicated oscillatory layers in "PDE-GCN: Novel Architectures for Graph Neural Networks Motivated by Partial Differential Equations" and in "Graph-Coupled Oscillator Networks".

2.While the experiments do show the significance of the proposed method compared to existing implicit gnns, it is hard to evaluate the performance of the proposed method compared to recent work in the field of gnns due to two reasons.
A. The authors do not compare where possible to recent methods. For example table 1 lacks many works that perform significantly better. See for example "Improving graph neural network expressivity via subgraph isomorphism counting.", "Weisfeiler and lehman go topological: Message passing simplicial networks." , "Weisfeiler and lehman go cellular: Cw networks." ,
B. Besides this experiment, all other datasets are not directly comparable with other works. This is not bad, but to really assess the performance of this method I think that it needs to include experiments that are more directly comparable like node classification on Cora,Citeseer,Pubmed,OGBN-ARXIV and others.

3.Regarding equation 1. This formulation of adding the initial features looks similar to the operation in "Simple and Deep Convolutional Graph Neural Networks" which is not discussed in this paper. Can the authors explain?

4. Regarding the claim at the bottom of the first page. I am not convinced that it is true that IGNN does not suffer from the very same problem. What is the difference here? Why should IGNNs that smooth the node features not oversmooth if the propagation is the same?

5.I understand that the proposed method requires less iterations, but I do not know how much time each iteration of the proposed method takes. I believe that if the authors add run times their results will be more convincing.

6.Proposition 2 is not clear - where does the final eqation W = ... comes from?

7.One of the hyperparameters of this method is the order of diffuion. However in different experiments the authors use different orders. It is hard to deduce what is the actual influence of the order this way. I think that the paper can benefit from an ablation study of the various hyperparameters.

8.What are the chosen hyperparameters in your experiments? How did you choose them?

**Summary Of The Paper:**

This paper presents a method to stabilize the training of implicit graph neural networks (gnns) and in less required iterations.
The proposition is based on learning a monotone operator and employing splitting techniques for a better solution of the iterative problem.

Several experiments are conducted and show the improvement of the proposed method compared to previous implicit gnns.

**Summary Of The Review:**

The paper suggest an interesting method to improve implicit graph neural networks and the results compared to such methods seem promising. However there is a lack of discussion with other relevant works, and the experimental part is limited and hard to compare with other methods.

---

> ### Author Response · Authors · 2022-11-11
> **Response to Reviewer FLVV**
>
> Thank you for your review and valuable feedback. We kindly invite the reviewer to read our general response. In what follows, we provide point-by-point responses to the raised comments.
>
> ---
>
> **Q1. The paper lacks discussion of relevant prior work. For example, the use of Cayley parameterization was previously proposed in "CayleyNets: Graph Convolutional Neural Networks with Complex Rational Spectral Filters". In the sense of learning long-range dependencies several works proposed various approaches, for instance, learning multiple hop filters "Path Integral Based Convolution and Pooling for Graph Neural Networks" or dedicated oscillatory layers in "PDE-GCN: Novel Architectures for Graph Neural Networks Motivated by Partial Differential Equations" and in "Graph-Coupled Oscillator Networks".**
>
> **Reply:** Thank you for pointing out these papers to us.
>
> We want first to point out that the **Cayley parameterization used in our paper and the Cayley polynomial in the CayleyNet [Lev] are totally different things**. By applying the Cayley transform on skew-symmetric matrices, the Cayley parameterization is an effective parameterization of the orthogonal weight matrices, which has been used in RNNs, and some related papers, e.g. [5,78,45,74,61,41] have been discussed in our paper. The Cayley polynomial in CayleyNet is a family of polynomial filters of the Laplacian matrix.
>
> We have discussed the other papers in the revision. We believe it is unfair to say that we lack the discussion of related works. **We discussed 72 references in our original submission**. Due to the page limit and the large volume of existing works on graph neural networks, it is impossible to discuss all works on graph neural networks. We focus on developing stable, efficient, and flexible implicit graph neural networks using monotone operator theory. **The most related works are existing approaches for implicit graph neural networks based on fixed point iteration, monotone operator theory, and operator splitting schemes, which have been thoroughly reviewed**. We used orthogonal parameterization, and thus we have also discussed the existing use of orthogonal parameterization in other models. The **general graph neural networks and related works on addressing over-smoothing issues are less closely related to our work; thus, we can only discuss some representative ones**. We hope the reviewer can understand this; in particular, there is a page limit for the submission. In the revision, we have further cited the papers the reviewer mentioned. In particular, the newly-cited papers are [27,28,60,67].
>
> [Lev] Levie, Ron, et al. Cayleynets: Graph convolutional neural networks with complex rational spectral filters. IEEE Transactions on Signal Processing 67.1 (2018): 97-109.
>
> [5] M. Arjovsky, A. Shah, and Y. Bengio. Unitary evolution recurrent neural networks. In International Conference on Machine Learning, pp. 1120–1128, 2016.
>
> [27] M. Eliasof, E. Haber, and E. Treister. PDE-GCN: Novel architectures for graph neural networks motivated by partial differential equations. Advances in Neural Information Processing Systems, 34:3836–3849, 2021.
>
> [28] M. Eliasof, E. Haber, and E. Treister. pathGCN: Learning general graph spatial operators from paths. In International Conference on Machine Learning, pp. 5878–5891.PMLR, 2022.
>
> [41] Kyle Helfrich, Devin Willmott, and Qiang Ye. Orthogonal recurrent neural networks with scaled Cayley transform. In International Conference on Machine Learning, pp. 1969–1978. PMLR, 2018.
>
> [45] L. Jing, Y. Shen, T. Dubcek, J. Peurifoy, S. Skirlo, Y. LeCun, M. Tegmark, and M. Soljacic. Tunable efficient unitary neural networks (eunn) and their application to RNNs. In Proceedings of the 34th International Conference on Machine Learning-Volume 70, pp. 1733–1741. JMLR. org, 2017.
>
> [60] Z. Ma, J. Xuan, Y. Wang, M. Li, and P. Lio. Path integral based convolution and pooling for graph neural networks. Advances in Neural Information Processing Systems, 33:16421–16433, 2020
>
> [61] Z. Mhammedi, A. Hellicar, A. Rahman, and J. Bailey. Efficient orthogonal parameterization of recurrent neural networks using householder reflections. In Proceedings of the 34th International Conference on Machine Learning-Volume 70, pp. 2401–2409. JMLR. org, 2017.
>
> [67] T. Rusch, B. Chamberlain, J. Rowbottom, S. Mishra, and M. Bronstein. Graph-coupled oscillator networks. In International Conference on Machine Learning, pp. 18888–18909. PMLR, 2022.
>
> [74] E. Vorontsov, C. Trabelsi, S. Kadoury, and C. Pal. On orthogonality and learning recurrent networks with long-term dependencies. In Proceedings of the 34th International Conference on Machine Learning-Volume 70, pp. 3570–3578. JMLR.org, 2017.
>
> [78] S. Wisdom, T. Powers, J. Hershey, J. Le Roux, and L. Atlas. Full-capacity unitary recurrent neural networks. In Advances in Neural Information Processing Systems, pp. 4880–4888, 2016.

---

> > ### Author Response · Authors · 2022-11-11
> > **Response to Reviewer FLVV (cont'd)**
> >
> > ---
> >
> > **Q2. While the experiments do show the significance of the proposed method compared to existing implicit gnns, it is hard to evaluate the performance of the proposed method compared to recent work in the field of gnns due to two reasons. A. The authors do not compare where possible to recent methods. For example table 1 lacks many works that perform significantly better. See for example "Improving graph neural network expressivity via subgraph isomorphism counting.", "Weisfeiler and Lehman go topological: Message passing simplicial networks.", "Weisfeiler and Lehman go cellular: CW networks.", B. Besides this experiment, all other datasets are not directly comparable with other works. This is not bad, but to really assess the performance of this method I think that it needs to include experiments that are more directly comparable like node classification on Cora, Citeseer, Pubmed, OGBN-ARXIV and others.**
> >
> > **Reply:**
> > We conducted the graph classification on datasets MUTAG, PTC, COX2, PROTEINS, and NCI1. The models from the suggested paper have been added to Table 2 in the revision. However, **except for MUTAG, none of the models in the suggested paper outperforms the best model in our paper**. We used the benchmark models that were used in a very recently published paper [22] at ICML 2022.
> >
> > As suggested, we have added some smaller-scale node classification results on Cora, Citeseer, and Pubmed in Table 1 of the revised paper. On these datasets, we can fairly compare the performance of MIGNN with existing IGNN-style models since these datasets were tested in the paper [22]. It is evident that MIGNN outperforms both IGNN and EIGNN. For larger-scale graph node classification, we have tested Amazon co-purchasing dataset; see Section 5.4 in our revised paper.
> >
> > [22] Q. Chen, Y. Wang, Y. Wang, J. Yang, and Z. Lin. Optimization-induced graph implicit nonlinear diffusion. In ICML, pp.3648–3661. PMLR, 2022.
> >
> > ---
> >
> > **Q3. Regarding equation 1. This formulation of adding the initial features looks similar to the operation in "Simple and Deep Convolutional Graph Neural Networks" which is not discussed in this paper. Can the authors explain?**
> >
> > **Reply:** Equation 1 is the model used in the original IGNN paper, where each iteration step resembles an aggregation step in recurrent GNN. The paper "Simple and Deep Convolutional Graph Neural Networks" presents a type of recurrent GNN model, and that’s why similar operations are observed. The key difference between IGNN and (explicit) recurrent GNN is that IGNN requires finding the fixed point of the equilibrium equation, and this fixed point will be the learned representation of the input graph. In contrast, recurrent GNN does not require the learned representation to be the fixed point. One major benefit of IGNN is that we can use implicit differentiation to perform backward propagation since the learned representation is the fixed point of the system. Compared to the standard backward propagation, implicit differentiation takes constant memory independent of the depth of the neural networks.
> >
> > We also notice that examples of recurrent GNN models also appeared in the early works, e.g. [38,30,57]. In the revision, we have made the notion of recurrent GNN more explicit in the discussion of GNN models in Section 1.2.
> >
> > [30] G. Claudio, and A. Micheli. Graph echo state networks. The 2010 international joint conference on neural networks (IJCNN). IEEE, 2010.
> >
> > [38] G. Marco, G. Monfardini, and F. Scarselli. A new model for learning in graph domains. Proceedings. 2005 IEEE international joint conference on neural networks. Vol. 2. No. 2005. 2005.
> >
> > [57] Y. Li et al. Gated Graph Sequence Neural Networks. Proceedings of ICLR'16. 2016.

---

> > > ### Author Response · Authors · 2022-11-11
> > > **Response to Reviewer FLVV (cont'd)**
> > >
> > > ---
> > >
> > > **Q4. Regarding the claim at the bottom of the first page. I am not convinced that it is true that IGNN does not suffer from the very same problem. What is the difference here? Why should IGNNs that smooth the node features not over-smooth if the propagation is the same?**
> > >
> > > **Reply:**
> > > Based on our experiments, IGNN suffers less from over-smoothing than many of the existing graph neural networks. This observation is consistent with the observation in several existing works that are related to IGNNs [22,39,58], as the authors of these papers have shown the remarkable performance of very deep IGNN or related models.
> > >
> > > Justifying whether IGNN can overcome over-smoothing is not what we are doing in this paper. But we can provide some of our understanding of why IGNN suffers less from over-smoothing compared to many other models in the rebuttal. Intuitively, the initial graph node features are further added to the model as a bias term in each iteration, which can alleviate the over-smoothing issue. Adding initial node features or its transformation into each iteration to alleviate over-smoothing has also been studied in other settings as well, e.g., [21,37,55,72].
> > >
> > > [21] M. Chen, Z. Wei, Z. Huang, B. Ding, and Y. Li. Simple and deep graph convolutional networks. In ICML, pp. 1725–1735.PMLR, 2020.
> > >
> > > [22] Q. Chen, Y. Wang, Y. Wang, J. Yang, and Z. Lin. Optimization-induced graph implicit nonlinear diffusion. In ICML, pp. 3648–3661. PMLR, 2022.
> > >
> > > [37] S. Gong, M. Bahri, M. Bronstein, and S. Zafeiriou. Geometrically principled connections in graph neural networks. In CVPR, pp. 11415–11424, 2020.
> > >
> > > [39] F. Gu, H. Chang, W. Zhu, S. Sojoudi, and L. El Ghaoui. Implicit graph neural networks. In NeurIPS, 2020.
> > >
> > > [55] G. Li, M. Muller, B. Ghanem, and V. Koltun. Training graph neural networks with 1000 layers. In ICML, pp. 6437–6449. PMLR, 2021.
> > >
> > > [58] J. Liu, K. Kawaguchi, B. Hooi, Y. Wang, and X. Xiao. EIGNN: Efficient infinite-depth graph neural networks. In NeurIPS, pp. 18762–18773, 2021.
> > >
> > > [72] M. Thorpe, T. Nguyen, H. Xia, T. Strohmer, A. Bertozzi, S. Osher, and B. Wang. GRAND++: Graph neural diffusion with a source term. In ICLR, 2022.
> > >
> > > ---
> > >
> > > **Q5. .I understand that the proposed method requires less iterations, but I do not know how much time each iteration of the proposed method takes. I believe that if the authors add run times their results will be more convincing.**
> > >
> > > **Reply:** In our original submission, we have compared the computational time of MIGNN with IGNN on the Amazon co-purchasing dataset, a large-scale graph node classification dataset; see the middle panel of Figure 6. In the revision, we have added in Figure 4 that when both models start learning LRD, MIGNN only takes around half of the time in each epoch compared with IGNN for directed chain classification. Furthermore, we have also included ablation studies on how the order of Neumann series approximation and the order of graph diffusion convolution affects the MIGNN's accuracy and computational time; see Appendix G and H for more details.
> > >
> > > ---
> > >
> > > **Q6. Proposition 2 is not clear - where does the final equation W = ... comes from?**
> > >
> > > **Reply:** The equation ${\mathbf{W}} = (1-m) {\mathbf{I}} - {\mathbf{C}}{\mathbf{C}}^\top + {\mathbf{F}} - {\mathbf{F}}^\top$ describes all possible parameterizations of weight matrix $\mathbf{W}$ that satisfy  ${\mathbf{W}} \preceq (1-m) {\mathbf{I}}$, and hence by Proposition 1, the fixed point is guaranteed to exist for the MIGNN model.
> > >
> > > ---
> > >
> > > **Q7. One of the hyperparameters of this method is the order of diffusion. However, in different experiments the authors use different orders. It is hard to deduce what is the actual influence of the order this way. I think that the paper can benefit from an ablation study of the various hyperparameters.**
> > >
> > > **Reply:**
> > > As you suggested, we have provided ablation studies of the diffusion order in Appendix H of the revision. It shows that high-order graph diffusion improves performance, especially for learning LRD, with a trade-off in the computational cost. See Appendix H for more details.
> > >
> > > ---
> > >
> > > **Q8. What are the chosen hyperparameters in your experiments? How did you choose them?**
> > >
> > > **Reply:**
> > > Regarding the order of Neumann series expansion and graph diffusion convolution, for most tasks, we simply choose the order of both to be 1. For the smaller scale tasks, we usually vary both orders from 1 to 5. In the revised paper, we have included ablation studies on the effects of the order of Neumann series approximation and graph diffusion convolution on the accuracy and computational efficiency of MIGNN; see Appendices G and H for details.
> > >
> > > The other hyperparameters are either adapted from the original IGNN paper or based on fine-tuning. In the newly added appendix K, we include details about the hyper-parameters used in our model.
> > >
> > > ---
> > >
> > > We look forward to and appreciate your further feedback.

---

> > > > ### Comment · Reviewer_FLVV · 2022-11-22
> > > > **Discussion**
> > > >
> > > > I thank the authors for the thorough rebuttal. I feel that most of my concerns were addressed and therefore I am happy to raise my score.
> > > >
> > > > My remaining issues are:
> > > > 1. The added experiments on cora/citeseer/pubmed do not really show a great benefit of the proposed method compared to other implicit GNNs (thats a major point),
> > > > 2. Compared to other GNNs, the performance of the proposed method is far from current SOTA methods.

---

> > > > > ### Author Response · Authors · 2022-11-22
> > > > > **Thank you for your response and addressing your further comments**
> > > > >
> > > > > We thank the reviewer for considering our rebuttal. Below we address your remaining concerns on our paper.
> > > > >
> > > > > -----
> > > > >
> > > > > **Q1. The added experiments on cora/citeseer/pubmed do not really show a great benefit of the proposed method compared to other implicit GNNs (that's a major point).**
> > > > >
> > > > > **Reply:** Learning long-range dependencies for these three small-scale graph node classification tasks is not crucial since the graph diameters are all quite small. As such, we only expect MIGNN-Mon, i.e., MIGNN with the expressive monotone parameterization, can outperform IGNN on these tasks. As shown in Table 1 in our paper, MIGNN-Mon outperforms IGNN and EIGNN on all three tasks, and ***MIGNN-Mon outperforms both IGNN and EIGNN in classifying Cora and Citeseer often by more than 1% classification accuracy.***
> > > > >
> > > > > In Table 1, IGNN and EIGNN are the direct benchmark implicit GNN models of MIGNN. GIND is another model of an implicit flavor, but its architecture is quite different from IGNN, EIGNN, and MIGNN. In particular, GIND utilizes a signed incidence matrix in neighborhood aggregation to produce nonlinear diffusion. When the graph is undirected, accommodating the requirement of a signed incidence matrix restricts the activation function to satisfy $\sigma(-x) = -\sigma(x)$. This restriction excludes ReLU. Moreover,  according to their implementation, **the depth of GIND is not adaptive**. Instead of setting a numerical tolerance for the fixed-point solving as existing fixed-point networks, GIND iterates a fixed number of times governed by a hyperparameter tuning --- requiring extra tuning effort. This approach makes the model either lose convergence guarantee with too few iterations or increase computational cost with excessive iterations.
> > > > >
> > > > > All the other models in Table 1 are not implicit architectures. We included these results for comparison following the paper "Chen et al. Optimization-Induced Graph Implicit Nonlinear Diffusion, ICML 2022".
> > > > >
> > > > > -----
> > > > >
> > > > > **Q2. Compared to other GNNs, the performance of the proposed method is far from current SOTA methods.**
> > > > >
> > > > > **Reply:** Based on our understanding and experience, the SOTA results are task- and model-dependent. There is no model to achieve SOTA performance on all tasks. Perhaps many of the strategies used in the existing GNNs can be integrated into the proposed MIGNN framework to boost the performance of MIGNN further. Indeed, how to synergistically integrate existing GNN techniques into the MIGNN framework is an interesting future direction.
> > > > >
> > > > > MIGNNs are based on new parameterizations of IGNNs and utilize operator splitting schemes to learn graph representations. A directly comparable benchmark for MIGNN is IGNN, and we show that MIGNN significantly outperforms IGNN on extensive benchmark graph learning tasks.
> > > > >
> > > > > Implicit GNNs --- including IGNN, EIGNN, and MIGNN --- have their advantages over other GNNs. For instance, effectiveness in learning long-range dependencies, adaptive depth, and training with memory efficiency. It has been shown in the original IGNN paper "Gu et al. Implicit Graph Neural Networks, NeurIPS, 2020" IGNNs achieve SOTA results over many existing GNNs on various benchmark tasks, and MIGNN further improves over IGNN through new parameterizations.
> > > > >
> > > > > -----
> > > > >
> > > > > We are glad to answer any further questions you have on our submission.
> > > > >
> > > > > -----

---

> ### Author Response · Authors · 2022-11-20
> **Request for feedback on the rebuttal**
>
> Dear Reviewer FLVV,
>
>    We thank the reviewer for the time and effort put into reviewing our paper. With our best appreciation, we have substantially revised our paper and addressed each comment from the reviewer in a very detailed manner. We believe we have clarified the misunderstandings of the reviewer in our paper, addressed our concerns on our paper, and discussed the papers the reviewer listed.
>
>    We kindly invite the reviewer to read our rebuttal and respond to us. We will take all the comments carefully and address the concerns with our best efforts. We thank the reviewer for the effort made and will make on our work.
>
>
> Regards,
>
> Authors of Paper #483

---

### Official Review · Reviewer_h7JA · 2022-10-27

**Confidence:** 4
**Correctness:** 2
**Technical Novelty And Significance:** 2
**Empirical Novelty And Significance:** 2
**Recommendation:** 6

**Clarity, Quality, Novelty And Reproducibility:**

The paper is clearly written. The novelty and contribution are limited. The experiments are not convincing.

**Strength And Weaknesses:**

# Strength

1. The paper makes a nice observation that the accuracy of IGNN seems to correlate with the eigenvalue of $W$ in the implicit model. This is an interesting observation and motivates the work in the paper.

2. The paper generalizes the well-posedness condition for IGNN and proposes more flexible constructions based on monotone operator theory. New parameterizations are proposed to define a new model MIGNN.

3. The paper comprehensively discusses the technical details of related works, which is helpful to understand the idea in the paper.

4. Extensive experiments are presented with discussions.

# Weakness

1. The contribution and significance of the proposed ideas are unclear. For instance, many techniques in MIGNN have been extensively studied in the literature and even in the study of GNNs, such as Anderson acceleration, Neumann series approximation, high-order graph diffusion, eigendecomposition, etc. Moreover, monotone operator theory and operator splitting are not new (even in the context of deep learning). Therefore, the proposed idea seems an incremental combination of many techniques but the main contribution is unclear.

2. Among all the techniques in the proposed MIGNN, the flexible parameterization in Section 3 seems to be new and interesting. However, the paper fails to justify the effectiveness of this key innovation due to the lack of a comprehensive ablation study. The improvements over IGNN may be due to other existing techniques that can be trivially applied in IGNN as well such as Anderson acceleration, Neumann series approximation, high-order graph diffusion, eigendecomposition, etc. In fact, Figure 3 partially confirms this since the high-order diffusion in IGNN-D5 significantly improves the performance of IGNN.

3. The comparison with IGNN is unfair. In fact, each forward iteration of MIGNN has more computation and graph aggregations than each forward iteration of IGNN due to the Neumann series approximation and diffusion convolutions. For instance, it is mentioned in the paper that "Each node can access information from its K-hop neighbors using the K-th order Neumann series approximated PR iteration". Therefore, the concept of iteration needs to be clearly and fairly defined when comparing their stability, accuracy, and efficiency.

4. There is a lack of computation complexity analysis. In fact, the computation cost of MIGNN is pretty high, and it is unclear how it outperforms IGNN or EIGNN theoretically and empirically.

5. The motivation for monotone parameterization in Section 3.2 is unclear. Why do you define G as L/2? What are the intuition and advantages？


**Summary Of The Paper:**

The paper proposes to improve the stability, accuracy, and efficiency of implicit graph neural networks (IGNN) by new parameterizations (i.e., the Cayley transform-based orthogonal parameterization and monotone parameterization) and advanced solvers (i.e., operator splitting, Anderson acceleration). Theoretical justification for the well-posedness for the proposed MIGNN is presented, and empirical experiments demonstrate the performance of MIGNN.

**Summary Of The Review:**

The paper introduces interesting observations and ideas to improve IGNN. However, the contribution and significance are unclear, and the evaluation is not convincing enough to justify the effectiveness of the proposed algorithm.

## After rebuttal
The revision significantly improves the paper. I am willing to increase my score.

---

> ### Author Response · Authors · 2022-11-11
> **Response to Reviewer h7JA --- Clarifying misunderstandings of our contribution (cont'd)**
>
> Fourth, let us explain why we use Anderson accelerated PR, with some approximations, to find the fixed point of MIGNN with orthogonal parameterization. Among the three choices PR is better than DR, as explained in Remark 2 in our revised paper. FB requires stronger conditions for convergence guarantee and converges slower than PR in general. When orthogonal parameterization is used, we can use the Neumann series to approximate the matrix inversion; see Section 4.1.2 for details. Moreover, we can further improve the performance of MIGNN with orthogonal parameterization using higher-order graph diffusion convolution. We name MIGNN with orthogonal parameterization using PR splitting accompanied by $K$-th order Neumann series approximation and $P$-th order graph diffusion convolution **MIGNN-N$K$D$P$**.
>
> In summary, our proposed MIGNN is motivated by the newly observed bottlenecks of existing IGNN and backed up by solid monotone operator theory. Moreover, MIGNN remarkably outperforms existing IGNNs on various benchmark tasks and achieves state-of-the-art performance on many benchmark tasks.
>
> ----
>
> **Q2. Among all the techniques in the proposed MIGNN, the flexible parameterization in Section 3 seems to be new and interesting. However, the paper fails to justify the effectiveness of this key innovation due to the lack of a comprehensive ablation study. The improvements over IGNN may be due to other existing techniques that can be trivially applied in IGNN as well such as Anderson acceleration, Neumann series approximation, high-order graph diffusion, eigendecomposition, etc. In fact, Figure 3 partially confirms this since the high-order diffusion in IGNN-D5 significantly improves the performance of IGNN.**
>
> **Reply:** In the revised paper, for MIGNN-N$K$D$P$, we have done ablation studies on the effects of the order of the Neumann series approximation and the graph diffusion convolution in Appendices G and H, respectively. We have not introduced any approximation or further parameters in MIGNN-Mon, and there is no need to perform an ablation study.
>
> **The existing IGNN uses Picard iteration to find the fixed point, and the weight matrix $\mathbf{W}$ is constrained using a tractable projected gradient descent method. The Anderson acceleration and Neumann series approximation are used to efficiently find the fixed point of the equilibrium equation of MIGNN with orthogonal parameterization.** The graph diffusion convolution is used to improve MIGNN for learning LRD. Among the aforementioned techniques, the only one that can be used to improve IGNN is high-order graph diffusion without much heavy lifting. How to apply the other techniques within the existing IGNN framework is unclear to us since the major challenge is that the tractable projected gradient descent method will not work anymore. Also, notice that the existing IGNN uses Picard iteration for finding the fixed point of the equilibrium equation rather than operator splitting schemes, and there is no matrix inversion that needs to be approximated by the Neumann series.
>
> Even when the high-order graph diffusion is used within the IGNN framework, the performance is still significantly worse than the proposed MIGNN. As shown in Figure 3, MIGNN-N3D5 is significantly more robust and accurate than IGNN-D5 for classifying directed chains.

---

> > ### Author Response · Authors · 2022-11-11
> > **Response to Reviewer h7JA --- Clarifying misunderstandings of our contribution (cont'd)**
> >
> > ---
> >
> > **Q3. The comparison with IGNN is unfair. In fact, each forward iteration of MIGNN has more computation and graph aggregations than each forward iteration of IGNN due to the Neumann series approximation and diffusion convolutions. For instance, it is mentioned in the paper that "Each node can access information from its K-hop neighbors using the K-th order Neumann series approximated PR iteration". Therefore, the concept of iteration needs to be clearly and fairly defined when comparing their stability, accuracy, and efficiency.**
> >
> > **Reply:**
> > We respectfully disagree with the comment that the comparison with IGNN is unfair.
> >
> > Regarding the computational efficiency comparison between IGNN and MIGNN, we want to point out that we have already included in Figure 6 the comparison of time elapsed in each epoch between IGNN and **MIGNN-N$1$D$1$** on the Amazon co-purchasing dataset, which is recognized as a benchmark dataset of LRD learning. Figure 6 clearly shows that the MIGNN is more computationally efficient. For further testing how the parameters $K$ and $P$ in **MIGNN-N$K$D$P$** affect the computational time, we have added in Figure 4 that when both models start learning LRD, the **MIGNN-N$2$D$5$** only takes around half of the time in each epoch compared with IGNN for directed chain classification. In the revision, we have also included ablation studies on how the order of Neumann series approximation and the order of graph diffusion convolution affects the MIGNN's accuracy and computational time; see Appendix G and H for more details.
> >
> > Regarding the stability and accuracy comparison between IGNN and MIGNN. We stress again that to learn LRD, IGNN has to push the magnitude of some eigenvalues of the weight matrix $\mathbf{W}$ to 1, resulting in slow convergence of Picard iteration. Also, during the training of IGNN, the magnitude of eigenvalues of $\mathbf{W}$ (starting from random Gaussian initialization) may not converge to 1, resulting in instability for learning LRD. Moreover, the parameterization used in IGNN constrains the magnitude of eigenvalues of $\mathbf{W}$ to be less than $1$, limiting the expressivity of IGNN. In contrast, the proposed MIGNN with orthogonal and monotone parameterizations are used to improve learning LRD and boost the model’s expressivity, respectively. In Figure 3, we have compared the performance between MIGNN with IGNN, in learning LRD, with and without using the higher-order graph diffusion convolution. In Table 1 and Table 2, we also compare the classification accuracy between MIGNN with IGNN on a few benchmark graph and graph node classification tasks.

---

> > > ### Author Response · Authors · 2022-11-11
> > > **Response to Reviewer h7JA --- Addressing other comments**
> > >
> > > ---
> > >
> > > **Q4. There is a lack of computation complexity analysis. In fact, the computation cost of MIGNN is pretty high, and it is unclear how it outperforms IGNN or EIGNN theoretically and empirically.**
> > >
> > > **Reply:** In each epoch that contains a forward and backward propagation, the IGNN model has time complexity $\mathcal{O}(Md|E|)$ where $M$ is the maximum number of iterations, $d$ is the feature dimension, and $|E|$ is the number edge. Meanwhile, the tractable project gradient used in IGNN that ensures $||\mathbf{W}||_{\infty}< 1$ takes roughly $\mathcal{O}(n^2)$ between two epochs.
> > >
> > > The time complexity of EIGNN is only $\mathcal{O}(d^3 + d^2n)$ where $n$ the number of nodes. However, EIGNN requires an eigendecomposition preprocessing before the training which has a prohibitive $\mathcal{O}(n^3)$ time complexity.
> > >
> > > **MIGNN-Mon** uses FB splitting and has the same complexity as IGNN. **MIGNN-N$K$D$P$** computes the $P$-th order diffusion matrix once in the preprocessing with time complexity $\mathcal{O}(nP|E_P|)$ where $|E_P|$ denotes the number of non-zero entries in $\mathbf{A}^P$. In each epoch, the parameter $K$ in the $K$-th order Neumann series affects the training time complexity linearly as $\mathcal{O}(KMd|E_P|)$. We have included the computational complexity analysis in the revision.
> > >
> > > In our experiments, thanks to the efficiency of PR splitting, which is observed in training deep equilibrium networks [77], **MIGNN-N$K$D$P$** takes fewer iterations than IGNN to converge and has an overall faster training time. This has been confirmed by the experimental results in Figures 4 and 6 in the revised paper.
> > >
> > > [77] Ezra Winston and J. Zico Kolter. Monotone operator equilibrium networks. In Advances in neural information processing systems, volume 33, pp. 10718–10728, 2020.
> > >
> > > ----
> > >
> > > **Q5. The motivation for monotone parameterization in Section 3.2 is unclear. Why do you define G as L/2? What are the intuition and advantages**
> > >
> > > **Reply:** The motivation for monotone parameterization in Section 3.2 is to allow the weight matrix $\mathbf{W}$ to be much broader than that used in the original IGNN, resulting in more expressive models. Notice that the magnitude of eigenvalues of $\mathbf{W}$ for the original IGNN is in the range $[0,1)$. In contrast, monotone parameterization enables the magnitude of eigenvalues of $\mathbf{W}$ to be in the range $[0,\infty)$. More precisely, monotone parameterization allows the real part of the eigenvalue in the range $(-\infty,1)$ and the imaginary part to be arbitrary.
> > >
> > > We set $\mathbf{G}$ to be ${\mathbf{L}}/2$ to ensure the eigenvalue of $\mathbf{G}$ lies in $[0,1]$, and thus $\frac{1}{2}({\mathbf{G}}^{\top}\otimes{\mathbf{W}} + {\mathbf{G}} \otimes {\mathbf{W}}^\top)\preceq (1-m){\mathbf{I}}$ when monotone parameterization is used for parameterizing $\mathbf{W}$. Notice that we cannot simply adopt the same $\mathbf{G}$ as used in the original IGNN, which can take negative eigenvalues.
> > >
> > > ----
> > >
> > > We look forward to and appreciate your further feedback.

---

> ### Author Response · Authors · 2022-11-11
> **Response to Reviewer h7JA --- Clarifying misunderstandings of our contribution**
>
> Thank you for your review and valuable feedback. We believe there are some misunderstandings about our work. We have revised our paper according to your feedback and to make it easier to read with significant changes highlighted in blue. We kindly invite the reviewer to read our general response. In what follows, we provide point-by-point responses to the raised comments.
>
> ----
>
> **Q1. The contribution and significance of the proposed ideas are unclear. For instance, many techniques in MIGNN have been extensively studied in the literature and even in the study of GNNs, such as Anderson acceleration, Neumann series approximation, high-order graph diffusion, eigendecomposition, etc. Moreover, monotone operator theory and operator splitting are not new (even in the context of deep learning). Therefore, the proposed idea seems an incremental combination of many techniques but the main contribution is unclear.**
>
> **Reply:**  We respectfully disagree with the comment that our idea seems to be an incremental combination of many techniques. There are some misunderstandings about our work, and it is very unfair to say that the proposed idea is an incremental combination of many techniques. Below we will clarify our contribution.
>
> Our proposed MIGNN was inspired by two observed bottlenecks of the existing IGNN based on Picard iteration for finding the fixed point: **1)** To learn long-range dependencies (LRD), IGNN has to push the magnitude of some eigenvalues of the weight matrix $\mathbf{W}$ to 1, which results in slow convergence of Picard iteration. Also, during the training of IGNN, the eigenvalue of $\mathbf{W}$ (starting from random Gaussian initialization) may not converge to 1, resulting in instability for learning LRD. **2)** The parameterization used in IGNN constrains the magnitude of eigenvalues of $\mathbf{W}$ to be less than $1$, limiting the expressivity of IGNN. To address the above two issues of IGNN, we propose MIGNN with orthogonal and monotone parameterizations. **Orthogonal parameterization aims to improve and stabilize MIGNN for learning LRD. Monotone parameterization is used to boost the expressivity of MIGNN.** Notice that MIGNN contains two ingredients: orthogonal or monotone parameterization for parameterizing the equilibrium equation and fast algorithms for finding the fixed point of the equilibrium equation. Below, we explain these two ingredients in detail.
>
> First, **as you mentioned, the flexible parameterizations in Section 3 are new and interesting.** Indeed, these new parameterizations are fundamental contributions of our paper, **as pointed out in Section 1.1.** **1)** We notice that the existing IGNN is unstable and inefficient in learning long-range dependencies (LRD), and we propose MIGNN with orthogonal parameterization to enhance the stability and efficiency of learning LRD. **2)** We further propose monotone parameterization for MIGNN to improve the expressivity for graph learning. Notice that, in the IGNN parameterization of $\mathbf{W}$, the magnitude of eigenvalues has to be less than $1$ to guarantee the well-posedness of IGNN. In contrast, MIGNN with monotone parameterization allows the real part of eigenvalues of $\mathbf{W}$ to be in the range $(-\infty, 1)$ and the imaginary part to be arbitrary, which is much wider than that of IGNN.
>
> Second, why do we need the techniques such as operator splitting, Anderson acceleration, etc., discussed in Section 4? This is because **both orthogonal and monotone parameterizations are not in the efficient convergence regime of Picard iteration.** We have to apply other numerical algorithms to find the fixed point of MIGNN with orthogonal or monotone parameterization. Particularly appealing numerical algorithms are based on Anderson-accelerated operator splitting schemes.
>
> Third, let us explain why we apply the Anderson accelerated forward-backward operator splitting (FB) to find the fixed point of MIGNN with monotone parameterization, resulting in the model **MIGNN-Mon**. There are three popular operator splitting schemes: FB, Peaceman-Rachford (PR), and Douglas-Rachford (DR). PR and DR require inverting large matrices, which is inefficient and unscalable. Moreover, we cannot use the Neumann series to approximate the inversion of the matrices when the monotone parameterization is used since the parameterization has eigenvalues whose real part can be much less than $-1$. As such, we select FB to find the fixed point of MIGNN with monotone parameterization.

---

> ### Author Response · Authors · 2022-11-20
> **Request for feedback on the rebuttal**
>
> Dear Reviewer h7JA,
>
>    We thank the reviewer for the time and effort put into reviewing our paper. With our best appreciation, we have substantially revised our paper and addressed each comment from the reviewer in a very detailed manner. We believe we have clarified the misunderstandings of the reviewer in our paper and addressed our concerns on our paper.
>
>    We kindly invite the reviewer to read our rebuttal and respond to us. We will take all the comments carefully and address the concerns with our best efforts. We thank the reviewer for the effort made and will make on our work.
>
>
> Regards,
>
> Authors of Paper #483

---

> > ### Comment · Reviewer_h7JA · 2022-11-22
> > **Further Comments**
> >
> > Dear authors,
> >
> > Thank you for the detailed response. The revision significantly improves the paper. I am happy to increase my score.
> >
> > Reviewer h7JA

---

> > > ### Author Response · Authors · 2022-11-22
> > > **Thank you for your response and support**
> > >
> > > We thank the reviewer for considering our rebuttal and the support of our submission.

---

### Official Review · Reviewer_Y8Ag · 2022-11-09

**Confidence:** 2
**Correctness:** 3
**Technical Novelty And Significance:** 2
**Empirical Novelty And Significance:** 3
**Recommendation:** 5

**Clarity, Quality, Novelty And Reproducibility:**

【Clarity】
The mathematical part is clearly written. Although I was unfamiliar with the monotone operator theory, I could understand the paper thanks to the brief review in the appendix.

P3 (3): At first reading, it was difficult to interpret the monotone inclusion problem, $0\in (\mathcal{F}+\mathcal{G})(x)$. It is better to clarify that $\mathcal{F}$ and $\mathcal{G}$ are set-valued functions and that $\partial f$ is subgradient.

Appendix E.2: Proposition 2 is in Section 3. However, its proof is in Appendix E.2, whose title is Proofs for Section 2.


【Quality】
This paper claimed at the beginning of Section 3 that the monotone parameterization increases the expressive power of IGNNs. However, I wonder how this claim is supported. If I understand correctly, the superiority of the monotone parameterization came from two points (1) $G$ is positive definite, and (2) $W$ can represent any matrix with eigenvalues less than 1. However, these two points have been achieved in the original IGNN. Therefore, I want to clarify this point.

Regarding the empirical evaluation, I would like to clarify how the authors chose the experiment setups. For example, I have the following questions:
- In Figure 3, IGNN-D5 is used as the baseline in addition to IGNN, while Figure 4 uses only IGNN.
- How were the N and D parameters determined?
- The synthesis dataset uses the orthogonal parameterization (Section 5.1), whereas the real dataset the monotone parameterization (Section 5.2). How were these parameterizations chosen?

One research question was that as $\lambda_1(W)$ approached 1, the convergence of the fixed-point calculation became slower while the prediction accuracy increased. The result in Figure 6 and Figure 7 effectively answered this problem to some extent. Specifically, IGNN and MIGNN-N1D1 achieve similar accuracy, but the elapsed time and number of iterations required for the computation are reduced. It implies that MIGNN has improvements on this problem.
On the other hand, in Section 5.2, the large value of $\lambda_1(|W|)$ and the better accuracy simultaneously happened to MIGNN. Therefore, it is not known from these results alone whether the MIGNN solved the above problems in this setting.

Related to the above, the interpretation of the quantity of $\lambda_1(|W|)$ looks somewhat inconsistent. Specifically, in some places (e.g., Section 3.1, Section 5.2), large $\lambda_1(|W|)$ is interpreted favorably, while smaller $\lambda_1(|W|)$ was considered good in other places (e.g., Section 4.1.1, Section 5.3). Therefore, I want to clarify what the authors think is the appropriate scale for $\lambda_1(|W|)$.


【Novelty】
If I understand correctly, the improvements proposed in this paper are as follows:
1. Orthogonal parametrization of the weight matrix $W$
2. monotone parameterization for $W$
3. Anderson accelerated operator splitting scheme to compute fixed points of IGNN
4. Use of higher order powers of adjacency matrix $A$ as $G$
As pointed out in this paper, modifications similar to the first and third points were observed in other NN models (RNN and FNN). On the other hand, the second point is a newly proposed parameterization based on the theoretical results of this paper (Proposition 1). The fourth point is an improvement method used in many GNNs (e.g., N-GCN [Abu-El-Haija et al., 20]).

[Abu-El-Haija et al., 20]: http://proceedings.mlr.press/v115/abu-el-haija20a.html


【Reproducibility】
The experimental code is provided in almost complete form. Therefore, I think we can check the details of the experiments, although I have not run this experimental code by myself.

**Details Of Ethics Concerns:**

N.A.

**Strength And Weaknesses:**

【Strengths】
- Provides basics of monotone operator theory, which enables the paper to be more accessible to those unfamiliar with monotone operator theory.


【Weaknesses】
- Novelty of the proposed method is limited because the methodology primarily relied on the combination of existing methods.
- Interpretation of $\lambda_1(W)$, which may be related to IGNN convergence and prediction accuracy, is somewhat questionable.

**Summary Of The Paper:**

This paper proposed a variant of Implicit Graph Neural Network (IGNN) based on the monotone operator theory. First, this paper formalized the convergence of IGNN as a monotone inclusion problem and derived the sufficient condition for well-posedness. Then, two types of parameterization for the weight matrix were proposed based on this condition: the orthogonal parametrization and the monotone parametrization. In addition, to speed up the convergence of IGNN, this paper proposed two acceleration methods based on the forward-backward splitting scheme and the Pearceman-Rachford splitting scheme, respectively. Finally, this paper applied the proposed model to the chain classification task on synthesis data and to the graph and node prediction tasks on real data to assess the prediction accuracy and computational efficiency.

**Summary Of The Review:**

Theoretical part is clearly written and sound. Also, it is accessible to those who are not familar with the monotone operator theory.
If I understand correctly, most improvements are the application of existing techniques for computing fixed points by iterative algorithms, except for the monotone parameterization. Also, I have questions about the design of experiment settings and the interpretation of the experiment results.

---

> ### Author Response · Authors · 2022-11-11
> **Response to Reviewer Y8Ag --- Clarifying misunderstandings of our contribution**
>
> Thank you for your thoughtful review and valuable feedback. We have revised our paper according to your feedback. We kindly invite the reviewer to read our general response. In what follows, we provide point-by-point responses to the raised comments.
>
> -----
>
> First of all, we fear there is a misunderstanding of MIGNN with monotone parameterization. We want to point out that **the monotone parameterization allows the real part of the eigenvalue of the weight matrix $\mathbf{W}$ to be in the range $(-\infty,1)$, and the imaginary part to be arbitrary**. In contrast, the well-posedness guarantee of IGNN requires $||\mathbf{W}||_\infty < 1$, which constrains the magnitude of eigenvalues of $\mathbf{W}$ to be less than $1$. Therefore, MIGNN with monotone parameterization provides a much more general weight matrix $\mathbf{W}$ than IGNN.
>
> -----
>
> **Q1. Novelty of the proposed method is limited because the methodology primarily relied on the combination of existing methods.**
>
> **Reply:**
> We respectfully disagree with the comment that the novelty of the proposed method is limited. We fear there are some misunderstandings of our contributions. Below we will clarify our contribution.
>
> ***Motivation:*** Our proposed MIGNN is motivated by two observed bottlenecks of the existing IGNN: **1)** To learn LRD, IGNN has to push the magnitude of some eigenvalues of the weight matrix $\mathbf{W}$ towards 1, resulting in **slow convergence** of Picard iteration. Also, during the training of IGNN, the magnitude of eigenvalues of $\mathbf{W}$ (starting from random Gaussian initialization) may not converge to 1, resulting in **instability** for learning LRD. **2)** The parameterization used in IGNN constrains the magnitude of eigenvalues of $\mathbf{W}$ to be less than $1$, limiting the expressivity of IGNN.
>
> ***Parameterization of MIGNN:***
> We propose MIGNN with orthogonal and monotone parameterizations to address the above two issues. **1)** We propose MIGNN with orthogonal parameterization to enhance the stability and efficiency of learning LRD --- as the magnitude of eigenvalues of orthogonal parameterization converges to one stably. **2)** We further propose monotone parameterization for MIGNN to improve the expressivity for graph learning. Note that the magnitude of eigenvalues of $\mathbf{W}$ in IGNN has to be in the range of $[0,1)$ to guarantee the well-posedness of IGNN. In contrast, MIGNN with monotone parameterization allows the real part of eigenvalues of $\mathbf{W}$ to be in the range $(-\infty, 1)$ and the imaginary part to be arbitrary, which is much wider than that of IGNN.
>
> ***Finding the fixed-point of MIGNN with new parameterizations:***
> However, **both orthogonal and monotone parameterizations are not within the efficient convergence regime of Picard iteration**; we have to apply other numerical algorithms to find the fixed point of MIGNN with orthogonal or monotone parameterization. Particularly appealing numerical algorithms are Anderson-accelerated operator splitting schemes.
>
> First, we apply the Anderson accelerated forward-backward operator splitting (FB) to find the fixed point of MIGNN with monotone parameterization, resulting in the model **MIGNN-Mon**. Let us briefly explain this choice, there are three popular operator splitting schemes: FB, Peaceman-Rachford (PR), and Douglas-Rachford (DR). PR and DR require inverting large matrices, which is inefficient and unscalable. Moreover, we cannot use the Neumann series to approximate the inversion of the matrices when the monotone parameterization is used since the parameterization has eigenvalues whose real part can be much less than $-1$. As such, we select FB to find the fixed point of MIGNN with monotone parameterization.
>
> Second, we use Anderson-accelerated PR, with approximations, to find the fixed point of MIGNN with orthogonal parameterization. The rationale of this choice is that among the three choices PR is better than DR, as explained in Remark 2 in the revision. FB requires stronger conditions for convergence guarantee and converges slower than PR in general. When orthogonal parameterization is used, we can use the Neumann series to approximate the matrix inversion; see Section 4.1.2 for details. Moreover, we can further improve the performance of MIGNN with orthogonal parameterization using higher-order graph diffusion. We name MIGNN with orthogonal parameterization using PR splitting accompanied by $K$-th order Neumann series approximation and $P$-th order graph diffusion **MIGNN-N$K$D$P$**.
>
> ***Performance:***
> We have compared the performance of MIGNN with the benchmark IGNN and several state-of-the-art GNNs on various benchmark graph classification tasks at both node and graph levels. The numerical results resonate with our theoretical results. In particular, the extensive numerical results consistently show that MIGNN significantly outperforms the benchmark IGNN, and often MIGNN achieves the best results among all the baseline models.

---

> > ### Author Response · Authors · 2022-11-11
> > **Response to Reviewer Y8Ag --- Addressing other comments**
> >
> > ---
> >
> > **Q2. Interpretation of $\lambda_1({\mathbf{W}})$, which may be related to IGNN convergence and prediction accuracy, is somewhat questionable.**
> >
> > **Reply:** Let us clarify the interpretation of $\lambda_1({\mathbf{W}})$. First, we stress that the existing IGNN finds the fixed point of the equilibrium equation using Picard iteration, and the resulting fixed point is the learned representation of the input graph. Second, to learn LRD, each node needs to aggregate information from the graph nodes that are far away. Since each node aggregates information from one additional hop of neighbors after one Picard iteration, the Picard iteration has to run for enough iterations before converging. This happens when the magnitude of eigenvalues approaches one. This claim is thoroughly tested on the chain dataset where learning LRD is essential, e.g. in Figure 1 and Figure 2, the IGNN only starts to learn LRD well when the magnitude of eigenvalue(s) gets closer to one. More numerical evidence is available in Appendix I.
> >
> > -----
> >
> > **Q3. P3 (3): At first reading, it was difficult to interpret the monotone inclusion problem, $0\in (\mathcal{F} + \mathcal{G})(x)$. It is better to clarify that $\mathcal{F}$ and $\mathcal{G}$ are set-valued functions and that $\partial f$ is subgradient.**
> >
> > **Reply:** Thank you for your suggestion, and we have revised the paper as you suggested.
> >
> > -----
> >
> > **Q4. Appendix E.2: Proposition 2 is in Section 3. However, its proof is in Appendix E.2, whose title is Proofs for Section 2.**
> >
> > **Reply:** Thank you for pointing out this; we have moved the proof of Proposition 2 to Appendix E.3 titled Proofs for Section 3.
> >
> > -----
> >
> > **Q5. This paper claimed at the beginning of Section 3 that the monotone parameterization increases the expressive power of IGNNs. However, I wonder how this claim is supported. If I understand correctly, the superiority of the monotone parameterization came from two points (1)  $\mathbf{G}$ is positive definite, and (2) $\mathbf{W}$ can represent any matrix with eigenvalues less than 1. However, these two points have been achieved in the original IGNN. Therefore, I want to clarify this point.**
> >
> > **Reply:** In order to achieve their well-posedness guarantee, the weight matrix $\mathbf{W}$ in **IGNN** is parametrized by requiring $||\mathbf{W}||_{\infty}<1$, which constrains the magnitude of eigenvalues of ${\mathbf{W}}$ to be less than $1$. Notice that the existing IGNN uses Picard iteration to find the fixed point of IGNN.
> >
> > In contrast, the monotone parameterization ${\mathbf{W}} = (1-m){\mathbf{I}} - {\mathbf{C}}{\mathbf{C}}^\top + {\mathbf{F}} - {\mathbf{F}}^\top$ **allows the real part of the eigenvalue of ${\mathbf{W}}$ to be in the range $(-\infty,1)$ and the imaginary part to be arbitrary**. Therefore, the eigenvalue of ${\mathbf{W}}$ allows a much wider range than that of IGNN.
> >
> > -----
> >
> > **Q6. Regarding the empirical evaluation, I would like to clarify how the authors chose the experimental setups. For example, I have the following questions: 1) In Figure 3, IGNN-D5 is used as the baseline in addition to IGNN, while Figure 4 uses only IGNN. 2) How were the N and D parameters determined? 3) The synthesis dataset uses orthogonal parameterization (Section 5.1), whereas the real dataset the monotone parameterization (Section 5.2). How were these parameterizations chosen?**
> >
> > **Reply:** 1). In Figure 3, we included IGNN-D5 as a benchmark model to show that MIGNN-N3D5 outperforms IGNN-D5; that is, though we can apply high-order graph diffusion convolution to both IGNN and MIGNN, MIGNN still remarkably outperforms IGNN. Indeed, this is a prevalent advantage of MIGNN over IGNN. For the sake of presentation, we only use IGNN-D5 as a benchmark model once.
> >
> > 2). For most tasks, we simply choose both N and D to be 1. For the smaller scale tasks, we usually vary N from 1 to 5 and D from 1 to 5. In the revised paper, we have included ablation studies on the effects of N and D on the accuracy and computational efficiency of MIGNN; see Appendices G and H for details.
> >
> > 3). Generally, for large-scale graphs and tasks that require learning LRD, we use MIGNN with orthogonal parameterization. For small-scale graphs, we can use MIGNN with either orthogonal or monotone parameterization.

---

> > > ### Author Response · Authors · 2022-11-11
> > > **Response to Reviewer Y8Ag --- Addressing other comments (cont'd)**
> > >
> > > ---
> > >
> > > **Q7. One research question was that as $\lambda_1({\mathbf{W}})$ approached 1, the convergence of the fixed-point calculation became slower while the prediction accuracy increased. The result in Figure 6 and Figure 7 effectively answered this problem to some extent. Specifically, IGNN and MIGNN-N1D1 achieve similar accuracy, but the elapsed time and number of iterations required for the computation are reduced. It implies that MIGNN has improvements on this problem. On the other hand, in Section 5.2, the large value of $\lambda_1({\mathbf{W}})$ and the better accuracy simultaneously happened to MIGNN. Therefore, it is not known from these results alone whether the MIGNN solved the above problems in this setting. Specifically, in some places (e.g., Section 3.1, Section 5.2), large $\lambda_1(|{\mathbf{W}}|)$ is interpreted favorably, while smaller $\lambda_1(|{\mathbf{W}}|)$ was considered good in other places (e.g., Section 4.1.1, Section 5.3). Therefore, I want to clarify what the authors think is the appropriate scale for $\lambda_1(|{\mathbf{W}}|)$.**
> > >
> > > **Reply:** Figure 5 of Section 5.2 plots the eigenvalues of **MIGNN with monotone parameterization**, i.e., MIGNN-Mon. Recall that MIGNN with monotone parameterization allows the eigenvalues of the weight matrix to take a much wider range than that of IGNN. Figure 5 in Section 5.2 shows that when MIGNN-Mon becomes expressive for graph classification, its eigenvalue can go beyond the well-posedness regime of the original IGNN, i.e., the magnitude of eigenvalues of MIGNN-Mon may go beyond the range $[0,1)$.
> > >
> > > In general, MIGNN with **orthogonal parameterization** is used to learn LRD efficiently and stably, in which case we desire that the magnitude of eigenvalues converges to one. MIGNN with **monotone parameterization** is designed to improve the expressivity of the IGNN model rather than improving learning LRD. And when MIGNN with monotone parameterization becomes very expressive, the magnitude of its eigenvalues can be much larger than 1.
> > >
> > > -----
> > >
> > > **Q8. Use of higher order powers of adjacency matrix $\mathbf{A}$ as $\mathbf{G}$  is an improvement method used in many GNNs (e.g., N-GCN Abu-El-Haija et al., 20).**
> > >
> > > **Reply:** Thank you for pointing out this reference to us, and we have discussed this in the revision.
> > >
> > > -----
> > >
> > > **Q9. If I understand correctly, most improvements are the application of existing techniques for computing fixed points by iterative algorithms, except for the monotone parameterization. Also, I have questions about the design of experiment settings and the interpretation of the experiment results.**
> > >
> > > **Reply:** We want to stress that the accuracy improvement comes from the model’s parameterization. These include MIGNN with orthogonal and monotone parameterizations. MIGNN with orthogonal parameterization can effectively and stably learn LRD as the magnitude of eigenvalues of the weight matrix in orthogonal parameterization can stably converge to one during the training. Also, MIGNN with monotone parameterization is more expressive than the standard IGNN since the monotone parameterization allows the weight matrix to be in a broader range.
> > >
> > > The operator splitting schemes are used to find the fixed point of the equilibrium effectively.
> > >
> > > -----
> > >
> > > We look forward to and appreciate your further feedback.

---

> > > > ### Comment · Reviewer_Y8Ag · 2022-11-17
> > > > **Response to authors' comments**
> > > >
> > > > I thank the authors for responding to my comments and updating the draft.
> > > >
> > > > > Therefore, MIGNN with monotone parameterization provides a much more general weight matrix $W$ than IGNN.
> > > >
> > > > I understand that the sufficient condition in terms of $W$ for MIGNN to be well-posed is looser than that for IGNN.
> > > >
> > > > Q1. I agree that extending the condition of $W$ is certainly a novel result. Also, I understand why the authors used the FB scheme for the monotone parametrization and the PR scheme for orthogonal parameterization. For readability, I suggest reversing the order of Sections 3.1 and 3.2 because, in section 4, the monotone appeared first, followed by the orthogonal.
> > > >
> > > > Q2. I understand that the authors' claim is as follows: (1) When we use Picard's method, and the task requires LRD, $\lambda_1(W)$ goes to $1$, and the model converges slowly. (2) compared with IGNN, MIGNN-N1D1 has small $\lambda_1(W)$ and converges fast (Fig. 6).
> > > >
> > > > Q3. OK
> > > >
> > > > Q4. OK
> > > >
> > > > Q5. See Q1
> > > >
> > > > Q6. OK. I appreciate that the authors added the ablation study for N and D.
> > > >
> > > > Q7. See Q2
> > > >
> > > > Q8. In my understanding, the accuracy improvement is what we should think of as significant rather than novelty. I intended to use novelty to refer to methodologically new things, although this is just a matter of the choice of words.

---

> > > > > ### Author Response · Authors · 2022-11-17
> > > > > **Response to Reviewer Y8Ag’s Commets**
> > > > >
> > > > > We thank the reviewer for the time in responding to our rebuttal. We have further modified the structure of the paper as the reviewer suggested. We hope the reviewer finds our paper merits publication at ICLR. We are happy to address further comments from the reviewer.

---

> ### Author Response · Authors · 2022-11-28
> **Further follow up with Reviewer Y8Ag**
>
> Dear Reviewer Y8Ag,
>
>    First of all, we thank the reviewer for the thoughtful review of our paper and prompt response to our rebuttal. Since our last response to your comments, we have further revised our paper, including restructuring the paper, discussing more related references, and adding more numerical results.
>
>   From the reviewer's response to our rebuttal, we believe we have addressed the reviewer's concerns about our paper. We believe our paper is novel and significant based on the reviewer's feedback. We hope the reviewer finds our paper merits publication at ICLR.
>
>   We would appreciate the reviewer's further feedback. Many thanks for the effort the reviewer made and will make on our work.
>
> Regards,
>
> Authors of Paper #483

---

### Author Response · Authors · 2022-11-11
**General Response and Summary of Revision**

Dear AC and reviewers,

   Thank you for your thoughtful reviews and valuable comments, which have helped us improve the paper significantly. We are glad that reviewers found: 1) the relation between IGNN accurately learning long-range dependencies (LRD) and eigenvalues of $\mathbf{W}$ is interesting and motivates the work (h7JA), and 2) the work is strong in its math-grounded modeling technique (Ti7k). We have updated our submission based on the reviewers' feedback and highlighted our revision in blue. In particular, we have added more results on graph node classification and ablation studies, and we have substantially restructured our paper with more navigation at the beginning of each section. **In this general response, our goal is to highlight our main contributions and address some common concerns raised by some reviewers.**

-----
First of all, we fear there are some **misunderstandings of our contributions** from the reviewers, which may be due to the exposition issue of the original submission as we tried to explain too many things. Below let us briefly summarize our contribution.

***Motivation:*** Our proposed MIGNN is motivated by two observed bottlenecks of the existing IGNN: **1)** To learn LRD, IGNN has to push the magnitude of some eigenvalues of the weight matrix $\mathbf{W}$ towards 1, resulting in **slow convergence** of Picard iteration. Also, during the training of IGNN, the magnitude of eigenvalues of $\mathbf{W}$ (starting from random Gaussian initialization) may not converge to 1, resulting in **instability** in learning LRD. **2)** The parameterization used in IGNN constrains the magnitude of eigenvalues of $\mathbf{W}$ to be less than $1$, limiting the expressivity of IGNN.

***Parameterization of MIGNN:***
To address the above two issues of IGNN, we propose MIGNN with orthogonal and monotone parameterizations. **1)** We propose MIGNN with orthogonal parameterization to enhance the stability and efficiency of learning LRD --- as the magnitude of eigenvalues of orthogonal parameterization stably converges to one. **2)** We further propose monotone parameterization for MIGNN to improve the expressivity for graph learning. Notice that the magnitude of eigenvalues of $\mathbf{W}$ in IGNN has to be in the range of $[0,1)$ to guarantee the well-posedness of IGNN. In contrast, MIGNN with monotone parameterization allows the real part of eigenvalues of $\mathbf{W}$ to be in the range $(-\infty, 1)$ and the imaginary part to be arbitrary, which is much wider than that of IGNN.

***Finding the fixed-point of MIGNN with different parameterizations:***
However, **both orthogonal and monotone parameterizations are not within the efficient convergence regime of Picard iteration**, and we have to apply other numerical algorithms to find the fixed point of MIGNN with orthogonal or monotone parameterization. Particularly appealing numerical algorithms are based on Anderson-accelerated operator splitting schemes.

First, we apply the Anderson accelerated forward-backward operator splitting (FB) to find the fixed point of MIGNN with monotone parameterization, resulting in the model **MIGNN-Mon**. Let us briefly explain this choice; there are three popular operator splitting schemes: FB, Peaceman-Rachford (PR), and Douglas-Rachford (DR). PR and DR require inverting large matrices, which is inefficient and unscalable. Moreover, we cannot use the Neumann series to approximate the inversion of the matrices when the monotone parameterization is used since the parameterization has eigenvalues whose real part can be much less than $-1$. As such, we select FB to find the fixed point of MIGNN with monotone parameterization.

Second, we use Anderson accelerated PR, with some approximations, to find the fixed point of MIGNN with orthogonal parameterization. The rationale of this choice is that among the three choices, PR is better than DR, as explained in Remark 2 in our revised paper. FB requires stronger conditions for convergence guarantee and converges slower than PR in general. When orthogonal parameterization is used, we can use the Neumann series to approximate the matrix inversion; see Section 4.1.2 for details. Moreover, we can further improve the performance of MIGNN with orthogonal parameterization using higher-order graph diffusion convolution. We name MIGNN with orthogonal parameterization using PR splitting accompanied by $K$-th order Neumann series approximation and $P$-th order graph diffusion convolution **MIGNN-N$K$D$P$**.

***Performance:***
We have compared the performance of MIGNN with the benchmark IGNN and several state-of-the-art GNNs on various benchmark graph classification tasks at both node and graph levels. The numerical results resonate with our theoretical results. In particular, the extensive numerical results consistently show that MIGNN significantly outperforms the benchmark IGNN, and often MIGNN achieves the best results among all the baseline models.

---

> ### Author Response · Authors · 2022-11-11
> **General Response and Summary of Revision (cont'd)**
>
> ----
>
> The reviewers also have some concerns about the computational time comparison between MIGNN and IGNN. We want to emphasize that we have provided the computational time comparison between **MIGNN-N$1$D$1$** and IGNN on the Amazon co-purchasing dataset, which is often used as a large LRD learning benchmark where MIGNN with orthogonal parameterization saves significantly on the number of iterations and computational time over IGNN; see Figure 6 for details. The fact that MIGNN with orthogonal parameterization can learn LRD more efficiently is also confirmed in the newly added experiment where **MIGNN-N$2$D$5$** and IGNN are contrasted on classifying directed chains; see Figure 4 in the revision for the comparison of IGNN with MIGNN-N$2$D$5$ in classification accuracy and computational time. We also provide extensive **ablation studies** of the impact of the order of the hyperparameter $K$ and $P$ on computational time for graph node and graph classification accuracy in Appendix G and H, respectively. Furthermore, we have included the time complexity discussion at the end of Section 4.1.2 in the revision.
>
> -----
>
> Incorporating the comments and suggestions from all reviewers, we have made the following changes in the revised paper besides fixing typos and notations.
>
> - We add to the revision Section 5.2 and Table 1, which contain node classification results on Cora, Citeseer, and Pubmed datasets.
>
> - In Table 2, we add the graph classification results from GSN [16], SIN [15], and CIN [14].
>
> - We provide extensive ablation studies on the impact of the order of the Neumann series and graph diffusion convolution on graph node and graph classification accuracy and computational time in Appendix G and H, respectively.
>
> - We include the computational complexity of MIGNN in the revision.
>
> - In Figure 4, we add a comparison on the classification of three class chains of length 140 between **MIGNN-N$2$D$5$** and IGNN where the efficiency and robustness of  **MIGNN-N$2$D$5$** are clearly demonstrated.
>
> - We revise the writing to improve exposition; for example, we clarify the bottlenecks of IGNN and how we address them in the introduction section, and we have clarified when we chose between **MIGNN-N$K$D$P$** and **MIGNN-Mon**. We have also added more navigation at the beginning of each section to help the reader to understand the paper.
>
> - In the newly added appendix K, we include details about the hyper-parameters used in our model.
>
> - We have added some references suggested by the Reviewers.
>
> -----
>
> [14] Cristian Bodnar, Fabrizio Frasca, Nina Otter, Yuguang Wang, Pietro Liò, Guido F Montufar, and Michael Bronstein. Weisfeiler and Lehman go cellular: CW networks. Advances in Neural Information Processing Systems, 34:2625–2640, 2021.
>
> [15] Cristian Bodnar, Fabrizio Frasca, Yuguang Wang, Nina Otter, Guido F Montufar, Pietro Liò, and Michael Bronstein. Weisfeiler and Lehman go topological: Message passing simplicial networks. In International Conference on Machine Learning, pp. 1026–1037. PMLR, 2021.
>
> [16] Giorgos Bouritsas, Fabrizio Frasca, Stefanos P Zafeiriou, and Michael Bronstein. Improving graph neural network expressivity via subgraph isomorphism counting. IEEE Transactions on Pattern Analysis and Machine Intelligence, 2022
>
>
> -----
>
> We are glad to answer any further questions you have on our submission.
>
> -----

---

### Decision · Program_Chairs · 2023-01-20

**Decision:**

Reject

**Justification For Why Not Higher Score:**

The evaluation of the paper is mixed. The methodology is interesting, but its performance is not fully demonstrated. The efficiency of the proposed method in practice is not verified yet.

**Justification For Why Not Lower Score:**

N/A

**Metareview: Summary, Strengths And Weaknesses:**

In this paper, the authors proposed an implicit graph neural network (IGNN), which is a new member of deep implicit layers.
Essentially, the proposed method designs a graph neural network layer by solving a fixed point equilibrium equation.
Some techniques are used to relax the constraints of existing IGNNs, e.g. applying flexible step sizes and an orthogonalization strategy.
Compared with existing GNNs based on spectral theory or spatial message passing, the proposed IGNN can take the long-range dependency among different nodes by a single optimization-driven layer rather than stacking multiple explicit layers.
The authors test the proposed method in various graph-related tasks, but the performance on node-level tasks is not as good as the authors claimed.

Strengths:
(1) The study of the deep implicit layer is an important topic. Designing GNN as an implicit layer is interesting and may attract lots of researchers in this community.
(2) To my knowledge, although the application of the monotone operator theory has been considered by other non-graph learning scenarios, it is the first attempt to apply it to GNN.
(3) Some efforts have been made to supporess the numerical issues (e.g., slow convergence and instability) of existing implicit modeling methods.

Weaknesses:
(1) The writing and the organization of this submission are questionable. The logic flow of this paper, especially the technical part, is not clear.
(2) The experimental results in the submission are not very strong. The performance on node-level tasks is not as good as the state-of-the-art method. The claims on the efficiency of the proposed method are not convincing.
(3) The technical contributions are incremental to some degree, which do not lead to significant improvements compared with existing IGNN work.

**Summary Of Ac-Reviewer Meeting:**

In the rebuttal phase, the reviewers proposed some comments about the solidness of the experiments and the novelty of the methodology. The authors made efforts to resolve the concerns, including adding more experiments, adding more explanations on the principle of the method, and highlighting some analytic content to explain their technical contributions to make implicit graph neural network work. The reviewers are satisfied with the replies, but some of them proposed some new comments, and none of them changed their scores yet.

AC had a virtual meeting with 3 of the 4 reviewers.

Reviewer Y8Ag: The authors solved most of the reviewer's concerns, and the reviewer will raise the score from 5 to 6. However, the significance and usefulness of applying the monotone theory to IGNN are not verified by the experiments.

Reviewer Ti7k: (1) The problem (implicit graph layer) and the monotone theory are not new, but applying the theory to graph learning scenarios is new. (2) Whether the proposed technical route is important and useful for graph learning is questionable --- some drawbacks on experiments (e.g., node-level classification) and the methodology itself are hard to solve at the current stage. The reviewer tends to give it 6 points.

Reviewer h7JA: Mostly agree with Reviewer Ti7k. Additionally, the reviewer thinks that the main technical contribution of this work is relaxing the constraints of existing IGNNs, e.g., using flexible step sizes and applying an orthogonalization method, but these modifications are incremental in the aspect of performance. The reviewer tends to give it 6 points.

Overall, AC and the reviewers agree that it is a paper marginally above the borderline. AC tends to reject this work because none of the reviewers strongly support it.